**TOOLS**

# Genetically encoded live-cell sensor for tyrosinated microtubules

Shubham Kesarwani[1,3], Prakash Lama[1,3], Anchal Chandra[2], P. Purushotam Reddy[2], A.S. Jijumon[4,5], Satish Bodakuntla[4,5], Balaji M. Rao[6], Carsten Janke[4,5], Ranabir Das[2], and Minhajuddin Sirajuddin[1]

**Microtubule cytoskeleton exists in various biochemical forms in different cells due to tubulin posttranslational modifications (PTMs). Tubulin PTMs are known to affect microtubule stability, dynamics, and interaction with MAPs and motors in a specific manner, widely known as tubulin code hypothesis. At present, there exists no tool that can specifically mark tubulin PTMs in living cells, thus severely limiting our understanding of their dynamics and cellular functions. Using a yeast display library, we identified a binder against terminal tyrosine of α-tubulin, a unique PTM site. Extensive characterization validates the robustness and nonperturbing nature of our binder as tyrosination sensor, a live-cell tubulin nanobody specific towards tyrosinated microtubules. Using this sensor, we followed nocodazole-, colchicine-, and vincristine-induced depolymerization events of tyrosinated microtubules in real time and found each distinctly perturbs the microtubule polymer. Together, our work describes a novel tyrosination sensor and its potential applications to study the dynamics of microtubule and their PTM processes in living cells.**

## Introduction

Microtubules are cytoskeleton tubular polymers that perform diverse cellular functions, including (but not limited to) intracellular cargo transport, chromosome segregation, and cell motility. These cellular processes are mediated by interactions between microtubules and a cohort of molecular motors and microtubule-associated proteins (MAPs). A key regulatory process that governs microtubule interaction with its cognate proteins is the diversity of tubulin genes and their variety of posttranslational modifications (PTMs; Janke and Magiera, 2020). Most tubulin PTMs are strictly reversible, controlled by modifying and reverse enzymes. Defects in the balance of these enzymes lead to abnormal levels of microtubule PTMs that are manifested in different disease pathologies (Magiera et al., 2018b), including neurodegeneration (Magiera et al., 2018a) and cardiomyopathies (Chen et al., 2018; Robison et al., 2016).

Among the tubulin PTMs, the tyrosination–detyrosination cycle at the α-tubulin C-terminal site was the first PTM reported (Arce et al., 1975; Barra et al., 1973) and was later reported in metazoans, ciliates, and flagellates. The genetically encoded C-terminal tyrosine residue can be enzymatically removed by vasohibin–SVBP (small vasohibin-binding protein) complexes, a recently identified detyrosinase (Aillaud et al., 2017; Nieuwenhuis et al., 2017). The tubulin tyrosine ligase (TTL; Ersfeld et al., 1993) reverses the detyrosination modification by adding tyrosine back to the terminal site of α-tubulin (Barra et al., 1973). Over the years, several tubulin PTMs, such as acetylation (L'Hernault and Rosenbaum, 1985), glutamylation (Eddé et al., 1990), and glycylation (Redeker et al., 1994), and their respective enzymes have been identified across species. These PTMs, with the exception of acetylation, occur at the C-terminal tails (CTTs) of either α- and/or β-tubulin gene products. The PTMs can also be combinatorial, overlapping with the diverse tubulin gene products and creating diverse biochemical forms of microtubules across cell types (Janke and Magiera, 2020), which makes tubulin PTM studies a challenging prospect. Recent advances in protein engineering and expression have allowed for the creation of homogenous microtubules with a particular PTM (Minoura et al., 2013; Sirajuddin et al., 2014; Souphron et al., 2019; Ti et al., 2018; Valenstein and Roll-Mecak, 2016; Vemu et al., 2014). This, in turn, has allowed for in vitro reconstitution studies that have highlighted how single PTMs can uniquely modulate molecular motors (Barisic et al., 2015; McKenney et al., 2016; Nirschl et al., 2016; Sirajuddin et al., 2014), MAPs (Bonnet

[1]Centre for Cardiovascular Biology and Diseases, Institute for Stem Cell Science and Regenerative Medicine, Gandhi Krishi Vigyan Kendra Campus, Bangalore, India; [2]National Center for Biological Sciences, Tata Institute of Fundamental Research, Gandhi Krishi Vigyan Kendra Campus, Bangalore, India; [3]Manipal Academy of Higher Education, Manipal, Karnataka, India; [4]Institut Curie, Paris Sciences et Lettres University, Centre National de la Recherche Scientifique UMR3348, Orsay, France; [5]Université Paris Sud, Université Paris-Saclay, Centre National de la Recherche Scientifique UMR3348, Orsay, France; [6]Department of Chemical and Biomolecular Engineering, North Carolina State University, Raleigh, NC.

Correspondence to Minhajuddin Sirajuddin: minhaj@instem.res.in.

et al., 2001), and severing enzymes (Lacroix et al., 2010; Valenstein and Roll-Mecak, 2016), providing first insights into the regulatory roles of tubulin diversity.

In light of these emerging functions of tubulin PTMs, the burning question of how they are dynamically generated and organized in living cells arises. Microtubule populations in cells can carry different tubulin PTMs side by side (Tas et al., 2017), and they can carry combinations of different PTMs at the same time. For example, the long-lived microtubules have been frequently shown to be highly detyrosinated and acetylated (Bulinski et al., 1988; Schulze et al., 1987; Webster and Borisy, 1989; Webster et al., 1987b). Similarly, glutamylation and glycylation can occur at multiple sites of the same tubulin CTTs and can coexist in axonemal microtubules (Wloga et al., 2017). A typical cellular or in vivo study of tubulin PTMs involves labeling microtubules using antibodies specific toward the respective PTM epitopes, such as the tyrosinated, detyrosinated, glutamylated, and glycylated states of microtubules (van Dijk et al., 2007; Gadadhar et al., 2017; Gundersen et al., 1984; Janke, 2014; Kilmartin et al., 1982). Although these antibodies have illuminated the tubulin PTMs in different cell types and organisms, it severely limits our understanding of the spatiotemporal component of tubulin PTMs. Therefore, a cellular sensor that can detect and track tubulin PTMs in real time will aid in studying their dynamics and function in vivo.

In general, the most common methods to label microtubules in living cells either involve fluorescently tagged α-tubulin (Gierke et al., 2010; Kamath et al., 2010; Rusan et al., 2001), MAPs (Bulinski et al., 1999) or silicon rhodamine (SiR)-tubulin, a Taxol derivative (Lukinavičius et al., 2014). None of these methods, in any case, can distinguish different types of microtubules. This could be achieved with nanobodies or single-chain antibodies, which have been successfully employed to study PTMs on other proteins (Helma et al., 2015), but without much success against microtubules (Traenkle and Rothbauer, 2017). So far, two studies have attempted in this direction. One study used nanobodies against microtubules to reconstruct superresolution structures of microtubules (Mikhaylova et al., 2015). Another study has reported the identification of a single-chain antibody (anti-tubulin single-chain variable fragment; scFv) against tyrosinated microtubules (Cassimeris et al., 2013). However, the nanobody could not be employed in living cells (Mikhaylova et al., 2015), and no further study of anti-tubulin scFv application has been reported to date. Altogether, there is a severe dearth of tools that can mark generic microtubules and/or tubulin PTMs in living cells.

To overcome this, we screened a yeast display library (Gera et al., 2012) against α-tubulin CTT and identified a binder molecule. We demonstrate that this binder is specific toward the tyrosinated state of tubulin and does not interfere with the cellular or microtubule-based functions when expressed in living cells. The tyrosination sensor reported here, therefore, becomes the first thoroughly characterized tubulin nanobody that can be employed to specifically follow tyrosinated or unmodified microtubules in living cells.

## Results

### Strategy for screening binders against tyrosinated microtubule

Several studies have successfully employed tubulin CTT peptides as epitopes to identify antibodies specific for a particular tubulin PTM (Bré et al., 1996; Gadadhar et al., 2017; Gundersen et al., 1984; Paturle-Lafanechère et al., 1994). Keeping this in mind, we synthesized the C terminus of TUBA1A (amino acids 440–451) with a biotin at the N terminus (biotin-TUBA1A 440–451). To obtain a binder protein specific for biotin-TUBA1A 440–451 (termed *Hs_*TUBA1A), we employed a combinatorial yeast display library of SSO7D mutants screen as described previously (Gera et al., 2012). To select binders that are specific for the tyrosinated form, we also performed a negative selection of SSO7D library against biotin-TUBA1A 440–450 (detyrosinated, termed *Hs_*TUBA1A-ΔY) and biotin-TUBA1A 440–451-[E] 445 (monoglutamylated at glutamate residue 445, termed *Hs_*TUBA1A-mG) peptides (Materials and methods; Fig. 1 A).

After FACS experiment enrichment, 10 single yeast colonies were analyzed to identify the abundance of enriched clones (Materials and methods; Fig. S1 A). Among them, two yeast clones, A1aY1 and A1aY2, represented 30% and 20% enrichments, respectively (Fig. S1 B), which were purified as recombinant GFP-tagged fusion proteins and subjected to binding experiments with the *Hs_*TUBA1A peptide (Materials and methods). The binding experiments strongly indicated that only A1aY1-GFP showed a positive response toward *Hs_*TUBA1A peptide (Materials and methods; Fig. S1 C).

### Biochemical and structural characterization of the A1aY1 binder

We further purified the A1aY1 binder without any tags (Materials and methods) and subjected it to titration experiments with *Hs_*TUBA1A, *Hs_*TUBA1A-ΔY, and *Hs_*TUBA1A-mG peptides, representing tyrosinated, detyrosinated, and monoglutamylated forms of TUBA1A CTTs (Materials and methods). The A1aY1 binds with a dissociation constant ($k_d$) of 1.6 μM, >60 μM, and 13.6 μM to tyrosinated, detyrosinated, and monoglutamylated TUBA1A CTT peptides, respectively (Fig. 1 B). The biochemical data strongly indicate the importance of tyrosine at the TUBA1A CTT peptide for recognition by the A1aY1 binder. Furthermore, the 10-fold reduced affinity toward tyrosinated monoglutamylated peptide suggests that in addition to terminal tyrosine, the A1aY1 binder might have interactions with the glutamate residues along the *Hs_*TUBA1A peptide, which can be perturbed by glutamylation modification.

To better understand the interaction of *Hs_*TUBA1A with A1aY1 binder, we performed nuclear magnetic resonance (NMR) experiments to gain 3D structural information (Materials and methods; Table 1). The highest-ranked ensemble structure shows that the *Hs_*TUBA1A binding site overlap with the diversified regions of SSO7D protein (Fig. 1 C and Fig. S1, D–H). The key interacting residues from *Hs_*TUBA1A peptide include Y451, E449, E447, and E445, the most common glutamylation site on brain α-tubulin (Fig. 1, C and D; Eddé et al., 1990). The terminal tyrosine (Y451, nth residue) is latched with the aid of L32 and Y29 residues of A1aY1 binder. Additionally, the side-chain

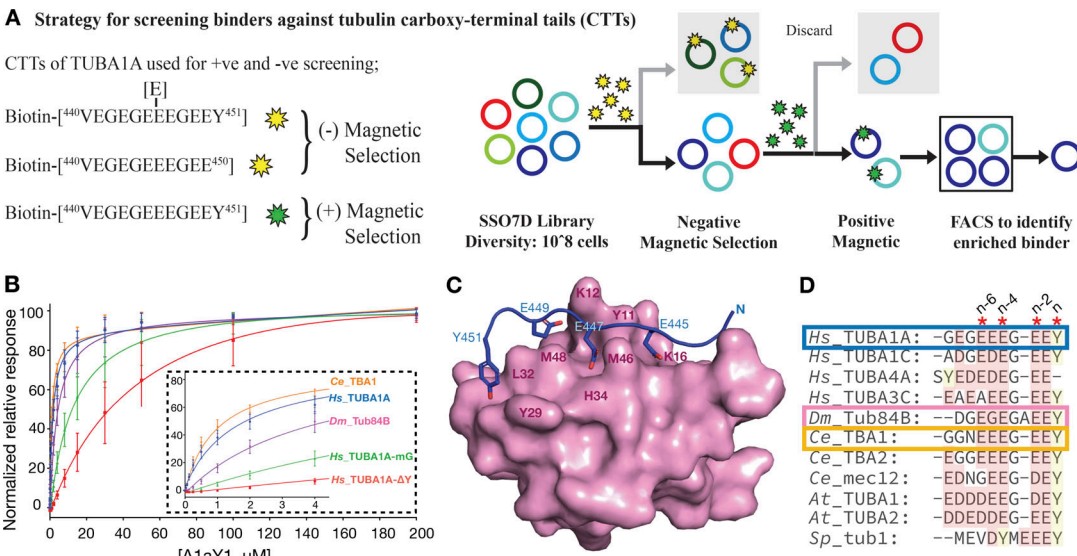

Figure 1. **Identification and biochemical and structural characterization of A1aY1 binder. (A)** Schematic overview of the strategy employed to identify binder from SSO7D yeast display library. The biotinylated TUBA1A with Y (green star) and mono-Glu and ΔY (yellow star) used for positive and negative selection, respectively. For a detailed description, see Materials and methods. **(B)** $k_d$ of A1aY1 binder against biotinylated α tubulin CTT peptides of human (Hs_TUBA1A; 1.6 ± 0.3 μM, Hs_TUBA1A-ΔY; >60 μM*, Hs_TUBA1A-mG; 13.6 ± 4.5 μM), Drosophila (Dm_Tub84B; 4.1 ± 0.9 μM) and C. elegans (Ce_TBA1; 1.0 ± 0.2 μM) measured using an SPR-based steady-state binding assay. Experiments were performed in triplicate with at least two different batches of the protein A1aY1 (asterisk represents that the binding affinities cannot be uniquely determined with the current fit). The inset shows the titration response up to 4 μM A1aY1 binder concentration. **(C)** The NMR structure of A1aY1 binder (magenta surface representation) bound to Hs_TUBA1A peptide (blue cartoon with key residues as stick representation). The key interacting residues from A1aY1 binder and α tubulin CTT are labeled in magenta and blue, respectively. **(D)** Sequence alignment of α tubulin CTTs from human (Hs), Drosophila (Dm), C. elegans (Ce), Arabidopsis thaliana (At), and Schizosaccharomyces pombe (Sp) with gene names as indicated. Asterisks indicate the residues involved in A1aY1 binder interaction, and the terminal tyrosine and alternating glutamate residues are indicated as n series.

hydroxy group of Y451 makes a hydrogen bond with the main-chain amino group of the G30 residue of A1aY1 binder (Fig. 1 C and Fig. S1, D–H). The remaining glutamic acid residues of Hs_TUBA1A peptide, E449 (n-second), E447 (n-fourth), and E445 (n-sixth) make alternative electrostatic contacts with K12, H34, and K16 of the A1aY1 binder, respectively (Fig. 1 C). This shows the capacity of A1aY1 binder to interact with all the potential glutamylation or glycylation modification sites of α-tubulin CTT.

Guided by the structure, we then compared the CTTs of human, Drosophila melanogaster, worm, plant, and fission yeast α-tubulins (Fig. 1 D). The terminal tyrosine (nth residue) and the alternating glutamic acids (n-second, n-fourth, and n-sixth residues) are conserved across different species (Fig. 1 D), suggesting that our binder could detect microtubules in many of these species. To test this, we titrated α-tubulin CTT peptides from Drosophila Dm_Tub84B and Caenorhabditis elegans Ce_TB1A against our A1aY1 binder (Materials and methods). The $k_d$ values obtained were similar to what we had previously measured for the Hs_TUBA1A peptide: 4.0 μM for Dm_Tub84B, and 1.0 μM for Ce_TB1A (Fig. 1 B).

In summary, the A1aY1 binder recognizes α-tubulin CTTs from different organisms, and the terminal tyrosine residue is an important element of this interaction.

**A1aY1 binder labels in vitro and cellular microtubules**
To check if A1aY1 can bind to microtubules, we performed in vitro labeling experiments with purified A1aY1-GFP on microtubules

assembled from HeLa tubulin, which contains pure tyrosinated form of tubulin (Souphron et al., 2019). Using this assay, we then tested recombinant A1aY1-GFP binding to microtubules that contain varying levels of tyrosinated tubulin, assembled from mixtures of different amounts of tyrosinated (HeLa tubulin) and detyrosinated tubulin (tubulin treated with carboxypeptidase A [CPA]; Fig. 2, A and B; Fig. S4 E; Materials and methods). Quantification of GFP fluorescence intensity over tyrosinated versus the differential percentage of detyrosinated microtubules showed a linear gradation in A1aY1 binding as a function of tyrosination levels of a given microtubule (Fig. 2, A and B).

Using the recombinant A1aY1-GFP protein, we next stained fixed U2OS and H9C2 cells to compare its labeling efficiency with commercially available antibodies specific towards tyrosinated and detyrosinated tubulin (Materials and methods). In the case of H9C2 cells, microtubules that show elevated levels of detyrosination staining show diminished A1aY1 staining (Fig. 2 C), similar to the results with the tyrosinated antibody. In contrast, U2OS cells, which are abundant with the tyrosinated form of tubulin, showed a majority of microtubules stained with both A1aY1 and anti-tyrosinated tubulin antibody but reduced staining with anti-detyrosinated tubulin antibody (Fig. 2 C). Thus, the A1aY1 binder could serve as a tool to quantify tyrosination/detyrosination levels in microtubules.

So far, our biochemical and structural experiments with A1aY1 binder were with purified proteins. To check whether

Table 1. **NMR and refinement statistics of the Binder/α-tubulin complex**

| NMR restraints | |
|---|---|
| Unambiguous restraints (intermolecular NOEs) | 58 |
| Ambiguous restraints (CSPs) | 10 |
| **HADDOCK parameters** | |
| Cluster Size | 200 |
| HADDOCK score | −71.7 (± 1.2) |
| Van Der Waals energy | −34.2 (± 3.4) |
| Electrostatic energy | −262.1 (± 27.8) |
| Restraints violation energy | +4.6 (± 0.7) |
| Buried surface area | +1163.3 (± 25.1) |
| All backbone | 0.4 |
| All heavy atoms | 0.6 |
| **RMS deviations**[a] | |
| Bond angles | 0.6° |
| Bond lengths | 0.004 Å |
| Molprobity clashscore[b] | 2.5 (99th percentile) |
| **Ramachandran statistics**[a] | |
| Most favored (%) | 90.8 |
| Additionally allowed (%) | 9.1 |
| Generously allowed (%) | 0.1 |
| Disallowed (%) | 0.0 |

[a]Calculated for an ensemble of 20 lowest energy structures.
[b]Calculated for the lowest energy structure.

A1aY1 binder can recognize microtubules in living cells, we generated a series of fluorescent protein fusion constructs and transiently expressed them in U2OS cells (Materials and methods). Among them, only TagBFP and TagRFP-T functioned equally well as N- and C-terminal fusions with the A1aY1 binder (Fig. 3 and Fig. S2). We then quantified the expression levels of A1aY1 binder both in low/medium- and high-expressing cells and found them to be roughly stoichiometric to the cellular tubulin levels (Fig. S3). To determine the behavior of these sensors in living cells, we generated U2OS cell lines stably expressing both A1aY1 fused to either TagBFP and TagRFP-T (called blue and red A1aY1 sensor, respectively; Fig. 3; Materials and methods). While high expression levels of the sensors led to occasional microtubule bundling in cells (Fig. S3), medium to low-level expression of the blue and red A1aY1 sensor did not show such artifacts, thus offering the application of the binder as a live-cell sensor for microtubules (Fig. 3 and Fig. S3).

## Specificity of A1aY1 binder toward tyrosinated microtubules

Our biochemical results show that A1aY1 binder is specific for tyrosinated microtubules (Fig. 2 A). We next compared its specificity toward tyrosinated, detyrosinated, and glutamylated microtubules in cells. U2OS cells stably expressing the red A1aY1 sensor were transfected with detyrosinase (Aillaud et al., 2017;

Nieuwenhuis et al., 2017) and polyglutamylation enzymes (van Dijk et al., 2007; Materials and methods). For detyrosinase, we used VASH2_X1+SVBP enzyme complex, as we found that VASH2_X1 an isoform of VASH2 with a particularly elevated enzymatic activity when overexpressed in cells (Fig. S4, A–D). We further expressed the glutamylases TTLL5, TTLL4, and TTLL7, as well as a catalytically inactive version of TTLL5 (van Dijk et al., 2007).

Transfection of detyrosinase in the cells stably expressing the A1aY1 red sensor leads to complete loss of microtubule labeling by the sensor, which confirms its specificity toward tyrosinated tubulin (Fig. 4, A and B). Expression of TTLL5 also leads to a strong reduction of A1aY1 sensor binding to microtubules, which is not the case when the catalytically inactive version of this enzyme is expressed (Fig. 4, A and B). Similarly, TTLL4, an enzyme that initiates glutamylation, also leads to strong reduction in A1aY1 binding to microtubules (Fig. 4, A and B). Conversely, expression of TTLL7, a β-tubulin–specific polyglutamylase (van Dijk et al., 2007), did not affect the labeling of cellular microtubules by A1aY1 sensor (Fig. 4, A and B). Our observations indicate that glutamylation of the α-tubulin CTT sterically interferes with the binding of the A1aY1 sensor (Fig. 4, C–E). This concours with our biochemical experiments using peptides mimicking the α-tubulin CTTs with different PTMs (Fig. 1 B) and our structural findings (Fig. 4, C–E).

A nanobody to tyrosinated tubulin, 2G4C-ScFv, had been previously reported to recognize tyrosinated microtubules (Cassimeris et al., 2013). To directly compare with our A1aY1 sensor, we cloned 2G4C-ScFv into an equivalent expression vector and observed its binding to tyrosinated, detyrosinated, and glutamylated microtubules in cells. Strikingly, 2G4C labeled microtubules in all PTM states (Fig. 5), thus underpinning the uniqueness of the A1aY1 binder as a tool to specifically label tyrosinated microtubules in living cells.

In summary, our biochemistry and specificity experiments unequivocally suggest that A1aY1 sensor specifically recognizes tyrosinated microtubules. Therefore, we hereafter refer A1aY1 binder as tyrosination sensor.

## Tyrosination sensor does not alter the cellular function of microtubules

A hallmark property of microtubules is their ability to undergo dynamic instability, which is essential for many of their cellular functions, such as mitotic spindle organization and chromosome segregation (Vicente and Wordeman, 2019). Therefore, to validate the tyrosination sensor as a live-cell marker, we tested whether it interferes with microtubule dynamics and function. We first checked the viability and proliferative ability of cells stably expressing blue and red tyrosination sensor (Materials and methods). Trypan blue assays, propidium iodide (PI), and DAPI-based flow cytometric analysis for both blue and red sensor–expressing stable lines show that ~95% of the cells are viable and in their proliferative state (Fig. S5, A and B). We further observed all the mitotic stages and imaged them using our tyrosination sensor (Fig. S5 C), thus confirming that constitutive expression of tyrosine sensor does not interfere with cell viability and division.

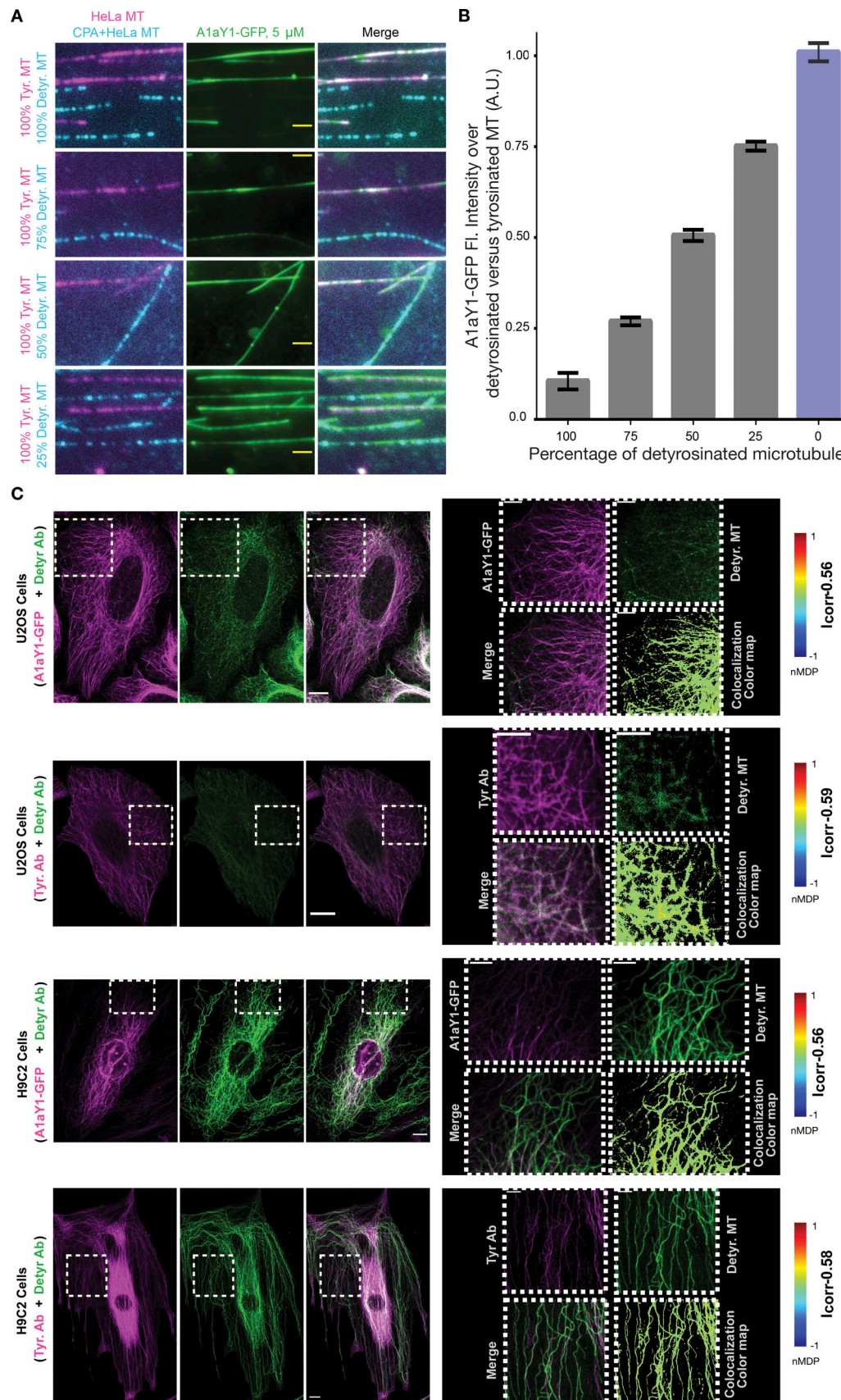

Figure 2. **Purified A1aY1 binder labels in vitro polymerized microtubules and fixed cellular microtubules. (A)** Representative TIRF images of 5 µM A1aY1-GFP (green) bound to tyrosinated (magenta) and detyrosinated (cyan) microtubules with varying percentage of detyrosinated tubulin (CPA treated)

incorporated during polymerization. Scale bars = 2 µm. **(B)** Mean ratio ± SEM of A1aY1-GFP fluorescence intensity bound to tyrosinated versus varying percentage of detyrosinated microtubules. Blue column bar represents pure tyrosinated microtubules. Data are derived from total 150 filaments analyzed for 100%, 75%, 50%, and 25% CPA microtubules (and 167 filaments from 0% CPA microtubules, $n$ = 3) from $n$ = 4 different experiments with two different batches of A1aY1 protein. **(C)** A1aY1-GFP staining and immunostaining of U2OS and H9C2 cells with anti-tyrosination and anti-detyrosination antibodies as indicated. Scale bars = 10 µm for the whole cell and 5 µm for the zoom panel. A.U., arbitrary units; MT, microtubule.

Next, we measured microtubule dynamics in cells expressing the red tyrosination sensor (Materials and methods). Time-lapse images show that the microtubule polymerization–depolymerization events can be followed by TagRFP-T_A1aY1 fluorescence signal. Microtubules labeled with TagRFP_T-A1aY1 undergo typical dynamic instability states of growth, pause, catastrophe, and rescue events (Fig. 6 A, Fig. S5, D–G; and Video 1), suggesting that the tyrosination sensor does not interfere with microtubule dynamics per se. Manual tracking of individual microtubule shows a threefold difference in between growth and depolymerization rates (Fig. S5 D), in line with studies reporting microtubule dynamics in cells (Kamath et al., 2010; Komarova et al., 2002; Picone et al., 2010; Zwetsloot et al., 2018). To further quantify the growth rates, we then transfected EB3-GFP and imaged growing microtubules plus-end via EB3 comets (Fig. 6 B, Fig. S5, H–J; Video 2; Materials and methods). Microtubules grew at 0.36 ± 0.1 µm/s ($n$ = 468) in the presence of the tyrosination sensor, which is almost identical to the rate measured in control cells (0.35 ± 0.09 µm/s [$n$ = 450]). Simultaneously, we measured EB3-GFP comets in the presence of 0.5 µM SiR-tubulin, which shows a significant decrease in

growth rates (0.19 ± 0.06 µm/s [$n$ = 414]), compared with the untreated cell or cells with the tyrosination sensor (Fig. 6 B and Fig. S5 J). Our live-cell imaging with and without EB3-GFP comets also reveals that the tyrosination sensor signal disappears promptly during microtubule depolymerization events (Fig. 6 A, Video 1, Video 2, and Video 3). Therefore, the tyrosination sensor can be used to follow microtubule polymerization and depolymerization events without affecting microtubule growth rates and dynamics (Video 1, Video 2, and Video 3).

To test if the binding of the tyrosination sensor interferes with plus-end tracking proteins that are sensitive to tyrosination state of α-tubulin, we analyzed the localization of CLIP170 (Bieling et al., 2008; Peris et al., 2006). Immunostaining of CLIP170 with a specific antibody (Coquelle et al., 2002) showed no difference between cells that expressed the tyrosination sensor and control cells (Fig. 6 C). In contrast, in cells expressing the detyrosinase enzyme (VASH2_X1 + SVBP) CLIP170 was undetectable at the microtubule plus ends, thereby confirming that endogenous CLIP170 requires tubulin tyrosination for localization to the plus end of microtubules (Nirschl et al., 2016; Peris

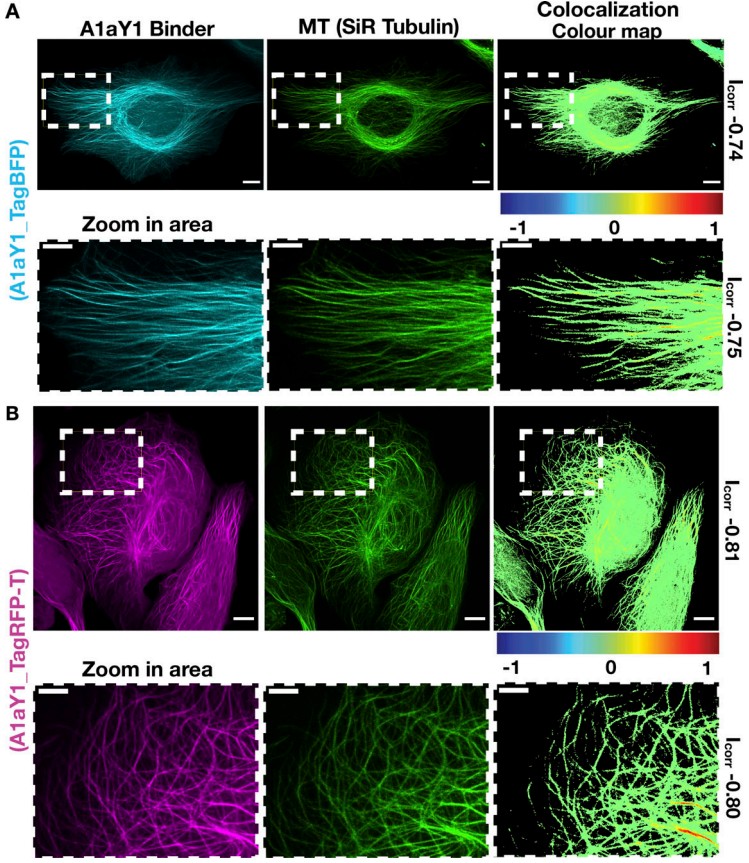

Figure 3. **Labeling cellular microtubules using A1aY1 fused to TagBFP/TagRFP-T.** Z-projection of confocal image stacks of U2OS cells stably expressing A1aY1 binder tagged with TagBFP (A) or TagRFP-T (B) at the N terminus, in cyan and magenta, respectively. The cells were additionally stained with SiR-tubulin (in green). The zoom panel shows a closer view of microtubules labeled with A1aY1 TagBFP and TagRFP-T as indicated. The colocalization colormap between A1aY1 binder versus SiR-tubulin is based on a color scale in which negative normalized mean deviation product values are represented by cold colors (segregation); values above 0 are represented by hot colors (colocalization). Scale bars = 10 µm and 5 µm for the whole cell and zoomed panel, respectively. MT, microtubule. Icorr respresents index of correlation measured from the colormap Fiji plugin.

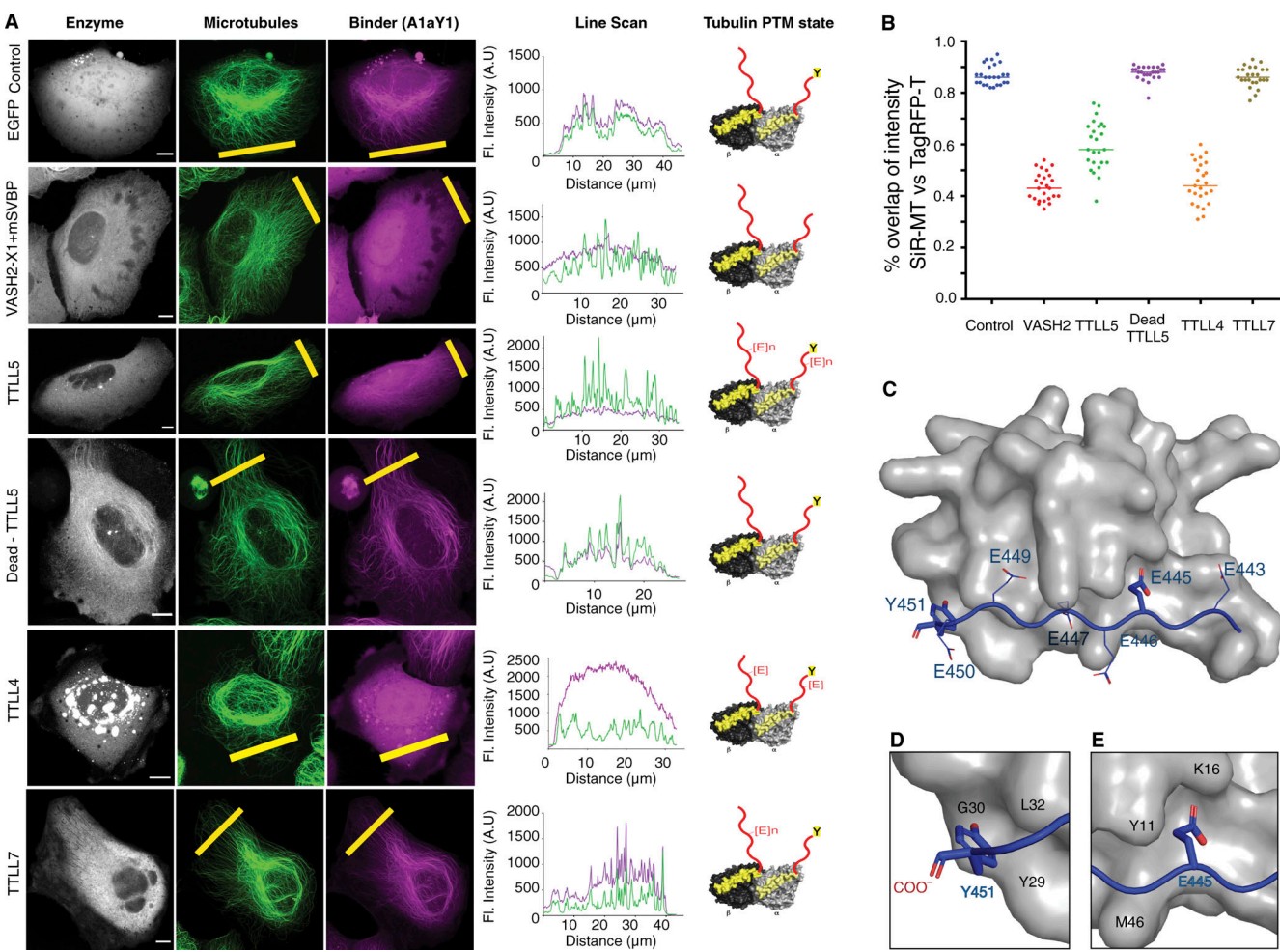

Figure 4. **Specificity of A1aY1 binder toward tyrosinated microtubules. (A)** Confocal image stacks of U2OS cells stably expressing TagRFP-T A1aY1, indicated as "Binder" (magenta) along with GFP control, vasohibin2-X1-GFP-2A-SVBP, TTLL5-EYFP, catalytically dead TTLL5-EYFP, TTLL4-EYFP, and TTLL7-EYFP, indicated as "Enzyme" (gray) along with SiR-tubulin, indicated as "Microtubules" (green). Scale bars = 10 μm. Line scans of SiR-tubulin (green) and TagRFP-T A1aY1 (magenta) fluorescence intensity signal for each panel as indicated by the yellow line. Cartoon representation of α/β-tubulin in the right panel indicates the PTM state of microtubules for the respective experiment. **(B)** Quantification of Pearson's coefficient (R-value) to calculate the correlation for a fraction of microtubules detected by TagRFP-T A1aY1 versus SiR-tubulin label, represented as "% overlap of intensity SiR-tubulin versus TagRFP-T." A typical diffused cytoplasmic signal will have ~60% or less overlap, whereas a complete colocalization will show near 100% overlap. n = 25 cells; each dot represents data from one cell. **(C)** NMR structure of A1aY1 (gray surface representation) and α-tubulin CTT peptide (blue cartoon representation); the C-terminal, tyrosine, and glutamic acid residues are indicated. **(D and E)** Closer view of the terminal tyrosine (Y451) and polyglutamylation site glutamic acid (E445) with key interacting residues from A1aY1 binder as indicated. A.U., arbitrary units; Fl., fluorescence; MT, microtubule.

et al., 2006). This demonstrates that the presence of the tyrosination sensor does not impede the microtubule interactions with proteins that bind specifically to tyrosinated tubulin.

Microtubules also function as tracks for motor proteins, which facilitates intracellular cargo transport. To examine if the tyrosination sensor interferes with motor movement along microtubules, we performed in vitro motility experiments using kinesin-1 (K560-SNAP) or kinesin-3 (KIF1A 1-393LZ-SNAP) on HeLa microtubules (Souphron et al., 2019) marked with the recombinant A1aY1-GFP as tyrosination sensor (Materials and methods). Single-molecule experiments show that both kinesin-1 and kinesin-3 do not show any major deviations from their normal motility behavior (Fig. 6, D and E).

Combinedly these experiments strongly suggest that the tyrosination sensor does not affect microtubule properties or their molecular interactions and related cellular function, thus validating the suitability of our sensor for live-cell experiments.

## Live-cell imaging and mechanism of drugs that target microtubules

To explore the suitability of our tyrosination sensor in studying microtubules in live cells, we employed drugs such as nocodazole, colchicine, and vincristine, which are known to target microtubules and are commonly used in cell biology studies to perturb microtubules. Although there are several reports about the mode of drugs action, so far, the depolymerization events they induce have seldom been studied in real time.

U2OS cells stably expressing the red tyrosination sensor were individually treated with 10 μM nocodazole, 500 μM colchicine,

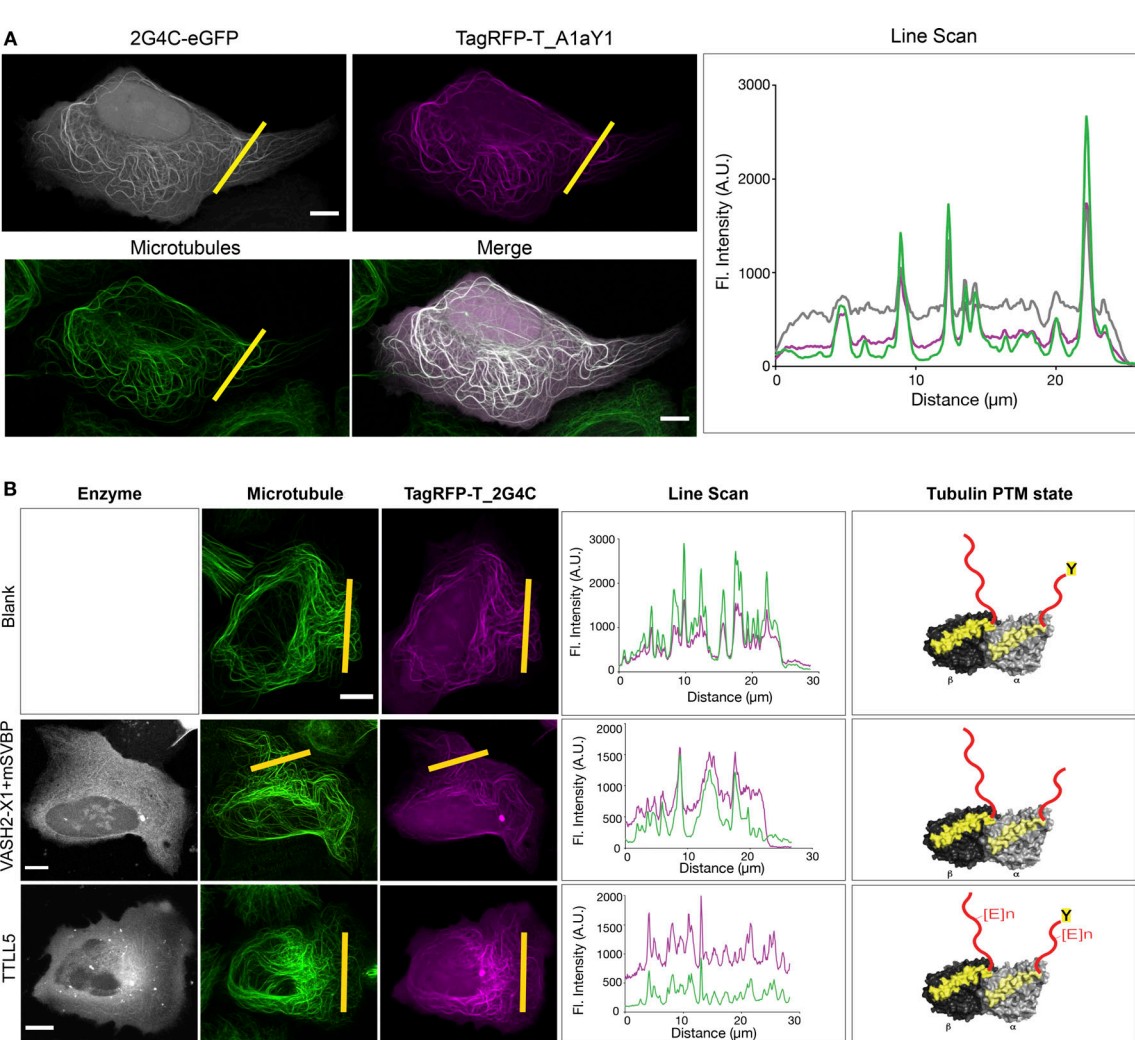

Figure 5. **Specificity experiments with 2G4C. (A)** Z-projection of confocal image stacks of U2OS cells stably expressing A1aY1 binder tagged with TagRFP-T at the N terminus in magenta and 2G4C tagged with EGFP at the C terminus in gray. The cells were additionally stained with SiR-tubulin (green). Line scan for the yellow line shows the intensity profile of 2G4C-EGFP, TagRFP-T_A1aY1, and SiR-tubulin fluorescence. **(B)** Confocal stacks of the cells expressing 2G4C alone, with vasohibin2-X1-GFP-2A-SVBP and TTL5 showing retention of binding of 2G4C. The line scan represents microtubule signal from the region of the cell marked in yellow. Scale bars = 10 μm. A.U., arbitrary units; Fl., fluorescence.

or 1 μM vincristine (Materials and methods). Upon nocodazole addition, we observed that the microtubules begin to shrink from the ends (Fig. 7, A, B, and E; and Video 4), which is similar to mitotic centromere-associated kinesin- or kinesin-13–mediated end depolymerization (Wordeman, 2005). In the case of colchicine, a majority of the microtubules undergo end-on depolymerization events, with frequent severing-like events (Fig. 7, C, E, and F; and Video 5). Similarly, when vincristine was applied to cells, the microtubules became brittle, reminiscent of a severing activity (Fig. 7, D–F; and Video 6). Quantification of the depolymerization events by nocodazole, colchicine, and vincristine show that each has a distinct mechanism of depolymerization, as attributed by structural and biochemical studies (Fig. 7, E and F; Gigant et al., 2005; Lee et al., 1980; Ravelli et al., 2004). While the mode of action for these drugs has been suggested earlier (Jordan and Kamath, 2007), here, we were for the first time able to capture and follow the depolymerization events in real time. Thus, the A1aY1 sensor presents

a great opportunity as a tool to study new microtubule-targeting drugs and understand their mechanism in living cells.

**Live-cell superresolution microscopy with tyrosination sensor**
Cytoskeleton filaments have been favorite test subjects for developing new methodology toward superresolution imaging (Demmerle et al., 2015). Here, we performed 3D structural illumination microscopy (SIM) on cells stably expressing tyrosination red sensor (Fig. 8 A) at interphase stage. Z-stacks were acquired for a total width of 2 μm, and all planes images were reconstructed and 3D volume rendered using α blending (Materials and methods). Correlative analysis of the SiR-tubulin versus TagRFP-T_A1aY1 signal shows a good agreement of colocalization (Pearson's coefficient, 0.75; Spearman's rank correlation, 0.88; Mander's coefficient, 0.93; Fig. 8 B), indicating the abundance of tyrosinated microtubules during interphase cells. Line-scan comparison between 3D-SIM

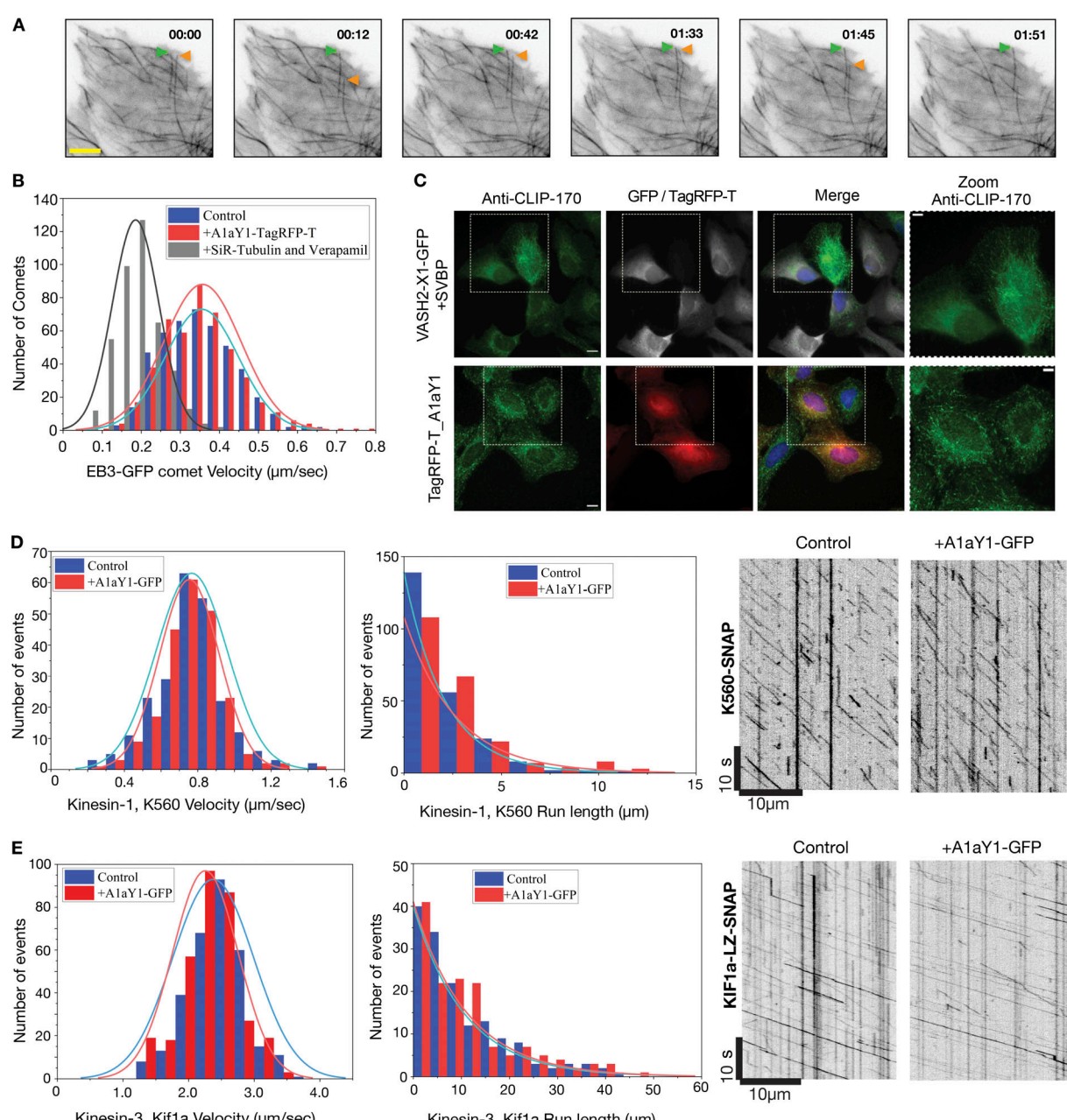

Figure 6. **Effect of tyrosination sensor on microtubule function. (A)** Observation of microtubule dynamics using tyrosination sensor (Tag-RFP-T_A1aY1). The green arrow indicates microtubule pause and the orange arrow shows dynamic instability behavior. Full movie of the frames can be found in Video 1. Scale bar = 5 μm and time in minutes:seconds as indicated. **(B)** Histogram of EB3 comet velocities, representing the microtubule plus-end growth rates (microns/second). The average growth velocities for EB3 comets are 0.36 ± 0.1 μm/s (n = 468), 0.35 ± 0.09 μm/s (n = 450), and 0.19 ± 0.06 μm/s (n = 414) for cells with the tyrosination sensor (red), without binder (blue), and with 0.5 μM SiR-tubulin (gray), respectively. **(C)** CLIP170 staining (green) of U2OS cells in the presence of A1aY1 binder (red) or VASH2-SVBP (gray) with DAPI (blue) as indicated. Scale bars = 10 μm; zoom panel scale bar = 5 μm. **(D and E)** Distribution of K560-SNAP and KIF1A-1-393-LZ-SNAP motor velocities and run lengths with (red) and without (blue) A1aY1 binder. Representative kymograph of K560-SNAP and KIF1a-LZ-SNAP motors with and without A1aY1 binder as indicated. Scale bar x axis = 10 μm and y axis = 10 s. The average velocity of K560-SNAP is 0.75± 0.16 μm, n = 220 (with A1aY1-GFP) and 0.77 ± 0.19 μm, n = 234 (without A1aY1) and KIF1A-1-393-LZ-SNAP are 2.25 ± 0.50 μm, n = 338 (with A1aY1-GFP) and 2.37 ± 0.62 μm, n = 330 (without A1aY1). The average run length of K560-SNAP is 1.8 μm (with A1aY1-GFP) or 1.5 μm (without A1aY1), and the average run length of KIF1A-1-393-LZ-SNAP is 10.1 μm (with A1aY1-GFP) or 9.8 μm (without A1aY1). n represents the number of motor particles analyzed.

versus widefield filament width shows a gain of ~300 nm in resolution (Fig. 8 C). Quantification of microtubules width for ~50 filaments showed a mean full-width half-maximum (FWHM) value of 107 ± 0.35 nm, in the region of half the

diffraction limit (FWHM) as compared with the conventional resolution limit of ~360 nm in a widefield setting (Fig. 8 D). Moreover, we were able to obtain 3D-SIM images for the mitotic stages in U2OS cells, demonstrating the suitability of

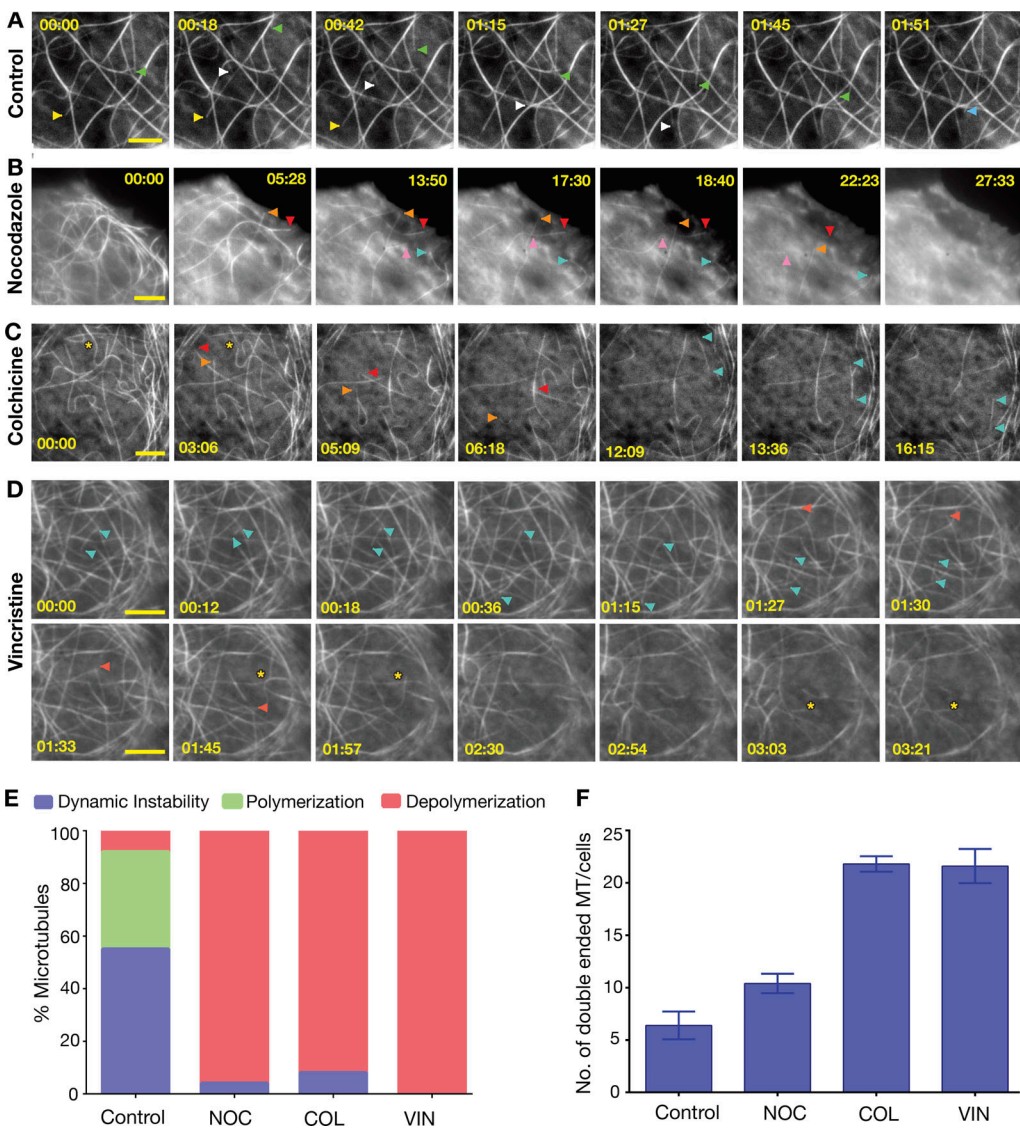

Figure 7. **Drug-induced microtubule depolymerization events in real time. (A)** Control (no drug) experiment of stable U2OS cells with the red tyrosination sensor; microtubule dynamics are indicated with yellow, white, blue, and green arrowheads. **(B–D)** Movie frames of stable U2OS cells with red tyrosination sensor treated with nocodazole, colchicine, and vincristine (two rows). The depolymerization and severing events are indicated with colored arrowheads and asterisks, respectively, for each panel. Full movie of the frames can be found in Videos 3–6 for A–D, respectively. Scale bars = 5 μm, and time in minutes: seconds (as indicated). **(E)** Quantification of microtubule dynamic instability (blue, characterized by a filament undergoing both catastrophe and growth event in the given time frame of the movie), polymerization (green, characterized by a continuous microtubule growth in the given time frame of the movie) and depolymerization (red, characterized by a continuous catastrophe event in the given time frame of the movie) events respectively, from the movies ranging from 5 to 45 min analyzed for control and drug-treated experiments (Materials and methods). Data derived from two to four independent experiments *n* = 100 events for each. **(F)** The number of microtubules where both the ends are visible, a proxy for microtubule severing events observed for control and drug-treated experiments. Data are derived from two to four independent batch of cells treated with drugs and averaged from five different cells; error bars represent SEM. MT, microtubule.

the probe for live-cell superresolution imaging over long time periods (Video 7).

## Discussion

### Advantages of tyrosination sensor

Fluorogenic nanobodies against cellular components, including the actin cytoskeleton, have been successfully employed in unraveling new biology. A notable exception has been the microtubule cytoskeleton (Traenkle and Rothbauer, 2017). In particular,

tubulin PTMs, with their well-characterized epitopes, which have yielded many specific antibodies, have so far no specific nanobodies for live-cell studies.

In this study, we have employed an α-tubulin CTT peptide as epitope and discovered a binder from SSO7D library. The binder was then developed and validated as an intracellular nanobody against the tyrosinated microtubules and thus called a tyrosination sensor. This sensor represents the first robust tubulin nanobody reported in the field that does not affect microtubule and cellular functions. Our imaging experiments with the

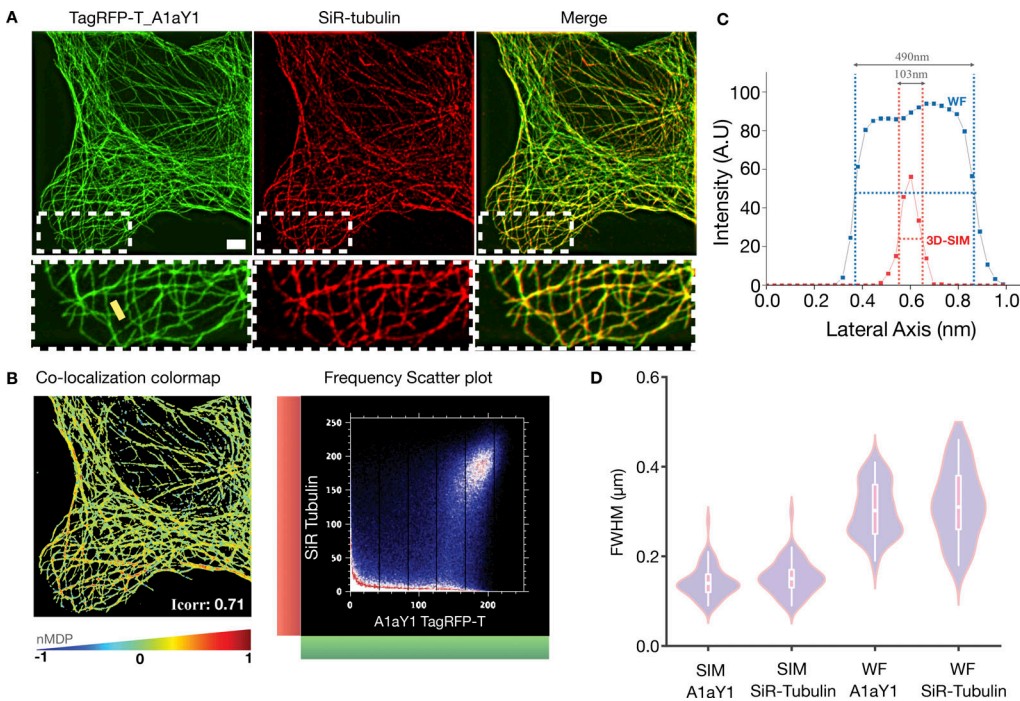

**Figure 8. 3D-SIM imaging of microtubules using tyrosination sensor. (A)** Z-stack of 3D-SIM reconstructed images of N-terminal TagRFP-T tagged to binder A1aY1 (red tyrosination sensor, shown in green here) stably expressed in interphase U2OS cells and labeled with SiR-tubulin (red; scale bar = 2 µm). The boxed areas are magnified below each panel. **(B)** Colocalization colormap contains spatial distribution of calculated normalized mean deviation product (nMDP) values (ranging from −1 to 1). The distribution is based on a color scale in which negative nMDP values are represented by cold colors (segregation); values above 0 are represented by hot colors (colocalization). Representative frequency scatter plot for colocalization and pseudo-colormap of correlations between pairs of corresponding pixels in TagRFP-T_A1aY1 and SiR-tubulin images, thereby offering quantitative visualization of colocalization. **(C)** Line scans of representative single microtubules, from the position shown in the insets with a yellow line, demonstrating the relative resolutions of 3D-SIM and widefield images. **(D)** Quantification of FWHM measurements of *n* = 50 microtubules for 3D-SIM and widefield (WF) images for both SiR-tubulin and red tyrosination sensor (TagRFP-T A1aY1). A.U., arbitrary units.

tyrosination sensor shows that single microtubule events can be followed in real time (Video 1, Video 2, Video 3, Video 4, Video 5, and Video 6). EB3 comet assay and CLIP170 immunostaining further showed that the tyrosination sensor does not interfere with the binding of plus-end tracking proteins. We further extended the imaging capability of our tyrosination sensor toward superresolution microscopy. This is an important advance, as the majority of the superresolution studies with microtubules have used tubulin antibody staining in fixed cells, except in one study, where SiR-tubulin was employed in live-cell SIM imaging (Lukinavičius et al., 2014). Using SIM, here we demonstrate the application of the tyrosination sensor in superresolution microscopy that yields resolution similar to the SiR-tubulin probe (Fig. 8). In contrast to SiR-tubulin, however, our tyrosination sensor does not affect microtubule dynamics (Fig. 6 B). Since most of the microtubules in the interphase stage of epithelial and fibroblast cells are tyrosinated, we envision that our tyrosination sensor can be applied as a generic microtubule marker.

### A1aY1 recognizes majority of the PTM sites within α-tubulin CTT

While the terminal tyrosine residue is an indispensable element in recognition by the sensor, the glutamate residues of α-tubulin CTTs also contribute in the binding to the sensor. A key element in this interaction is the third alternating glutamic acid residue

(n-sixth residue) of the α-tubulin CTT, the most common site for glutamylation modification in brain tubulin (Eddé et al., 1990). Indeed, we demonstrated that glutamylation of α-tubulin strongly reduces sensor binding, suggesting a steric hindrance by the branched glutamate.

α-Tubulin CTTs bearing terminal tyrosine are known to be specifically recognized by CAP-Gly domain–containing proteins (Honnappa et al., 2006; Mishima et al., 2007; Steinmetz and Akhmanova, 2008), vasohibins (Liao et al., 2019; Zhou et al., 2019), and kinesin-13 and kinesin-2 motors (Sirajuddin et al., 2014). Additionally, the CAP-Gly domains bind to the C-terminal EEY motif of end-binding proteins, which is the consensus to the α-tubulin C terminus (Honnappa et al., 2006). Structural studies show that the sextet acidic motif (EEGEEY/F) of α-tubulin CTTs, together with the EEY motif of end-binding proteins, is important for the docking of the CAP-Gly domain (Honnappa et al., 2006; Mishima et al., 2007). Vasohibin bound to α-tubulin CTTs also shows that the last five residues of α-tubulin CTT (-EGEEY) bind to the active site of the enzyme (Liao et al., 2019). In both cases (tyrosine recognition by CAP-Gly and VASH proteins), the free main-chain carboxyl group of the C-terminal tyrosine residue is essential for the molecular interactions. In contrast, our A1aY1–α-tubulin CTT complex structure reveals a novel and unique mode of tyrosine sensing that involves the interaction of phenyl ring and the hydroxyl group of the tyrosine

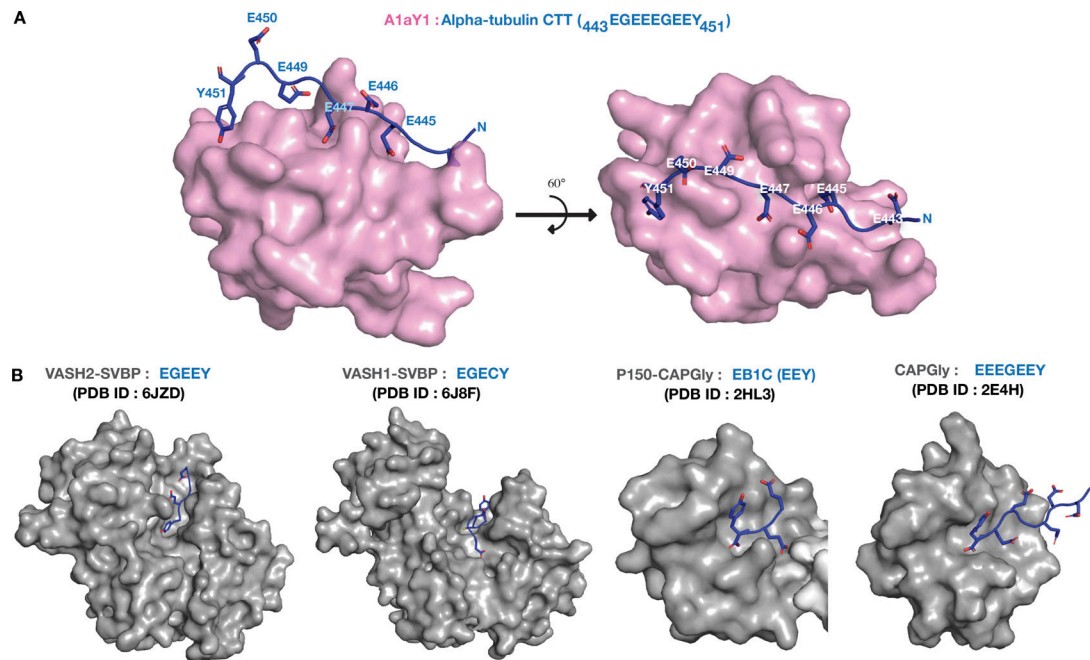

Figure 9. **Structural comparison of α-tubulin CTT-binding proteins. (A)** NMR structure of A1aY1 binder (magenta) in complex with α tubulin CTT (blue), up to seven residues of CTT interact with A1aY1 binder. **(B)** Structures of VASH2-SVBP:EGEEY (PDB: 6JZD), VASH1-SVBP:EGECY (PDB: 6J8F), P150-CAP-Gly:EB1C (PDB: 2HL3), and CAP-Gly:EEEGEEY (PDB: 2E4H), with α-tubulin CTT in blue and proteins in gray.

side chain, but not the main-chain carboxyl group (Figs. 1 C and 4 D). This difference in molecular recognition of terminal tyrosine by A1aY1 explains why our tyrosination sensor does not interfere with the binding of other tyrosine-sensing proteins to microtubules. Another unique feature of the A1aY1 binder is that it extends recognition toward the glutamylation site (i.e., n-sixth residue), which has so far never been observed for naturally occurring proteins that bind to the α-tubulin CTT (Fig. 9). Therefore, we conclude that our tyrosination sensor senses the glutamylation state of α-tubulin, in addition to the tyrosination state of microtubules.

### Application in studying drugs targeting microtubules and tubulin PTMs

Microtubules are a well-known target for anticancer therapeutics, and several reports describe drugs that stabilize and destabilize microtubules (Jordan and Kamath, 2007). Most of the drugs that are known have a detailed account of their activity and binding site, from which the mechanism of action has been proposed (Steinmetz and Prota, 2018). However, live-cell imaging studies of drug-induced microtubule depolymerization events are scarce. Here, we applied our tyrosination sensor to image nocodazole-, colchicine-, and vincristine-induced microtubule depolymerization events in real time (Fig. 7). Nocodazole is known to bind free tubulin dimers, thus preventing their addition to microtubule polymer (Lee et al., 1980). Our results show that upon nocodazole treatment, the microtubules are in a constant state of catastrophe without any rescue events, in line with the proposed mode of nocodazole action. Colchicine, a widely used microtubule-depolymerizing drug, was originally used to identify tubulin (Borisy and Taylor, 1967). Biochemical

and structural investigations suggested that colchicine can bind to both soluble tubulin as well as microtubule lattice (Jordan and Kamath, 2007; Ravelli et al., 2004). Our finding reveals that colchicine-treated cells show a combination of severed polymers and end depolymerization events, which underscores dual binding mode and twofold action of colchicine. Vincristine is a potent anticancer drug that binds only to microtubule polymer and destabilizes the lattice (Dhamodharan et al., 1995; Gigant et al., 2005; Jordan and Kamath, 2007; Jordan et al., 1992). In line with previous findings, with vincristine, we observed more polymer severing and rapid depolymerization of microtubules with free minus ends.

While the three drugs studied here show distinct modes of microtubule depolymerization, uniquely, they also have commonalities among them. For example, when nocodazole and colchicine bind to free tubulin, they trap tubulin in a curved state, which is incompatible with the microtubule lattice (Brouhard and Rice, 2014). Similarly, when colchicine or vincristine binds to the microtubule, it induces lattice defects by kinking the longitudinal interactions between tubulins (Gigant et al., 2005; Ravelli et al., 2004). These lattice defects are then amplified, leading to breaking of polymer, akin to severing-like activity, which we have observed here for the first time in cells.

### Conclusions

In summary, the tyrosination sensor described here can be applied to study microtubule behavior and mechanisms pertaining to microtubule-destabilizing drugs. Also, since our tyrosination sensor marks unmodified microtubules without affecting dynamics, we predict that our sensor will be a valuable tool in screening new drugs that target microtubules. Our specificity

experiments described in Fig. 4 outset potential application toward studying tubulin PTMs and discovering drug specific to their enzymes, such as vasohibin and TTLLs. Finally, the tyrosination sensor described here presents a unique opportunity to study microtubules and tubulin PTMs in living cells using fluorescence and superresolution microscopy methods. It further provides the prospect to expand this methodology to generate sensors against other tubulin PTMs such as detyrosination, acetylation, glutamylation, and glycylation, thus opening the exciting possibility to decode the still-enigmatic tubulin code in living cells and organisms.

## Materials and methods

### Peptides used in screening binders from the library
The peptide sequences mimic the tubulin PTMs and are derived from the amino acid sequence of CTT of α-tubulin (tyrosination: biotin-[[440]VEGEGEEEGEEY[451]], referred to as Hs_TUBA1A, resembling biotinylated CTT of TUBA1A 440–451; detyrosination: biotin-[[440]VEGEGEEEGEE[450]], referred to as Hs_TUBA1A-ΔY, resembling biotinylated CTT of TUBA1A 440–450; monoglutamylation: biotin-[[440]VEGEG[E]EEGEEY[451]], referred to as Hs_TUBA1A-mG, resembling biotinylated CTT of TUBA1A 440–451-[E]445). Hs_TUBA1A, Hs_TUBA1A-ΔY, and Hs_TUBA1A-mG peptides were synthesized from Thermo Fisher Scientific and used in screening and surface plasmon resonance (SPR) SA sensor chip immobilization. Drosophila and C. elegans α-tubulin CTT peptides, biotin-Tub84B 438–450 (biotin-Dm_Tub84B; biotin-[438]SGDGEGEGAEEY[450]) and biotin-Tba-1 437–449 (biotin-Ce_TBA1; biotin-[437]SNEGGNEEEGEEY[449]), were synthesized from LifeTein and were used in SPR SA sensor chip immobilization. Tyrosinated and detyrosinated peptides used in NMR titration with A1aY1 protein were synthesized from LifeTein without any biotinylation.

### Yeast surface display library screening
A combinatorial SSO7d yeast display library was obtained as a kind gift from Dr. Balaji M. Rao's laboratory, and the detailed protocol for screening binders was adopted as described earlier (Gera et al., 2011). To screen a binder for CTT of human α-tubulin specific for terminal tyrosine residue, SSO7d library was applied for a stringent negative selection with biotinylated detyrosinated (biotin-Hs_TUBA1A-ΔY) and monoglutamylated (biotin-Hs_TUBA1A-mG) peptides to reduce the diversity and nonspecific binders from the library, followed by a positive selection with tyrosinated peptide (biotin-Hs_TUBA1A), which yielded a population of binders with varying degrees of binding affinities.

The library (diversity ~$10^8$ cells) was propagated in 10-fold excess of its diversity ($10^9$ cells) in fresh glucose-containing SDCAA media (20 g/liter D-(+)-glucose [Sigma-Aldrich; catalog no. G5767], 6.7 g/liter yeast nitrogen base without amino acids [BD Difco; catalog no. 291940], 5 g/liter casamino acids [BD Difco; catalog no. 90001–726], 5.4 g/liter $Na_2HPO_4$ [Thermo Fisher Scientific; catalog no. S374], 8.6 g/liter $NaH_2PO_4.H_2O$ [Merck; catalog no. 106346], and 1× penicillin-streptomycin [PenStrep; Gibco]) thrice at 30°C, 250 rpm for 18–24 h before

screening. The freshly grown yeast cells were induced ($10^9$ cells) in galactose-containing SGCAA media (20 g/liter D-(+)-galactose [Sigma-Aldrich; catalog no. G0750], 6.7 g/liter yeast nitrogen base, 5 g/liter casamino acids, 5.4 g/liter $Na_2HPO_4$, and 8.6 g/liter $NaH_2PO_4.H_2O$) at 20°C for 20–24 h for the expression of binders on the surface of yeast cells. The freshly induced culture was used for magnetic screening (negative and positive selection with the target peptides) and further sorted using FACS to obtain the yeast cells with higher binding affinity for Hs_TUBA1A.

### Magnetic screening
200 µl ($5 × 10^6$ beads) magnetic beads (Invitrogen; Dynabeads Biotin Binder, catalog no. 11047) prewashed with PBS-BSA (8 g/liter NaCl, 0.2 g/liter KCl, 1.44 g/liter $Na_2HPO_4$, 0.24 g/liter $KH_2PO_4$, and 1 g/liter BSA) was incubated with 1 µM of each biotin-tagged peptide (biotin-TUBA1A 440–451, biotin-TUBA1A 440–450, and biotin-TUBA1A 440–451-[E]-445) in separate centrifuge tubes in PBS-BSA at 4°C overnight on a rotatory rod. Approximately $10^9$ ($OD_{600}$ = 1 is ~$10^7$ cells) freshly induced cells were pelleted and incubated with 200 µl beads coated with biotin-TUBA1A 440–450 and biotin-TUBA1A 440–451-[E]-445 peptide (for negative selection against detyrosinated and monoglutamylated peptides) for 1 h at 4°C on a rotatory rod. Yeast cells bound to the beads were separated using a magnetic stand and discarded. The rest of the unbound cells were incubated for positive selection with 200 µl beads coated with biotin-Hs_TUBA1A (tyrosinated peptide biotin-TUBA1A 440–451) at 4°C on a rotatory rod for 1 h. The cells bound to the beads were pulled with the magnetic stand and washed five times with PBS-BSA at room temperature for 5–10 min. The washed beads were transferred to a fresh 5 ml SDCAA media for growth at 30°C, 250 rpm for 48 h. Beads were then removed from the grown culture, and cells were grown in large culture volume for stocks preparation ($10^9$ cells) and use for FACS.

### FACS
The freshly grown cells from positive magnetic screening were induced in galactose-containing SGCAA media ($10^9$ cells) at 20°C for 24 h for surface expression of binder proteins on yeast cells. Approximately $10^7$ cells ($OD_{600}$ = 1) from the culture were pelleted in the microcentrifuge tube. The cells were resuspended in 100 µl PBS-BSA and incubated with biotin-TUBA1A 440–451 peptide (100 µM) and chicken anti-c-Myc antibody (Invitrogen; catalog no. A21281) in 1:250 dilution at room temperature for 1 h. Cells were pelleted, washed twice with PBS-BSA, and incubated with goat anti-chicken IgG Alexa Fluor 633 (Invitrogen; catalog no. A21052) and neutravidin fluorescein conjugate (Invitrogen; FITC, catalog no. A2662, used for first FACS) or Streptavidin R phycoerythrin (PE) conjugate (Invitrogen; Strep-PE, catalog no. S866, used for second, third, and fourth FACS) secondary reagent in 1:250 dilutions for 15 min on ice. Cells were then washed to remove excess secondary reagents and sorted on BD FACS Aria Fusion for double-positive cells, keeping unstained and single stained controls (Central Imaging and Flow Cytometry facility at the National Center for Biological Sciences [NCBS]). 0.1–1% double-positive cells (for FITC/PE and Alexa Fluor 633 fluorophores) were collected to the count of ~4,000–10,000

cells and grown in 5 ml fresh SDCAA media. The freshly grown cells were propagated in larger volumes of SDCAA media (250–500 ml) to make stocks and further rounds of sorting experiments. After four rounds of sorting, the cells were plated on an agar plate (20 g/liter dextrose, 6.7 g/liter yeast nitrogen base, 5 g/liter casamino acids, 5.4 g/liter $Na_2HPO_4$, 8.6 g/liter $NaH_2PO_4.H_2O$, 182 g/liter sorbitol, and 15 g/liter agar), and 10 single yeast colonies were analyzed to identify the most abundant sequence/clone enriched in post-sorted culture.

### Protein purification

The gene sequences of A1aY1 and A1aY2 binders were cloned in pET28a(+) vector between 5′-NdeI and 3′-NotI restriction sites using forward (5′-agcggcctggtgccgcgcggcagccatatggcgaccgtgaaatttaaatataaaggcg-3′) and reverse (5′-cagtggtggtggtggtggtgctcgagtgcggccgcttattttttctgttttttccagcatctgcagcag-3′) primers/oligos in two-fragment Gibson cloning. The constructs were designed to have an N-terminal 6xHis-tag followed by a thrombin site, the gene sequence 5′-atgggcagcagccatcatcatcatcatcacagcagcggcctggtgccgcgcggcagccatATGGCGACCGTGAAATTTAAATATAAAGGCGAAGAAAAACAGGTGGATATTAGCAAAATTCTATCAGTGGGTCGCTACGGCAAATTAATTCATTTTCTCTATGATCTGGGCGGCGGCAAAGCGGGCATGGGCATGGTGAGCGAAAAAGATGCGCCGAAAGAACTGCTGCAGATGCTGGAAAAACAGAAAAAAtaa-3′, and the protein sequence MGSSHHHHHHSSGLVPRGSH*MATVKFKYKGEEKQVDISKILSVGRYGKLIHFLYDLGGGKAGMGMVSEKDAPKELLQMLEKQKK** (the A1aY1 gene sequence is marked in uppercase; italicized amino acids refer to the protein sequence of the A1aY1 binder).

Protein expression was achieved in Rosetta (DE3)–competent cells by inducing the culture at $OD_{600}$ of 0.5 with 1 mM IPTG (Sigma-Aldrich; catalog no. I6758) at 25°C for 8–10 h in terrific broth (yeast extract 24 g/liter, tryptone 20 g/liter, 17 mM $KH_2PO_4$, 72 mM $K_2HPO_4$, and 4 ml/liter glycerol) and 50 µg/ml kanamycin. Only A1aY1 could be purified successfully, as A1aY2 protein becomes toxic to the bacterial expression host (DE3 Rosetta) and hence was purified with a C-terminal GFP tag (mentioned below). Overnight-induced culture of A1aY1 was harvested in 50 mM Tris-HCl, pH 7.5, 100 mM NaCl, and 1 mM PMSF with one tablet of EDTA-free protease inhibitor cocktail (Roche; catalog no. 11836170001) for 1 liter of the culture. The cells were lysed with the Avestin Emulsiflex C3 homogenizer (ATA Scientific Instruments), and protein was purified using 5 ml Ni-NTA affinity column after a 10–column volume wash with 50 mM Tris-HCl, pH 7.5, 500 mM NaCl, and 25 mM imidazole, pH 7.5, and elution in 5 column volumes of elution buffer (50 mM Tris-HCl, pH 7.5, 100 mM NaCl, and 350 mM imidazole, pH 7.5). The eluted protein was concentrated up to 5 ml using a 3-kD Millipore Amicon filter (Merck; UFC900324). Further, size exclusion chromatography was performed in 50 mM Tris-HCl, pH 7.5, and 100 mM NaCl buffer in a Superdex-75, 16/600 column (GE; catalog no. 28989333). The resulting fraction of the pure A1aY1 protein was concentrated (mol wt 9.3 kD) and frozen in small aliquots for long-term storage at –80°C.

For NMR experiments, $^{13}C$-$^{15}N$ isotope–labeled A1aY1 protein was purified from the DE3 Rosetta bacterial cells grown in M9 media ($Na_2HPO_4$ 6 g/liter, $KH_2PO_4$ 3 g/liter, NaCl 0.5 g/liter, 15

$NH_4Cl$ 1 g/liter, 13C-labeled glucose 2 g/liter, divalent cations, vitamin B12, thiamine and trace elements, and 50 µg/ml kanamycin antibiotic) using Ni-NTA affinity chromatography in Tris buffer as mentioned above. Further purification using size exclusion chromatography was performed in 50 mM sodium phosphate buffer, pH 6.5, with 200 mM NaCl. Purified protein was set for thrombin cleavage (thrombin from bovine plasma; Sigma-Aldrich; T-4648-10KU [15–20 U thrombin per milligram of the protein]) at room temperature for 4 h. The cleaved protein was purified again with size exclusion chromatography (using S75 16/600 column) in 50 mM sodium phosphate buffer, pH 6.5, with 200 mM NaCl. The resulting protein corresponds to 7.8 kD, which was concentrated and frozen in –80°C in small aliquots for future use in NMR experiments.

A pET28a(+) vector with a GFP sequence was used to clone A1aY1 and A1aY2 gene sequences using forward (5′-gcaaatgggtcgcggatccatggcgaccgtgaaatttaaatataaaggcgaag-3′) and reverse (5′-tctcctttactcatggtacctttttctgttttttccagcatctgcagcag-3′) primers/oligos between 5′-BamHI and 3′-KpnI restriction sites using two-fragment Gibson cloning. The final constructs contain an N-terminal 6x-histine tag, a thrombin site, T7 leader sequence, and A1aY1/A1aY2 gene sequence followed by a C-terminal GFP sequence separated with a glycine-serine linker from the binder: 5′-atgggcagcagccatcatcatcatcatcacagcagcggcctggtgccgcgcggcagccatatggctagcatgactggtggacagcaaatgggtcgcggatccATGGCGACCGTGAAATTTAAATATAAAGGCGAAGAAAAACAGGTGGATATTAGCAAAATTCTATCAGTGGGTCGCTACGGCAAATTAATTCATTTTCTCTATGATCTGGGCGGCGGCAAAGCGGGCATGGGCATGGTGAGCGAAAAAGATGCGCCGAAAGAACTGCTGCAGATGCTGGAAAAACAGAAAAAAggtacc*atgagtaaaggagaagaacttttcactggagttgtcccaattcttgttgaattagatggtgatgttaatgggcacaaattttctgtcagtggagagggtgaaggtgatgcaacatacggaaaaacttaccettaaatttatttgcactactggaaaactacctgttccatggccaacacttgtcactactctgacttatggtgttcaatgcttttcaagatacccagatcatatgaaacagcatgacttttttcaagagtgccatgcccgaaggttatgtacaggaaagaactatattttttcaaagatgacgggaactacaagacacgtgctgaagtcaagtttgaaggtgatacccttgttaatagaatcgagttaaaaggtattgattttaaagaagatggaaacattcttggacacaaattggaatacaactataactcacacaatgtatacatcatggcagacaaacaaaagaatggaatcaaagttaacttcaaaattagacacaacattgaagatggaagcgttcaactagcagaccattatcaacaaaatactccaattggcgatggccctgtccttttaccagacaaccattacctgtccacacaatctgccctttcgaaagatcccaacgaaaagagagacacatggtccttcttgagtttgtaacagctgctgggattacacatggcatggatgaactatacaaataa*-3′ (the A1aY1 gene sequence is marked in uppercase letters, and the GFP gene sequence is marked in italicized lowercase letters).

The binder A1aY2, cloned with a C-terminal GFP tag, was purified from the soluble fraction. Both the proteins were purified in 50 mM potassium phosphate buffer, pH 6.0, with 100 mM potassium chloride and 5 mM β-mercaptoethanol (BME) using Ni-NTA chromatography. Further purification of the proteins was performed using S-200, 16/600 column for size exclusion chromatography and were used in SPR experiments for testing the binder by diluting them in 1×HBS-P+ buffer (10 mM Hepes, 150 mM NaCl, and 0.05% vol/vol surfactant P20; GE; catalog no. BR100671; Fig. S1 C). For all the in vitro experiments using total internal reflection fluorescence (TIRF) microscopy, A1aY1-GFP was purified in 50 mM Pipes, pH 6.8, 100 mM KCl, and 5 mM β-mercaptoethanol. Further purification

was achieved using S-75, 16/600 column for size exclusion chromatography in 1×BRB80 (80 mM Pipes, 1 mM MgCl₂, and 1 mM EGTA, pH 6.8) buffer. This purified A1aY1-GFP was used in later experiments with the polymerized HeLa microtubules and staining microtubules in fixed cells.

### K560-SNAP and KIF1A-SNAP purification

Truncated rat kif1a (1–393 amino acids) followed by a GCN4 leucine zipper was cloned into a pET-17b vector with a SNAP-tag followed by a 10× histidine-tag at the C terminus. K560-Snap and Kif1a-LZ-Snap were expressed using the Rosetta (DE3) bacterial expression system. Transformed cells were grown at 37°C to OD$_{0.4–0.6}$ followed by induction with 0.5 mM IPTG (Sigma-Aldrich) with overnight shaking at 24°C. Cells were harvested and lysed in buffer A (25 mM Pipes, pH 6.8, 100 mM KCl, 5 mM MgCl₂, 5 mM β-mercaptoethanol, and 30 mM imidazole). The supernatant was loaded onto a Ni-NTA column, followed by a high-salt wash with buffer B (buffer A with 300 mM KCl and 200 µM ATP) followed by a high imidazole wash with buffer C (buffer A with 50 mM imidazole) and elution with buffer E (buffer A with 350 mM imidazole). Pure proteins were obtained by further subjecting the Ni-NTA elute to gel filtration using a S200 16/1600 column (GE) in 1×BRB80. SNAP surface Alexa Fluor 647 (New England Biolabs; catalog no. S9136S) labeling of SNAP-tag proteins were performed as per the manufacturer's protocol on the New England Biolabs website.

### Cell culture experiments

Wild-type U2OS and HEK293-T cells used in this study were obtained from Prof. Satyajit Mayor's laboratory (NCBS, Bangalore, India) as a gift. Heart ventricular origin H9C2 cardiomyocytes were gifted from Dr. Dhandapany Perundurai's laboratory (in-Stem, Bangalore, India). U2OS cells were grown in a humidified 37°C incubator with 5% CO₂ in McCoy's 5A (Sigma-Aldrich; M4892) media supplemented with 2.2 g/liter sodium bicarbonate (Sigma-Aldrich; catalog no. S5761), 10% FBS, and 1× PenStrep (Gibco, Thermo Fisher Scientific; catalog no. 15–140-122). HEK293-T and H9C2 cells were grown in DMEM media (Gibco) supplemented with 1× sodium pyruvate (1 mM; Gibco), 1× GlutaMAX (Gibco), and 1× PenStrep.

### Primers used to clone A1aY1 in mammalian expression vectors

A codon-optimized gene sequence of A1aY1 followed by (Gly4Ser)3 linker was synthesized from Genscript in pUC57 vector with the gene sequence 5′-ATGGCAACAGTCAAGTTCAAATACAAGGGG GAGGAAAAGCAGGTGGACATTAGTAAGATTCTGAGCGTCGGA AGATACGGGAAGCTGATCCACTTCCTGTACGACCTGGGAGGA GGCAAGGCAGGAATGGGCATGGTGAGCGAGAAGGATGCCCCC AAGGAGCTGCTCCAGATGCTGGAGAAGCAGAAGAAGggggggagga gggtcaggaggggggaggctccggaggtggcgggtct-3′ and the protein sequence *MATVKFKYKGEEKQVDISKILSVGRYGKLIHFLYDLGGGKAGMG MVSEKDAPKELLQMLEKQKKGGGGSGGGGSGGGGS* (the codon-optimized A1aY1 gene sequence is marked in uppercase, and the (Gly4Ser)3 linker DNA sequence is marked in lowercase letters; italicized amino acids refer to the protein sequence of A1aY1 binder).

All transient transfections with A1aY1 fused to different fluorophores were performed using pIRESneo mammalian expression vector with a cytomegalovirus (CMV) promoter (Addgene; plasmid #12298; Rusan et al., 2001) that was repurposed to clone A1aY1 fused with different fluorophores using 5′-NheI and 3′-BamHI or NotI restriction sites. The following fluorophores and primers were used to clone the A1aY1 binder in the pIRESneo vector.

A1aY1 cloned with a C-terminal EGFP or mCherry or mCitrine in pIRESneo CMV vector: 5′-gatcgatatctgcggcctagctagcgctaccgg tcgccaccatggcaacagtcaagttcaaatacaaggggg-3′ forward primer and 5′-tcctcgcccttgctcaccatagacccgccacctccgga-3′ reverse primer to amplify codon-optimized A1aY1 gene from pUC57 vector. To amplify EGFP/mCherry/mCitrine, one forward primer (5′-agg gtcaggagggggaggctccggaggtggcgggtctgtgagcaagggcgagga-3′) and two reverse primers (5′-tcaggcgctccaggggggggcagctgcagaccagagg atcccttgtacagctcgtccatgccg-3′ as reverse primer 1 and 5′-agcaca ctggatcagttatctatgcggccgcgttagtccagggtcaggcgctccaggggggggcag-3′ as reverse primer 2) were used to amplify EGFP/mCherry/mCitrine followed with a nucleus export signal (GSSGLQLPPLERLTLD) sequence in a two-step PCR reaction. Both the amplicons of A1aY1 and EGFP/mCherry/mCitrine-nucleus export signal were cloned in between NheI and NotI restriction sites of pIRESneo vector using three-fragment Gibson cloning method.

A1aY1 cloned with a C-terminal TagBFP in pISESneo CMV vector: The above vectors were digested with 5′-BspEI and 3′-BamHI restriction site to remove EGFP/mCherry/mCitrine fluorophores. To amplify TagBFP, a 5′-agggtcaggagggggaggctccgg aggtggcgggtctagcgagctgattaaggag-3′ forward primer and 5′-cca ggggggggcagctgcagaccagaggatccattaagcttgtgccccag-3′ reverse primer were used and cloned in pIRESneo CMV vector in place of EGFP/mCherry/mCitrine using the two-fragment Gibson cloning method.

A1aY1 cloned with a C-terminal TagRFP-T: The above A1aY1 with EGFP/mCherry/mCitrine/TagBFP construct was digested with 5′-BspEI and 3′-BamHI restriction site to remove the above fluorophore sequence. The vector was further cloned using the two-fragment Gibson cloning method with a C-terminal TagRFP-T using forward 5′-gggaggagggtcaggagggggaggctccgg agtgtctaagggcgaagag-3′ and reverse 5′-ccaggggggggcagctgcagac cagaggatccttacttgtacagctcgtc-3′ primers.

A1aY1 cloned with an N-terminal TagBFP in pIRESneo CMV vector: 5′-gatcgatatctgcggcctagctagcgctaccggtcgccaccatgagcgagc tgattaag-3′ forward primer and 5′-tcgcccccactcgagatctgagtccgg aattaagcttgtgccccag-3′ reverse primer were used to amplify TagBFP, and 5′-aaactggggcacaagcttaattccggactcagatctcgagtggca acagtcaagttcaaatac-3′ forward primer and 5′-cagttatctatgcggccg cggatccttacttcttctgcttctcc-3′ reverse primer were used to amplify the A1aY1 gene to clone them in pIRES vector using three-fragment Gibson cloning between NheI and BamHI restriction sites.

A1aY1 cloned with an N-terminal TagRFP-T and N-terminal 6×-histidine–tagged TagRFP-T in pIRESneo CMV vector: The above N-terminal TagBFP_A1aY1 construct was digested with 5′-NheI and 3′-BspEI restriction site to remove TagBFP gene sequence and clone TagRFP-T gene at this site using two-fragment Gibson cloning. The TagRFP-T gene sequence was amplified using 5′-gatcgatatctgcggcctagctagcgctaccggtcgccaccATGGTGTCT AAGGGCGAAG-3′ forward primer and 5′-ttgccactcgagatctgagtc

cggaCTTGTACAGCTCGTCCATG-3′ reverse primer. Further, to clone N-terminal 6×-histidine TagRFP-T, the N-terminal TagRFP-T_A1aY1 construct was digested with 5′-NheI and 3′-BspEI restriction site to replace TagRFP-T gene with 6×-histidine–tagged TagRFP-T gene using forward primer 5′-gat cgatatctgcggcctagctagcgctaccggtcgccaccatgggcagctcccatcatcatc atcatcacagctccggcgtgtctaagggcgaagag-3′ and reverse primer 5′-ttgccactcgagatctgagtccggacttgtacagctcgtccatg-3′ to amplify the gene and clone by the two-fragment Gibson assembly cloning method.

For stable cell line generation, the TagBFP and TagRFP-T constructs of A1aY1 were cloned in a pTRIP chicken β-actin (CAG) vector (Gentili et al., 2015) with a CMV enhancer and CAG promoter lentiviral vector. The genes were inserted between 5′-NheI and 3′-BamHI restriction sites. The following primers were used to clone TagBFP and TagRFP-T constructs.

A1aY1 with a C-terminal TagBFP in pTRIP-CAG vector: The A1aY1-TagBFP gene was amplified from the pIRESneo vector using 5′-tttggcaaagaattattccgctagcgccaccatggcaacagtc-3′ forward primer and 5′-tctcgaggtcgacactagtggatccttaattaagcttgtgccccag-3′ reverse primer and cloned in pTRIP vector between 5′-NheI and 3′-BamHI restriction sites using the two-fragment Gibson cloning method.

A1aY1 with an N-terminal TagBFP in pTRIP-CAG vector: The TagBFP_A1aY1 gene was amplified from the pIRESneo vector using 5′-tttggcaaagaattattccgctagcgccaccatgagcgagctg-3′ forward primer and 5′-tctcgaggtcgacactagtggatccttacttcttctgcttctcca gcatc-3′ reverse primer and cloned in pTRIP vector between 5′-NheI and 3′-BamHI restriction sites using the two-fragment Gibson cloning method.

A1aY1 with an N-terminal TagRFP-T in pTRIP-CAG vector: The TagRFP-T_A1aY1 gene was amplified from the pIRESneo vector using 5′-tttggcaaagaattattccgctagcgccaccatggtgtctaaggg cgaag-3′ forward primer and 5′-tctcgaggtcgacactagtggatccttact tcttctgcttctccagcatc-3′ reverse primer and cloned in pTRIP vector between 5′-NheI and 3′-BamHI restriction sites using the two-fragment Gibson cloning method.

## Stable cell line expressing tyrosination sensor
HEK293-T cells were cultured in complete DMEM media with 10% FBS, 1× GlutaMAX (Gibco), 1× sodium pyruvate and 1× PenStrep in a humidified 37°C incubator with 5% $CO_2$. Freshly passaged HEK293-T cells were grown up to 70–80% confluency. The lentiviral vectors (pTRIP vector) cloned with TagRFP-T_A1aY1 or TagBFP_A1aY1, under CMV enhancer and chicken β-actin promoter (CAG promoter) flanked with 5′and 3′ long terminal repeat sequences (Gentili et al., 2015), were used for transfection of HEK293-T cells. For a 100-mm dish transfection, 5 µg lentiviral plasmid cloned with the gene of red or blue tyrosination sensor, 3.75 µg psPAX2 (Addgene; #12260), and 1.25 µg pmDG2 (Addgene; #12259) plasmid were mixed together in 500 µl OptiMEM media with 20 µl P3000 (Invitrogen; Lipofectamine-3000 transfection reagent, catalog no. L300015) or 10 µl PLUS reagent (Invitrogen; LTX transfection reagent, catalog no. L15338100). In a separate microcentrifuge vial, 500 µl OptiMEM was taken, and 30 µl Lipofectamine-3000 or Lipofectamine-LTX reagent was added to it. This Lipofectamine-

containing solution was added to the plasmid and incubated at room temperature for 20 min. Cell media was changed to no-PenStrep–containing DMEM media, and the transfection mix was added drop by drop. Cells were incubated for 15–18 h, and then the media was changed to PenStrep-containing media. Following that lentivirus containing supernatant media was collected at 48 h, 72 h, and 96 h post-transfection with the replacement of 10 ml fresh media every time. The virus supernatant was pooled together and concentrated in a 50-kD Millipore Amicon filter (Merck; UFC905024) at 1,000$g$ to ~1–3 ml. The concentrated supernatant was supplemented with one-third volume of Lenti-X concentrator (Takara; catalog no. 631231) and incubated at 4°C overnight. The viruses were pelleted at 1,500$g$ for 45 min at 4°C. The white pellet of lentivirus was resuspended in 1–2 ml DMEM or McCoy's complete media (10% FBS) and stored for long-term use at –80°C in 300–500-µl aliquots.

The lentiviral transduction was performed in the 60% confluent culture of wild-type U2OS in 5 ml complete McCoy's media (with 10% FBS, 1× PenStrep, and 2.2 g/liter NaHCO$_3$) with 1–2 µg/ml polybrene (Merck; catalog no. TR-1003-G) and lentiviruses (0.5–1 ml thawed at 37°C). After 24 h of transduction, media was changed with the fresh media, and cells were propagated as normal cell line stably expressing the red or the blue tyrosination sensor.

## Mitotic arrest experiment (Fig. S5 C)
Stable U2OS cells expressing A1aY1 with an N-terminal TagRFP-T (red tyrosination sensor) were grown up to 50% confluency in McCoy's 5A media (Sigma-Aldrich; M4892) in 37°C in a 5% $CO_2$ incubator. Cells were arrested at S phase of the cell cycle with 2.5 mM thymidine for 16–20 h at 37°C and released for cell cycle progression for 8–9 h in fresh media. Similarly, a second S-phase arrest in 2.5 mM thymidine was performed for 20–24 h and released for 8 h in fresh media, followed by 20 ng/ml nocodazole (Sigma-Aldrich; M1404) treatment for 4 h. The synchronized cells (30–40%) were imaged for mitosis after staining the nucleus of the cells with DAPI (1 µg/ml) for 15 min or with NucBlue live ready probes reagent (Invitrogen; catalog no. R37605).

## Determining the intracellular expression level of the sensor (Fig. S3, C–I)
Wild-type U2OS cells were transfected with N-terminal 6x-His TagRFP-T_A1aY-1 construct. Transfected cells were resuspended in 2.5% FBS containing 1× PBS solution. Cells were sorted on BD FACS Aria III into two populations (low-medium and high) based on the level of expression of TagRFP-T. Approximately 0.5 million and 0.25 million cells were sorted for low-medium and high-level expression of N-terminal 6x-His TagRFP-T_A1aY1, respectively. The sorted cells were pelleted and washed with 1× PBS and then further lysed in RIPA buffer (Sigma-Aldrich; catalog no. R0278) with 1× PIC (Thermo Fisher Scientific; Hal Protease Inhibitor Cocktail, EDTA-free [100×], catalog no. 87785) and 1 mM PMSF (Sigma-Aldrich; catalog no. P7626). The concentration of the samples (cell lysates) was measured using a Pierce BCA protein estimation kit. To determine the level of the sensor in the lysate, 12% SDS-PAGE was performed loading 10 µg

lysate along with 1 µg, 0.8 µg, 0.4 µg, 0.2 µg, 0.1 µg, 0.05 µg, 0.02 µg, and 0.01 µg purified A1aY1-GFP-6xHis and purified goat brain tubulin. The gels were set for wet transfer on a methanol-preactivated polyvinylidene difluoride membrane for 2 h at 100 V at 4°C. The blots were probed with 1:10,000 dilution of HRP-conjugated rabbit anti-6xHis tag antibody (Abcam; catalog no. AB1187) and 1:800 dilution of mouse monoclonal DM1A antibody (Sigma-Aldrich; catalog no. T9026) for A1aY1-GFP-6xHis and goat brain tubulin, respectively, at room temperature for 2 h. Blots were washed thrice in the blocking solution for 15 min each time. The DM1A blot was further probed with HRP-conjugated anti-mouse secondary antibody (1:10,000 dilution) at room temperature for 2 h, followed by TBST wash thrice. The blots were imaged for chemiluminescence on Invitrogen iBright instrument (FL1000) for different exposures using Pierce ECL substrate (Thermo Fisher Scientific; catalog no. 32106). The blots were processed and analyzed on Fiji (ImageJ) for the signal from respective antibodies. The amount of sensor and tubulin present in the lysate was determined from the scatter plot using at least four concentrations of purified A1aY1-GFP-6xHis and tubulin. The concentration of the sensor present in a cell was estimated by the ratio of picomoles of sensor/tubulin in the lysate. We saw two bands on the SDS gel for the sensor and also for the purified recombinant A1aY1-GFP.

### Measurement of microtubule dynamic instability in stable U2OS cells (Fig. S5, D–G)

Stable U2OS cells with the red tyrosination sensor were imaged on a H-TIRF microscope with a time interval of one frame per second. The microtubule growth and depolymerization rates were measured by manually tracking individual filaments undergoing these events. Catastrophe frequency was calculated by considering the continuous catastrophe event over total time (per minute) and the length of the microtubule (per micrometer) undergoing depolymerization. Graphical representation of dynamic instability is shown in Fig. S5 F based on the calculation of the time microtubule spend in growth, pause, catastrophe and rescue phases of dynamic instability.

### Microtubule growth rate measurements in stable U2OS cells transfected with EB3-GFP (Fig. 6 B and Fig. S5, H–J)

Stable U2OS cell line expressing the red tyrosination sensor (A1aY1 with N-terminal TagRFP-T) were transiently transfected with 100–500 ng plasmid cloned with end-binding protein-3 tagged with GFP (EB3-GFP) using Jetprime transfection reagent (catalog no. 114–15; Polypus Transfection). EB3-GFP cloned vector was obtained as a gift from Dr. Carsten Janke's laboratory. Cells were imaged live on Nikon TIRF microscope for the moving EB3-GFP comets with 3-s frame intervals on 37°C stage with 5% $CO_2$ for live-cell imaging. Short movies (1–5 min) were acquired and comets were analyzed on Fiji (ImageJ) by generating kymographs (distance versus time) and manually measuring the slope to calculate the velocity of the individual comets. Experiments were performed in triplicate on two different days for control (in wild-type U2OS cells without any binder expression), stable U2OS cells (expressing TagRFP-T_A1aY1), and wild-type

U2OS cells treated with 0.5 µM SiR-tubulin + 10 µM verapamil and imaged after 30 minutes of SiR-tubulin labeling. More than 400 comets were analyzed from 1-min movies (3-s frame interval) for all the three sets (450, 468, and 414 comets analyzed for control, binder-expressing, and SiR-tubulin–treated cells, respectively) to plot a distribution of the comet velocities from each set. The mean velocities and standard deviations for each set were calculated, and distribution of velocities were plotted on Origin laboratory software.

### Drug-induced microtubule depolymerization assays (Fig. 7, A–D; and Videos 3–6)

Stable U2OS cells expressing the red tyrosination sensor (A1aY1 with an N-terminal TagRFP-T) were treated with 10 µM nocodazole (Sigma-Aldrich; catalog no. M1404), 0.5 mM colchicine (Sigma-Aldrich; catalog no. C9754), or 1 µM vincristine on a microscope stage set at 37°C with 5% $CO_2$ for live-cell imaging. A time-lapse movie was recorded for microtubule depolymerization events on a Nikon TIRF microscope with a 1–3-s frame interval. More than 30-min movies were acquired for nocodazole and colchicine-treated cells, and 5–15-min movies were acquired for vincristine-treated cells to observe near-complete depolymerization events of microtubules. Cells were analyzed manually for plotting the parameters of polymerization, depolymerization, dynamic instability, and severing events from the complete frames of the movies. Graphs were plotted on GraphPad Prism6 software.

### Cell viability assay (Trypan blue, PI, and DAPI staining; Fig. S5, A and B)

1 million wild-type U2OS cells (as control) and stable U2OS cells expressing the red or the blue tyrosination sensor were diluted 1,000 times in complete McCoy's media in separate microcentrifuge tubes. 10 µl of these 1,000-times-diluted cultures was mixed with 10 µl of 0.4% Trypan blue (Gibco; catalog no. 15250061), and the number of dead cells was counted as Trypan blue–positive cells on an automated cell counter from Invitrogen (Fig. S5 A).

For flow cytometry analysis, PI and DAPI staining was performed on 3–5 million stable U2OS cells expressing the blue tyrosination sensor (A1aY1 with a C-terminal TagBFP) and red tyrosination sensor (A1aY1 with an N-terminal TagRFP-T), respectively (Fig. S5 B). Wild-type U2OS cells were used as a control for both staining methods. Cells were pelleted at 1,000 rpm for 5 min and resuspended in 2 ml 1× PBS (8 g/liter NaCl, 0.2 g/liter KCl, 1.44 g/liter $Na_2HPO_4$, and 0.24 g/liter $KH_2PO_4$) with 5% FBS. Cells were stained with a 1-µg/ml concentration of PI and DAPI to mark the dead cells in each culture. For DAPI staining, an incubation of 15 min was performed on ice. Forward and side scatter was adjusted with a blue (488 nm) laser to mark the dense population of the cells for analysis. Percentage viability was calculated by subtracting the number of dead cells (DAPI- or PI-positive cells) from the total number of cells counted in stable U2OS cells with the tyrosination sensor and comparing it with the wild-type U2OS control. Cells stained for DAPI and PI were analyzed with a violet (405 nm) and green (561 nm) laser, respectively, on a BD FACS Aria

Fusion cell sorter at the Central Imaging and Flow Facility at NCBS.

## Transient transfection protocol

All transient transfections in mammalian cells were performed at ~60–70% cell confluency in a 35-mm ibidi dishes (ibidi; catalog no. 81156 and 81218 for polymer coverslip and glass-bottom coverslip surface, respectively) using Jetprime transfection reagent (catalog no. 114–15 polypus transfection) with 3–5 µl of the reagent used per µg of plasmid DNA as per the protocol. For transient transfection of plasmids cloned with the binder (A1aY1) tagged fluorophores (EGFP, mCherry, mCitrine, TagBFP, or TagRFP-T) in mammalian cells, 1 µg of the respective plasmids were used. Plasmids cloned with detyrosinase VASH2_X1-GFP-2A-SVBP (vasohibin2_X1-GFP separated with 2A sequence followed by SVBP), and polyglutamylases such as EYFP-TTLL5, EYFP-TTLL5, catalytic-dead mutant (E366G, ATP-deficient mutant), EYFP-TTLL4 and EYFP-TTLL7 were obtained as a kind gift from Dr, Carsten Janke's laboratory. Approximately 500 ng of these plasmids were used for transfection in stable U2OS cells expressing the red tyrosination sensor (A1aY1 with amino-terminus TagRFP-T). All transfections were performed in 10% serum–containing media, and cells were transfected for 4–8 h, followed by which the media was changed with fresh complete media. Cells were stained with 0.5–1 µM SiR-tubulin (Cytoskeleton; catalog no. CY-SC002 Spirochrome kit) for 1 h before imaging. Cells were imaged on FV3000 Olympus confocal microscope after 24 h of transfection. For 2G4C-EGFP, we used 500 ng plasmid along with 500 ng of N-terminal TagRFP-T_A1aY1 in U2OS cells. Similarly, cotransfection with 1 µg N-terminal TagRFP-T_2G4C was performed with 500 ng VASH2_X1-GFP-2A-SVBP or polyglutamylase EYFP-TTLL5.

## SPR steady-state binding assay

All the binding assays were performed on Biacore-T200 instrument from GE Healthcare Life Sciences. A GE streptavidin (SA) sensor chip (catalog no. BR100531) was immobilized with 1 µg/ml concentrations of the following biotinylated peptides; biotin-TUBA1A 440–451 (biotin-$Hs$_TUBA1A), biotin-TUBA1A 440–450 (biotin-$Hs$_TUBA1A-ΔY), biotin-TUBA1A 440–451-[E]-445 (biotin-$Hs$_TUBA1A-mG), biotin-Tub84B 438–450 (biotin-$Dm$_Tub84B), and biotin-TBA1 437–349 (biotin-$Ce$_TBA1). A single SA chip can be immobilized with three different peptides at flow channel 2 (FC2), FC3, and FC4, keeping the FC1 as a blank for buffer. Two SA sensor chips were used to perform assays with the five peptides mentioned above keeping biotin-$Hs$_TUBA1A as common on both the chips. The first chip was immobilized with biotin-$Hs$_TUBA1A, biotin-$Hs$_TUBA1A-ΔY, and biotin-$Hs$_TUBA1A-mG peptides, while the second chip was immobilized with biotin-$Hs$_TUBA1A, biotin-$Dm$_Tub84B, and biotin-$Ce$_TBA1 peptides. The surface of the SA sensor chip was preactivated with NaCl and NaOH solutions (as per the manufacturer's protocol in the GE manual for SA surface immobilization) before peptide immobilization. All the peptides were immobilized in the range of 100–300 response units. All assays were performed at a 30-µl/min flowrate in 1× HBS-P+ buffer (10 mM Hepes, 150 mM NaCl, and 0.05% vol/vol

surfactant P20; GE; catalog no. BR100671) in triplicate with two different batches of the protein. The GFP-tagged binders (A1aY1 and A1aY2) were tested for their binding parameters (association and dissociation) by titrating 8 µM of both the proteins (A1aY1-GFP and A1aY2-GFP) on the SA sensor chip immobilized with biotin-$Hs$_TUBA1A peptide (biotin-TUBA1A 440–451) and the relative response (in response units) was plotted for 180-s contact time, 300-s dissociation time, and two regeneration steps with 10 mM glycine-HCl (each time with pH 2.5) for 30 s. The sensogram obtained in Fig. S1 C showed positive binding with A1aY1-GFP but no apparent binding with A1aY2-GFP. To determine the binding interaction ($k_d$) of A1aY1, a steady-state binding assay (considering 1:1 binding) with a range of different concentrations (0.031 µM, 0.062 µM, 0.125 µM, 0.25 µM, 0.5 µM, 1 µM, 2 µM, 4 µM, 8 µM, 15 µM, 30 µM, 50 µM, 100 µM, and 200 µM) were titrated to all the immobilized peptides on FC2, FC3, and FC4 of the SA chip with a 120-s contact time, 180 s dissociation, and two regeneration steps of 30 s each with 10 mM glycine-HCl, pH 2.0 and 2.5, respectively, followed by a 60-s stabilization period between two titrations. The relative responses of binding with each peptide were determined and normalized (maximum response as 100) by subtracting the blank FC1 (as FC2-1, FC3-1, and FC4-1), and was plotted with the corresponding concentrations to fit a curve (one site total fitting on GraphPad Prism6) and determined the value of $k_d$ (Fig. 1 B).

## NMR spectroscopy

Protein samples were prepared as described above. All NMR spectra were acquired at 25°C on 800 MHz/600 MHz Bruker Avance III spectrometers equipped with a 5-mm triple-resonance CryoProbe. The sample was loaded in a 5 mm Shigemi tube. $^{1}H$–$^{15}N$ heteronuclear single-quantum coherence (HSQC) experiments were performed with 2,048 × 256 complex data points. All NMR spectra were processed using NMRPipe (Delaglio et al., 1995) and analyzed using Sparky (Lee et al., 2015).

Assignment of the backbone resonances ($^{1}H$, $^{13}C$, and $^{15}N$) of A1aY1 was performed by using the 3D triple-resonance BEST experiments (b_HNCO, b_HNCACO, b_HNCACB, and b_CBCA-CONH). $^{1}H$ and $^{13}C$ resonance assignments of side-chain resonances of A1aY1 were obtained by collecting 3D H(CC)CONH (H) CC(CO)NH spectra. The resonance list for above experiments generated by using Sparky and submitted to I-PINENMR (Lee et al., 2019) server for automatic assignment. The I-PINE results are manually cross-checked and corrected. The $^{1}H$ resonances of the $Hs$_TUBA1A peptide were assigned by 2D $^{1}H$-$^{1}H$ total correlated spectroscopy (TOCSY) and 2D $^{1}H$-$^{1}H$ nuclear overhauser effect spectroscopy (NOESY) on the free peptide. A 3D $^{13}C$/$^{15}N$-filtered (f1), $^{13}C$-edited (f3) NOESY-HSQC was collected with the complex of N$^{15}$, C$^{13}$-labeled A1aY1 and nonbiotinylated $Hs$_TUBA1A peptide to measure the intermolecular nuclear overhauser effects (NOEs). 2D $^{13}C$/$^{15}N$-filtered (f1, f2) NOESY and $^{13}C$/$^{15}N$-filtered (f1, f2) TOCSY was also collected on the complex to assign the peptide in the bound conformation and detect intrapeptide NOEs. All experimental data were processed using NMRPipe and TOPSPIN3.2 software. Analysis of NMR data

was performed using Sparky software. Backbone assignments were obtained by NMRPIPE and confirmed manually. The structural model of a1aY1 was calculated by CS-ROSETTA (Shen et al., 2009). The structural model of Hs_TUBA1A peptide was calculated in Xplor-NIH. The structural model of the complex was calculated in HADDOCK using the structures of A1aY1, peptide, Hs_TUBA1A and the measured intermolecular NOEs and chemical shift perturbations (CSPs). Rigid-body energy minimization generated 1,000 initial complex structures, and the best 200 lowest-energy structures were selected for torsion angle dynamics and subsequent Cartesian dynamics in an explicit water solvent. Default scaling for energy terms was applied. Following the standard benchmarked protocol, cluster analysis of the 200 water-refined structures yielded a single ensemble cluster. The refinement statistics of the cluster is given in Table 1.

NMR titration was performed by titrating nonbiotinylated Hs_TUBA1A and Hs_TUBA1A-ΔY peptides (ligand) to the 300-μM sample of $^{13}$C- and $^{15}$N-labeled A1aY1 (protein) on Bruker Ascend 600 MHz spectrometers equipped with triple-resonance cryoprobes and field gradients. $^1$H-$^{15}$N HSQC spectra of each residue were taken for all the titrations. For each titration point (typically 0.5, 1, 2, 3, 4, 5, 6, and 7 equivalents of ligand), a 2D water-flip-back $^{15}$N-edited HSQC spectrum was acquired with 2,048 (256) complex points and 100-ms (60-ms) acquisition times, apodised by 60 shifted squared (sine) window functions, and filled to 1,024 (512) points for $^1$H and $^{15}$N, respectively. CSPs were calculated for individual amino acids in the $^{13}$C- and $^{15}$N-labeled A1aY1 protein from the saturating titration (1:7 molar excess of the protein/peptide) with the tyrosinated Hs_TUBA1A peptide (Fig. S1 E). For each residue, the weighted average of the $^1$H and $^{15}$N CSP was calculated as CSP = $[(\Delta\delta_{HN})^2 + (\Delta\delta_N)^2/25)]^{1/2}$ (Grzesiek et al., 1996). The nonbiotinylated tyrosinated ($^{440}$VEGEGEEEGEEY$^{451}$) and detyrosinated ($^{440}$VEGEGEEEGEE$^{450}$) peptides used in the NMR titration assays were synthesized from LifeTein.

### In vitro motor gliding assay on HeLa microtubules (Fig. 6, D and E)

HeLa tubulins were obtained from Dr. Carsten Janke's laboratory. 10 μM tubulin was set for polymerization in 1× BRB80 buffer (80 mM Pipes, 1 mM MgCl$_2$, and 1 mM EGTA, pH 6.8, with KOH) in presence of biotin-labeled tubulin (1/10 molar ratio), Alexa Fluor 561–labeled tubulin (1/10 molar ratio), 2 mM GTP (Sigma-Aldrich). and 20 μM Taxol (Sigma-Aldrich) at 37°C. A glass chamber made up with coverslip glass (0.17 mm thick) was passivated with BSA-biotin (Thermo Fisher Scientific; catalog no. 29130, 1 mg/ml, 5 min), followed by a 1× BRB80 wash and 0.5 mg/ml SA (Thermo Fisher Scientific; catalog no. 43–4302) coating for 5 min. The surface was blocked with 5% Pluronic-F127 (Sigma-Aldrich; catalog no. P2443) and 1.25 mg/ml β-casein (Sigma-Aldrich; catalog no. C6905) in 1× BRB80 buffer. 10–20-fold–diluted polymerized HeLa microtubules were flowed in the flow chamber and then washed with 1× BRB80 containing 0.05% Pluronic-F127, 1.25 mg/ml β-casein, and 20 μM Taxol. The binder (2.5 μM A1aY1-GFP) and a single-molecule (nanomolar) concentration of the motors (K560-SNAP and KIF1A-LZ-SNAP labeled with 640 SNAP-tag) flowed in the flow chamber and

imaged the flow chamber with blue (488 nm), green (561 nm), and red lasers (640 nm) for binder, HeLa microtubule, and single-molecule motor movement, respectively, on a Nikon H-TIRF microscope.

### CPA treatment of tubulin and Western blot quantification (Fig. S4 E)

The method was adopted from a previous study (Webster et al., 1987a). 500 μl (200 μM) goat brain tubulin was diluted to 1 ml with 20% glycerol, 2 mM GTP, and a 2.5-μg/ml concentration of CPA from bovine pancreas (Sigma-Aldrich; catalog no. C9268) in 1× BRB80. The reaction was incubated on ice for 10 min and then at 37°C for 20 min. Immediately, the reaction was stopped by adding 20 mM DTT, and microtubules were polymerized at 37°C for another 10 min. Microtubules were pelleted at 100,000g for 40 min at 37°C. The microtubule pellet was resuspended in 1 ml ice-cold 1× BRB80 on ice (obtained concentration, 75 μM). For Western blot, 12% SDS-PAGE gel was loaded with 2 μg of the CPA-treated tubulin and goat tubulin (control) in triplicate. The blot was transferred on activated polyvinylidene difluoride membrane using a trans-blot turbo transfer system (Bio-Rad; catalog no. 1704150) for 30 min. The blot was blocked in 5% skim milk made in 1× TBST. The blot was cut in three parts, each containing CPA tubulin and goat tubulin. The blots was incubated for 2 h at room temperature with 1:4,000 mouse anti-tyrosinated α-tubulin antibody (Sigma-Aldrich; clone YL1/2, catalog no. MAB1864-I), 1:4,000 rabbit anti-detyrosinated α-tubulin antibody (Abcam; catalog no. ab48389), and 1:800 mouse anti-α tubulin antibody (Sigma-Aldrich; catalog no. T9026) in the above blocking solution, respectively. Blots were washed thrice in the blocking solution for 15 min each time. The blots were further kept for secondary antibody incubation with 1:10,000 dilution of peroxidase-conjugated anti-mouse or anti-rabbit antibodies (Sigma-Aldrich). The blots were imaged for chemiluminescence on Invitrogen iBright instrument (FL1000) for different exposures using Pierce ECL substrate (Thermo Fisher Scientific; catalog no. 32106). The blots were processed and analyzed on Fiji (ImageJ) for the signal from respective antibodies. The graph was plotted for the signal of tyrosinated versus detyrosinated tubulin over total tubulin (dm1a) using GraphPad Prism6 software (Fig. S4 E).

### In vitro microtubule binding assay (Fig. 2, A and B)

The CPA-treated tubulins were polymerized in presence of different ratios of HeLa tubulins (0%, 25%, 50%, 75%, and 100%) with 2% (Alexa Fluor 561/640) labeled tubulin in presence of GTP and Taxol as described above. Similarly, HeLa tubulins were also polymerized (as a control) and labeled with 2% labeled tubulin (Alexa Fluor 561 or 640). Both the differently fluorescent-labeled microtubules (Alexa Fluor 561 and Alexa Fluor 640) were flowed into the same glass chamber, which was passivated by kinesin-3 (Kif1a, 1-357) motor bed to facilitate the surface immobilization (in absence of ATP) of the microtubules. The immobilised microtubules were further incubated with 5 μM A1aY1-GFP. The chamber was imaged for HeLa microtubules, chimeric microtubules of HeLa and CPA tubulins, and binder-GFP using red (640 nm), green (561 nm), and blue (488 nm)

lasers on a Nikon H-TIRF microscope. The signal of binder over CPA/HeLa microtubules (different levels of detyrosination) was divided by binder signal on HeLa (pure tyrosinated) microtubules for different fractions of CPA tubulins and plotted on GraphPad Prism6. For 0%, 25%, 50%, and 75% HeLa microtubules, experiments were performed on two different days ($n = 4$), with two different batches of purified A1aY1-GFP and a total of 150 microtubules analyzed for each set by measuring the signal over CPA/HeLa microtubule/pure HeLa tubulin (subtracting the nearby background fluorescence for each filament). For 100% HeLa microtubules, a total of 167 filaments (subtracted background fluorescence) were analyzed from $n = 3$ experiments (Alexa Fluor 640 HeLa microtubule signal/Alexa Fluor 561 HeLa microtubule signal).

### Immunostaining of U2OS and H9C2 cells using purified A1aY1-GFP (Fig. 2 C)

U2OS cells were plated on ibidi glass-bottom dishes, and H9C2 cells were plated on ibidi polymer-bottom dishes at ~50–70% density. Cells were processed for fixation after at least 24 h from seeding. Cells were washed twice with 1× BRB80 and fixed in ice-cold methanol for 5 min at –20°C. Cells were washed and permeabilized using 0.1%-Triton-X-100 and 0.05–0.1%-Saponin made in 1× BRB80 for 10 min at room temperature. Cells were blocked for 1 h at room temperature with 5% BSA made in 1× BRB80. Cells were further kept for primary antibody incubation with 1:1,000 mouse anti-tyrosinated α-tubulin antibody (Sigma-Aldrich; clone YL1/2, catalog no. MAB1864-I), 1:1,000 rabbit anti-detyrosinated α-tubulin antibody (Abcam; catalog no. ab48389) overnight at 4°C. Cells were washed thrice for 15 min each at room temperature and kept for secondary antibody incubation with 1:500 Alexa Fluor 546 goat anti-rabbit and 1:500 Alexa Fluor-647 goat anti-mouse for 2 h at room temperature. Cells were washed thrice and kept for overnight incubation with 12 µM A1aY1-GFP at 4°C. Cells were washed thrice and imaged on Olympus FV3000 confocal microscope.

### 3D-SIM (Fig. 8)

For 3D-SIM acquisition, U2OS cells stably expressing the red tyrosination sensor (TagRFP-T_A1aY1) were plated on glass-bottom (#1.5H) ibidi dishes and examined using Nikon N-SIM fitted with 100×/1.49 SR Plan Apo TIRF oil-immersion objective. Image stacks (z-steps of 0.2 µm) were acquired with Andor iXon3 (DU-897) electron multiplying charge-coupled device camera. Exposure conditions were adjusted to get a typical yield of ~3,000 maximum counts while keeping the bleaching minimal. Image acquisition, SIM image reconstruction, and data alignment were performed using NIS-Elements 4.2 software (Nikon). Image reconstruction was done by varying image modulation contrast at 0.5–1.0, high-resolution noise suppression at 0.50–1.0, and out-of-focus blur suppression at 0.1–0.2.

### *FWHM estimation*

Line profiles of several microtubules from 10 labeled cells were obtained and fitted with a Gaussian distribution using Fiji image processing software. For the profile, a line width of 165 nm was chosen to cover a minimum width of four pixels in SIM images.

Microtubule widths in SIM stacks were measured in the z-axis section where the microtubule fluoresced most strongly. The FWHM was calculated with the following formula, where σ is the Gaussian width parameter: $FWHM = 2.355\sigma$.

### Cell fixation, immunofluorescence, and imaging for VASH2_X1-GFP-2A-SVBP (Fig. S4, A–D) and CLIP-170 staining (Fig. 6 C)

HeLa cells plated on glass coverslips were transduced with lentiviruses encoding different constructs of mVash-mSVBP for 24 h and fixed according to previously described protocols (Magiera and Janke, 2013). Briefly, the cellular proteins were cross-linked using the homobifunctional cross-linker dithio-bis(succinimidyl propionate) (Thermo Fisher Scientific; #22585) diluted in microtubule-stabilizing buffer, followed by fixation with 4% PFA for 15 min. The cells were then permeabilized with 0.5% Triton-X-100 in microtubule-stabilizing buffer for 5 min and blocked in 5% BSA prepared in PBS containing 0.1% Triton-X-100 (PBS-T).

Cells were incubated with primary antibodies (anti-tyrosinated tubulin antibody YL1/2, 1/5,000 [Abcam; #ab6160], anti-detyrosinated tubulin antibody, 1/1,000 [Merck; #AB3201], anti-α-tubulin antibody 12G10, 1/1,000 [developed by J. Frankel and M. Nelson, obtained from the Developmental Studies Hybridoma Bank, developed under the auspices of the National Institute of Child Health and Human development (NICHD), and maintained by the University of Iowa]) diluted in PBS-T containing 5% BSA (blocking solution) for 2 h at room temperature. Cells were then incubated with secondary antibodies (goat anti-rabbit Alexa Fluor 568, 1:1,000; Thermo Fisher Scientific, #A11036; goat anti-rat Alexa Fluor 594, 1:1,000; Thermo Fisher Scientific, #A-11007; and goat anti-mouse Alexa Fluor 568, 1:5,000; Thermo Fisher Scientific, #A11019) prepared in blocking solution and incubated for 1 h at room temperature. Coverslips were mounted using ProLong Gold anti-fade medium (Thermo Fisher Scientific; #P36930).

CLIP-170 antibody was a gift from Dr. Anna Akhmanova (Utrecht University, Utrecht, Netherlands). U2OS cells were transduced with lentivirus encoding the red tyrosination sensor (N-terminal TagRFP-T_A1aY1) or with VASH2_X1-GFP-2A-SVBP for 48 h, fixed in –20°C cold methanol for 15 min, and immediately transferred to 4% PFA for 15 min at room temperature. Cells were washed three times with 0.15% Triton-X-100, blocked for 1 h in 1% BSA solution prepared in PBS containing 0.15% Tween-20. Cells were then incubated with CLIP170 antibody (1:300 dilution in blocking solution) for 2 h. Washes after primary and secondary antibodies were performed using blocking solution as described above.

Images were acquired using Optigrid (Leica) with a 63× (NA 1.40) oil-immersion objective or 100× oil-immersion objective for CLIP170 images and the ORCA-Flash4.0 camera (Hamamatsu), operated through Leica MM AF imaging software. Images were processed using ImageJ v1.51a (National Institutes of Health), and final figures were prepared using Adobe Photoshop and Illustrator.

### Sample preparation and immunoblotting

HeLa cells transduced with lentiviruses encoding for different constructs of mVash-mSVBP for 48 h were directly collected in

2× Laemmli buffer (180 mM DTT [Sigma-Aldrich; #D9779], 4% SDS [VWR; #442444H], 160 mM Tris-HCl, pH 6.8, 20% glycerol [VWR; #24388.295], and bromophenol blue). The samples were then boiled at 95°C for 5 min, spun down at 20,000*g* for 5 min using a tabletop centrifuge, and stored at −20°C. Immunoblotting was performed according to previously described protocols(Magiera and Janke, 2013). Briefly, SDS-PAGE gels were prepared at 375 mM Tris-HCl, pH 9.0, 0.1% SDS (Sigma-Aldrich; #L5750) and 10% acrylamide (40% acrylamide solution [Bio-Rad; #161-0140] supplemented with 0.54% bis-acrylamide [wt/vol] powder [Bio-Rad; #161-0210]). Samples were loaded on SDS-PAGE gels, separated, and transferred onto a nitrocellulose membrane (Bio-Rad; #1704159) using Bio-Rad Trans-Blot Turbo system, according to manufacturer's instructions. Membranes were blocked for 1 h in 5% nonfat milk prepared in PBS-T. After blocking, membranes were incubated with primary antibodies (anti-tyrosinated tubulin antibody YL1/2, 1/5,000 [Abcam; #ab6160], anti-detyrosinated tubulin antibody, 1/5,000 [Merck; #AB3201], polyclonal rabbit anti-GFP antibody, 1:5,000 [Torrey Pines Biolabs; TP401], anti-α-tubulin antibody 12G10, 1/1,000 [developed by J. Frankel and M. Nelson, obtained from the Developmental Studies Hybridoma Bank, developed under the auspices of the NICHD, and maintained by the University of Iowa]) for prepared in PBS-T containing 2.5% nonfat milk for 2 h. Membranes were washed four times with PBS-T and then incubated with HRP-conjugated secondary antibodies (goat anti-rabbit, 1:10,000 [Bethyl; #A120-201P], goat anti-mouse, 1:10,000 [Bethyl; #A90-516P], and goat anti-rat, 1:10,000 [Bethyl; #A110-236P]) for 1 h. Membranes were then washed four times in PBS-T, and chemiluminescence signal on the membrane was revealed using Clarity Western ECL substrate (Bio-Rad; #1705060) solution.

## DNA constructs and lentivirus production for vasohibin and SVBP

2G4C is synthesized from Eurofins as a fragment with ∼15 bp of sequence homology to the ends of target vector. The lyophilized fragment is resuspended in water, cloned via a sequence- and ligation-independent cloning (Jeong et al., 2012) method into pTRIP vector (having EGFP sequence) at the NheI site, and verified by sequencing. For later use in mVASH2-X1-GFP_2A_mSVBP and TTLL5 experiments, 2G4C was cloned with TagRFP-T containing pTRIP vector. mVash and mSVBP were cloned in to pTRIP lentiviral vectors using sequence- and ligation-independent cloning (Jeong et al., 2012), which was described in detail previously (Bodakuntla et al., 2020a). Briefly, mSVBP was amplified from cDNA prepared from cultured hippocampal neurons using primers with at least 15 bp of homology sequence to the ends of the pTRIP vector with CMV-enhanced CAG promoter at the XhoI site. mSVBP was cloned after a self-cleavable 2A peptide sequence (Kim et al., 2011), which was in frame with the GFP. mVash2_X3 was amplified from brain cDNA and cloned into pTRIP-mSVBP vector at the NheI site to generate mVash2_X3-2A-mSVBP. mVash1 and mVash2_X1 were amplified from testis cDNA and cloned into pTRIP-mSVBP vector at the NheI site to generate mVash1-2A-mSVBP and mVash2_X1-2A-mSVBP, respectively. Primers used for amplification are as follows: mSvbp-FS-2A: 5′-tccactagtgtcgacATGACTACTGTCCCA

TTGTGCAGG-3′, mSvbp-RS-2A: 5′-ttttctaggtctcgagTtACTCCCCAGGCGGCTGCATCTG-3′, mVash2-FS1: 5′-agaattattccgctagcATGACCGGCTCTGCCGCCGACAC-3′, mVash2-RS1: 5′-accatGGTGGCgctagc GATCCGGATCTGATAGCCCACTTCG-3′, mVash1-FS1: 5′-agaattattccgctagcATGCCAGGGGGAAAGAAGGTGGTC-3′ and mVash1-RS1: 5′-accatGGTGGCgctagcCACCCGGATCTGGTACCCACTGAG-3′. Packaging plasmids psPAX2 and pCMV-VSVG were gifts from D. Trono (École polytechnique fédérale de Lausanne, Lausanne, Switzerland; Addgene; plasmid #12260) and B. Weinberg (Whitehead Institute for Biomedical Research, Cambridge, MA; Addgene; plasmid #8454), respectively.

Lentivirus particles were produced using protocols described earlier (Bodakuntla et al., 2020b). Briefly, X-Lenti 293T cells (Takara; #632180) were cotransfected with 1.6 µg plasmid of interest and packaging plasmids (0.4 µg of pCMV-VSVG and 1.6 µg of psPAX2) using 8 µl TransIT-293 (Mirus Bio; #MIR 2705) transfection reagent per well of a 6-well plate. The next day, the culture medium of the X-lenti cells was changed to Neurobasal medium (Thermo Fisher Scientific; #21103049) containing 1× PenStrep (Life Technologies; #15140130). After 24–30 h of media change, the virus-containing supernatants were passed through a 0.45-µm filter and either used fresh or aliquoted and stored at −80°C. To determine the optimal amount of virus for transduction, lentivirus aliquots were thawed, and different volumes were added to HeLa cells. Based on the expression levels of the GFP protein, the desired amount of virus to achieve maximum transduction efficiency was determined.

## Image acquisition and analysis

All confocal imaging was performed on Olympus FV3000 inverted microscope equipped with 60× oil objective (1.42 NA), six solid-state laser lines (405, 445, 488, 514, 561, and 640 nm), two high-sensitivity spectral detectors (SDs) and two SD detectors. All the images were acquired in 2,048 × 2,048 frame (at 60× objective) marking the region of interest for acquisition. Optical sections of 0.5–1 µm (step size) are z-projected in all image panels shown in this study. For all image acquisitions, high-sensitivity photomultiplier tubes were used as a detector (HSDs) for sequential or simultaneous imaging of samples labeled with two different fluorophores having overlapping or far-separated spectral properties, respectively. For more than two-color image acquisition, the HSDs were used in sequential mode separating the fluorophores in two phases. All images acquired were analyzed on Fiji software (ImageJ) using the appropriate plugins.

For the colormap, Fiji plugin "colocalization colormap" was used to obtain the colocalization value (index of correlation) and the colormap distribution. Pearson's correlation coefficient (R-value) calculation in Fig. 4 B was carried on z-projections of the images acquired on FV3000 confocal microscope for each enzyme overexpression. For each enzyme experiment, the confocal stacks were z-projected for TagRFP-T (561-nm laser channel), enzyme channel (488-nm laser), and SiR-tubulin channel (640-nm laser channel). A 50-pixel background was subtracted from 561 (sensor) and 640 (SiR-tubulin) channels. The colocalization was calculated using ImageJ coloc-2 plugin. Pearson's R-value (no threshold) was calculated for each image, which represents

the percentage overlap of the fluorescent intensity TagRFP-T over SiR-tubulin.

Single-plane microscopy imaging for microtubule dynamics, growth rate (EB3 experiments), motor movement, and drug-induced microtubule depolymerization experiments was performed on a Nikon Ti2 H-TIRF microscope with a 100× oil objective (1.49 NA) connected with four solid-state lasers lines (405, 488, 561, and 640 nm) under total internal reflection mode. The images were acquired using an appropriate filter set using the s-CMOS camera (Hamamatsu; Orca Flash 4.0) controlled by Nikon NIS-elements software. All the images were analyzed on Fiji (ImageJ) software to construct movies and kymographs to measure the velocities/processivity (run length) of motors and EB3 comets. For measuring the number of comets per square micrometer per minute, a single-cell image from the TIRF movies was extracted by cropping the area around cell boundary and making the intensity outside the cell as zero, using Fiji software. Total five cells were considered each for control and TagRFP-T_A1aY1 expressing cells transfected with EB3-GFP. All the images were thresholded for all 20 frames (3-s interval, 1 min total movie) to completely remove the background from the image. The binary image obtained was analyzed to count the number of comets in all 20 frames, which was further divided with the area of the cell to plot the graph in Fig. S5 H.

All live-cell experiments using U2OS cells were performed in McCoy's 5A complete media (with 10% FBS and antibiotics) on a 37°C preheated microscopy stage in a moist-air chamber regulated with 5% $CO_2$.

### Graphs and statistical analysis
All graphs and statistical analyses were performed on Origin laboratory (learning edition) or commercially available Graph-Pad Prism6 softwares. The SPR steady-state binding graph in Fig. 1 B; line scan intensity profile in Fig. 4 A; Fig. 5; Pearson's coefficient graph in Fig. 4 B; sensogram of Fig. S1 C; CSP plot in Fig. S1 G; and graphs in Fig. 7, E and F; and Fig. S5, F–H were plotted on GraphPad Prism6 software. All graphs in Fig. 6, B, D, and E for EB3-GFP comet, K560, and Kif1a motor velocities and processivity and Fig. S5 D were plotted for histograms/distributions and Gaussian fit on Origin laboratory software. For microtubule growth and depolymerization rate measurements, manual tracking of individual microtubules was performed using Fiji software by drawing a segmented line over the microtubule undergoing such events. The distributions of the growth and depolymerization rates were plotted on origin laboratory software to calculate mean ratio and SD (Fig. S5 D). Catastrophe frequency was calculated by considering continuous catastrophe events over time (total; per minute) and the length of the microtubule (per micrometer) undergoing depolymerization. The values were plotted on GraphPad Prism6 software (Fig. S5 G). All the kymographs displayed are analyzed on Fiji software. Statistical analysis performed for mean ratio, SD, SEM, and $n$ value are mentioned in respective figure legends.

### Data availability
The NMR structure coordinates for the A1aY1–α-tubulin CTT complex have been deposited in the PDB under accession no. 7C1M.

### Online supplemental material
Fig. S1 shows FACS-based screening for enrichment of the A1aY1 sequence and NMR structural determination and titrations with the α CTT peptides. Fig. S2 shows z-projected confocal images of U2OS cells transfected with EGFP, mCherry, mCitrine, TagBFP, and TagRFP-T fused A1aY1 constructs. Fig. S3 shows differential expression of the A1aY1 binder at low/medium and high levels and quantification of total binder concentration inside the cell. Fig. S4 shows comparison and characterization of different vasohibins and CPA treatment of tubulin. Fig. S5 shows cell viability assays and measurement of microtubule dynamic instability parameters. Video 1 shows visualization of microtubule dynamics using red tyrosination sensor (Fig. 6 A). Video 2 shows microtubule growth rate measurements (Fig. 6 B; and Fig. S5, H–J) using EB3-GFP (green), transfected in stable U2OS cells expressing red tyrosination sensor (gray). Video 3 shows stable U2OS expressing the red tyrosination sensor as a control (no drug treatment) undergoing normal dynamic instability behavior (Fig. 7 A). Video 4 shows microtubule depolymerization induced by treating stable cells expressing the red tyrosination sensor with 10 µm nocodazole (Fig. 7, B and E). Video 5 shows microtubule depolymerization and severing events induced by treating stable cells expressing the red tyrosination sensor with 500 µm colchicine (Fig. 7, C, E, and F). Video 6 shows microtubule depolymerization and severing events induced by treating stable cells expressing the red tyrosination sensor with 1 µm vincristine (Fig. 7, D–F). Video 7 shows 3D-SIM reconstructions of interphase and mitotic phase of U2OS cells stably expressing red tyrosination sensor (Fig. 8).

## Acknowledgments
The authors acknowledge the NMR facility of Biophysics Core and the Central Imaging and Flow Facility at NCBS/inStem/Centre for Cellular and Molecular Platforms campus. The authors thank V. Henriot (Institut Curie, Paris, France) for technical assistance and A. Akhmanova (University of Utrecht, Utrecht, Netherlands) for kindly providing essential reagents.

S. Kesarwani and P. Lama are supported by the inStem Graduate Program. A. Chandra is a Wellcome Trust Department of Biotechnology (DBT) India Alliance Early Career Fellow (IA/E/15/1/502339). C. Janke is supported by the Institut Curie, Agence Nationale de la Recherche award ANR-17-CE13-0021, Institut National du Cancer grant 2014-PL BIO-11-ICR-1, and the Fondation pour la Recherche Medicale grant DEQ20170336756. S. Bodakuntla was supported by Fondation pour la Recherche Medicale grant FDT201805005465. Jijumon A.S was supported by the European Union's Horizon 2020 research and innovation program under Marie Skłodowska-Curie grant agreement 675737, and Fondation pour la Recherche Medicale grant FDT201904008210. The NMR facility is supported by a DBT B-life grant (BT/PR5081/INF/156/2012). The project was partly supported by Tata Institute of Fundamental Research. R. Das acknowledges support from a DBT-Ramalingaswamy fellowship (BT/HRD/23/02/2006). M. Sirajuddin acknowledges support from inStem core grants from the Department of Biotechnology, India, DBT/Wellcome Trust India Alliance Intermediate Fellowship

(IA/I/14/2/501533) and European Molecular Biology Organization Young Investigator award. C. Janke and M. Sirajuddin are jointly funded by CEFIPRA (5703-1).

S. Kesarwani, B.M. Rao, and M. Sirajuddin are inventors on a provisional patent application related to the usage and application of A1aY1 binder sequence. The remaining authors declare no competing financial interests.

Author contributions: S. Kesarwani and M. Sirajuddin conceptualized the project. S. Kesarwani screened, validated, and performed the microscopy experiments with the binder. P. Lama carried out the motor motility and in-vitro binding experiments. A. Chandra and S. Kesarwani performed 3D-SIM imaging. S. Bodakuntla, A.S. Jijumon, and C. Janke provided and validated the reagents for PTM specificity work and HeLa tubulin. S. Bodakuntla performed the CLIP-170 experiments in cells. P. Purushotam Reddy and R. Das determined the NMR structure. B.M. Rao provided the yeast display library and valuable inputs in screening. M. Sirajuddin supervised the project. S. Kesarwani and M. Sirajuddin wrote the manuscript with inputs from all authors.

Submitted: 19 December 2019

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

# Supplemental material

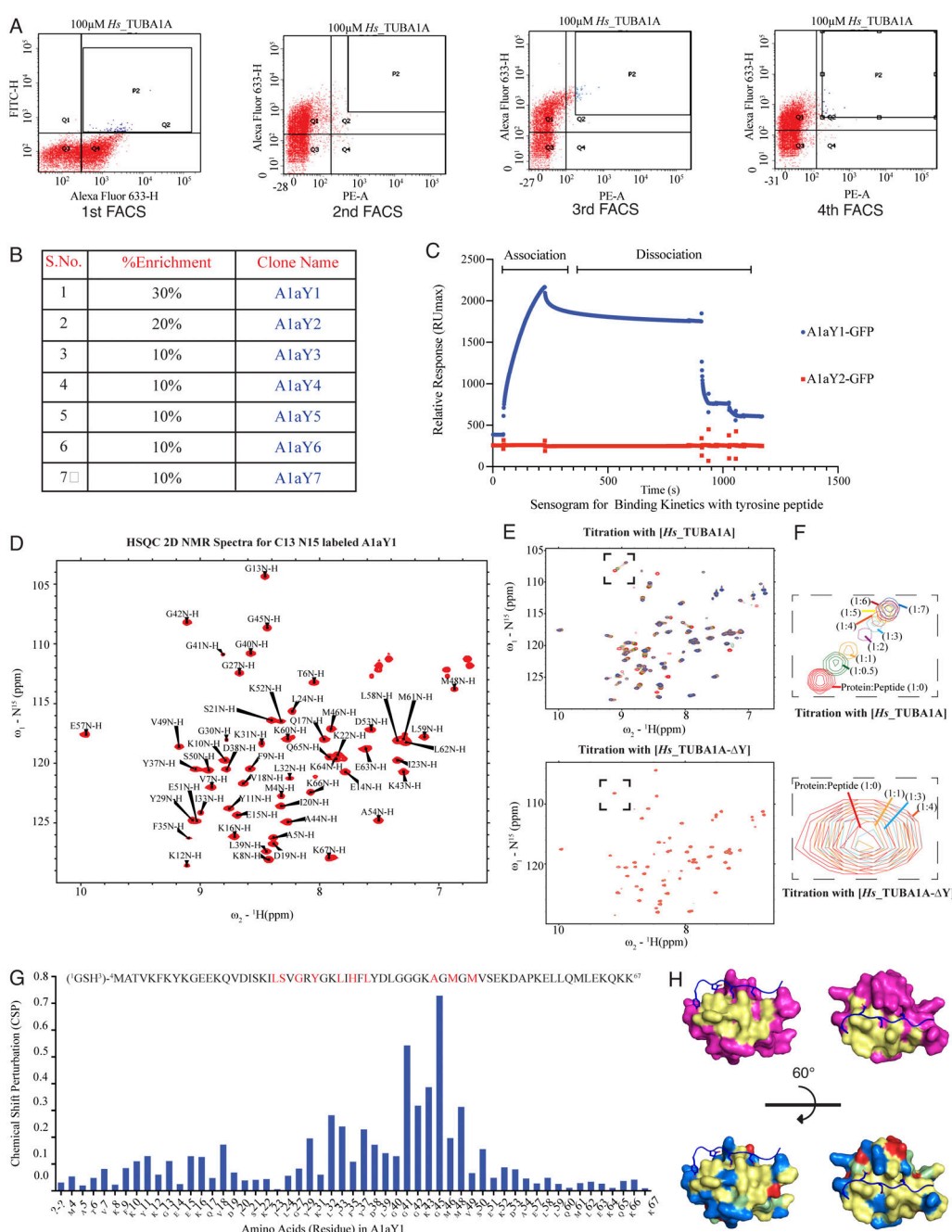

Figure S1. **FACS enrichment and NMR-based structural determination and titrations with the α CTT peptides. (A)** Representative images for the 10,000 yeast cells sorted on BD FACS Aria fusion cell sorter. Four rounds of sorting experiments were performed. The population P2, marked in blue (Q2 quadrant), is collected (>4,000 cells during each sorting). Q3 quadrant represents the unstained population. Expression of the binders in the SSO7D library was marked by labeling the c-myc tag in Alexa Fluor 633 channel (Q4 in first FACS and Q3 in second, third, and fourth FACS), and the binder-bound peptide was labeled with FITC (first FACS) or PE (second, third, and fourth FACS) channel (see Materials and methods). **(B)** The table represents the abundance (% enrichment) of the clones obtained after the fourth sorting after analyzing 10 single yeast colonies. S.No., serial number. **(C)** Sensogram obtained from the immobilized biotin-Hs_TUBA1A peptide on an SA surface in the SPR experiment to determine the binding parameters of the two abundant clones (A1aY1 and A1aY2) for their binding parameters. Binders (A1aY1 and A1aY2) were purified with GFP tag at the C terminus and were used at ~8 µM concentrations to obtain the binding sensogram. RU, response unit. **(D)** 2D NMR spectra (HSQC spectra) and assignment of the peaks for corresponding amino acids in A1aY1 protein labeled with $^{15}$N and $^{13}$C isotopes. The x axis of the HSQC spectra represents the proton ($^1$H) shift and the y axis the $^{15}$N shift. **(E)** HSQC spectra of the $^{13}$C $^{15}$N–labeled A1aY1 protein in presence of α-tubulin CTT peptides (Hs_TUBA1A and Hs_TUBA1A-ΔY). A titration of A1aY1 protein (200 µM) with 1:0.5, 1:1, 1:2, 1:3, 1:4, 1:5, 1:6, and 1:7 excess of Hs_TUBA1A peptides showed a positive chemical shift in the residues of A1aY1, while titration with Hs_TUBA1A-ΔY in 1:1, 1:2, 1:3, and 1:4 excess did not show any significant change in the position of the residues. **(F)** Zoomed-in residue from the titration of Hs_TUBA1A and Hs_TUBA1A-ΔY (marked with black dashed box). **(G)** Values of the CSPs calculated from the ppm shift in the residues upon titration with Hs_TUBA1A. The protein sequence of the binder, A1aY1 with diversified residues (marked in red) are marked above the graph. **(H)** Left: The NMR structure of A1aY1 binder (magenta) showing the randomized residues (yellow) in SSO7D structure lying in the binding pocket for Hs_TUBA1A peptide. Right, A1aY1 binder colored according to surface charge; basic (blue), acidic (red), hydrophobic (yellow) and polar (green).

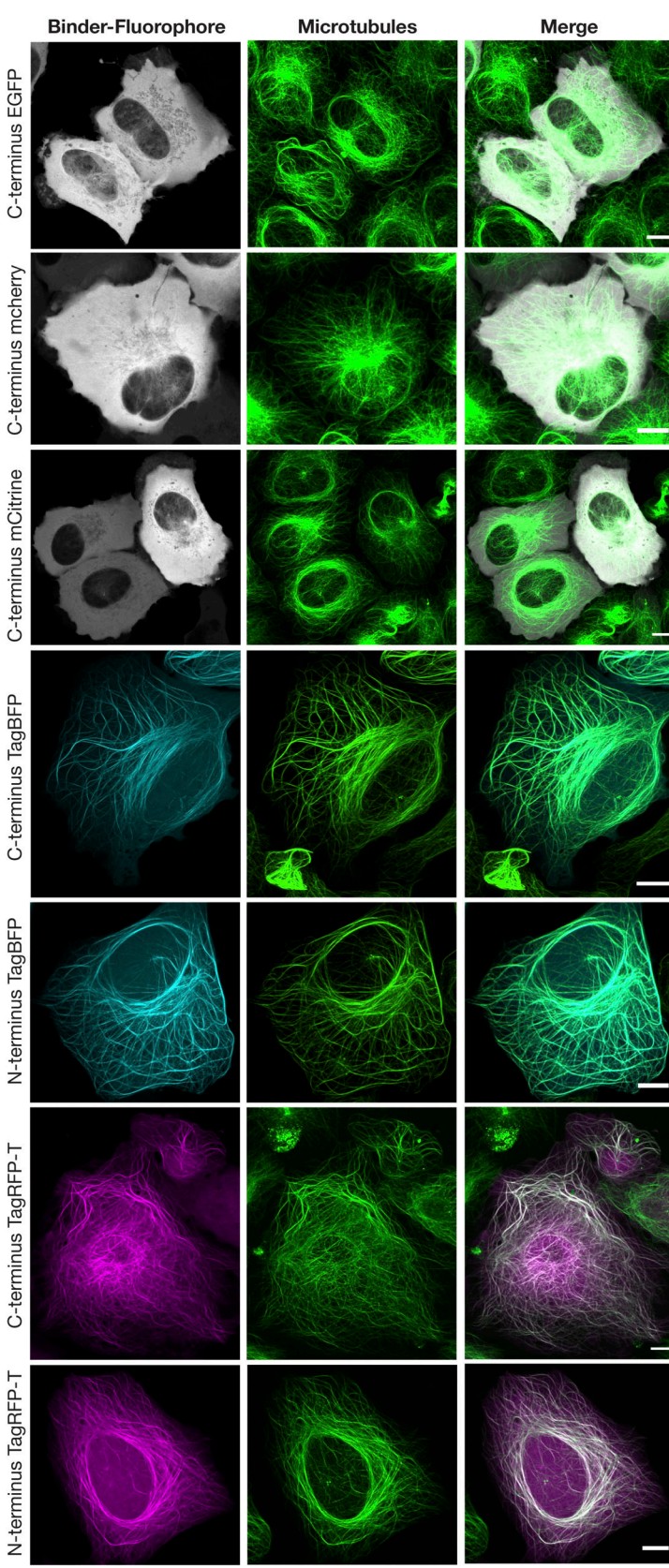

Figure S2. **Transient transfection in wild-type U2OS cells with C-terminal EGFP (gray), mCherry (gray), mCitrine (gray), TagBFP (cyan), TagRFP-T (magenta) and N-terminal TagBFP (cyan), and TagRFP-T (magenta) fluorophore-tagged A1aY1 binder.** Total cellular microtubules were labeled with SiR-tubulin (shown in green). Scale bars = 10 µm.

Figure S3. **Differential expression of the A1aY1 binder stably incorporated in U2OS cells. (A)** Blue sensor (TagBFP A1aY1). **(B)** Red sensor (TagRFP-T A1aY1). Upper and lower panels show representative images of cells expressing low-medium and high levels of A1aY1 fused to either TagBFP or TagRFP-T, respectively. Scale bars = 10 µm. **(C and D)** Representative image of FACS of 10,000 nontransfected cells (C, marked in red) and transfected cells with N-terminal 6xHis TagRFP-T_A1aY1 (D, marked in purple and blue). The cells were sorted into two bins, low/medium (in purple) and high expression (blue), based on the fluorescence intensity of TagRFP-T. The left panels in C and D represent the forward- versus side-scattered plot of the cell population considered for sorting and marked as P1 population. **(E and F)** Western blot images for the estimation of the N terminus 6xHis TagRFP-T_A1aY1 (6xHis tag red tyrosination sensor) and total tubulin levels. A total of 10 µg protein lysate was loaded for each low/medium- and high-level expression FACS sorted cell lysate samples (Materials and methods). Purified 6xHis-A1aY1-GFP and goat brain tubulin of known concentrations were loaded to estimate the concentration of the sensor and tubulin in the lysate. Blots were probed with anti-hexahistidine (E) and mouse monoclonal DM1A antibody (F) for detection of 6xHis-A1aY1-GFP and tubulin samples, respectively. The expected bands for the sensor and the tubulin is marked in red box on top of the blot. The sensor and the recombinant purified A1aY1-GFP migrates as two bands in the SDS gel. **(G and H)** Scatter plots for the gel band intensities versus concentrations of purified 6xHis-A1aY1-GFP (G) and goat brain tubulin (H). The titration curve was used to estimate the concentration of sensor and total tubulin in the cell lysate as shown in I. **(I)** Table comparing the concentration of the sensor in FACS sorted cell lysates from low-medium and high-level expression of the N-terminal 6xHis TagRFP-T_A1aY1. * and ** denote the low/medium- and high-expression cell samples and is expressed as ~1.5- and ~3-fold excess compared with the total tubulin levels in the cells, respectively. A.U., arbitrary units.

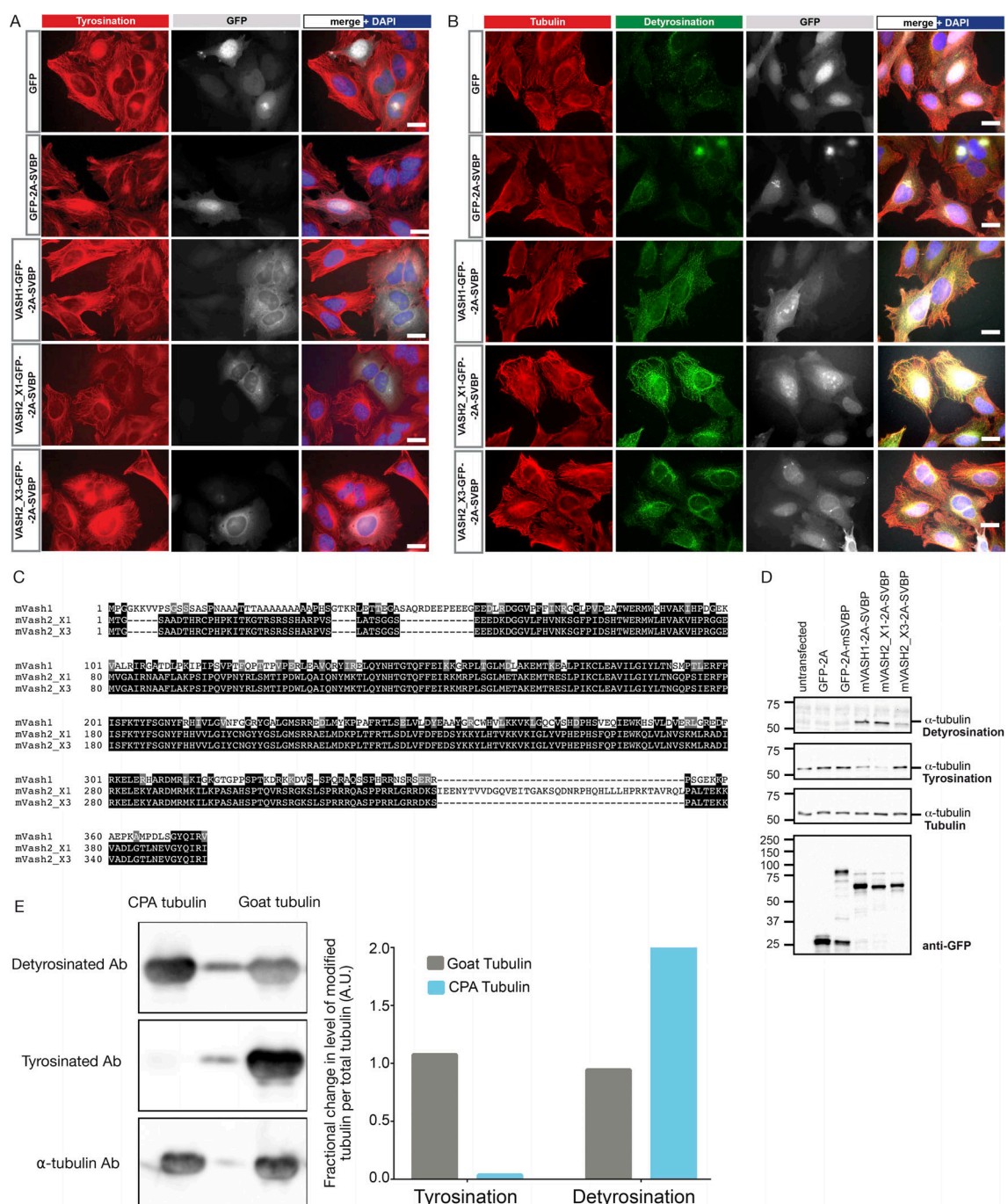

Figure S4. **Optimisation of detyrosination activity of vasohibins and CPA enzymes. (A)** HeLa cells transduced with lentivirus encoding different constructs of mVash-mSvbp, namely GFP-2A-mSVBP, mVash1-GFP-2A-mSVBP, mVash2_X1-GFP-2A-mSVBP, mVash2_X3-GFP-2A-mSVBP, and GFP-alone were fixed and stained with tyrosination antibody (shown in red color). GFP expression and nuclear staining (DAPI) were shown in gray and blue colors, respectively. Of all the vasohibin-constructs, mVash1 and mVash2_X1 show a clear reduction in microtubule tyrosination levels. Scale bars = 20 μm. **(B)** HeLa cells transduced with lentivirus encoding different constructs of mVash-mSvbp; GFP-2A-mSVBP, mVash1-GFP-2A-mSVBP, mVash2_X1-GFP-2A-mSVBP, mVash2_X3-GFP-2A-mSVBP, and GFP-alone were fixed and stained with antibodies against tubulin (shown in red color) and detyrosination (shown in green). GFP expression and nuclear staining (DAPI) are shown in gray and blue, respectively. Note that in contrast to mVash1 and mVash2_X3, most of the microtubules in cells expressing mVash2_X1 are detyrosinated. Scale bars = 20 μm. **(C)** Sequence alignment of mVash1 and splice isoforms of mVash2 (mVash2_X1 and mVash2_X3) is shown. Identical amino acids across these proteins are highlighted in black and amino acids with similar charge are highlighted in gray. mVash2_X1 contains an extra stretch of amino acid sequence in the N-terminal region. **(D)** Immunoblot analyses of HeLa cell lysates transduced with lentivirus encoding different constructs of mVash-mSVBP. Blots were probed with antibodies against detyrosination, tyrosination, and α-tubulin and with anti-GFP for visualizing the expression of GFP-tagged mVash proteins. Note the increase of detyrosination and decrease of tyrosination in the extracts expressing mVash1 and mVash2_X1. **(E)** Western blot analysis and quantification of detyrosination levels of CPA-treated tubulin used in Fig. 2 A. The gap lane between CPA-treated and untreated goat brain tubulin lanes showing tubulin staining with antibodies is likely due to spillage from either of the neighboring lanes. A.U., arbitrary units.

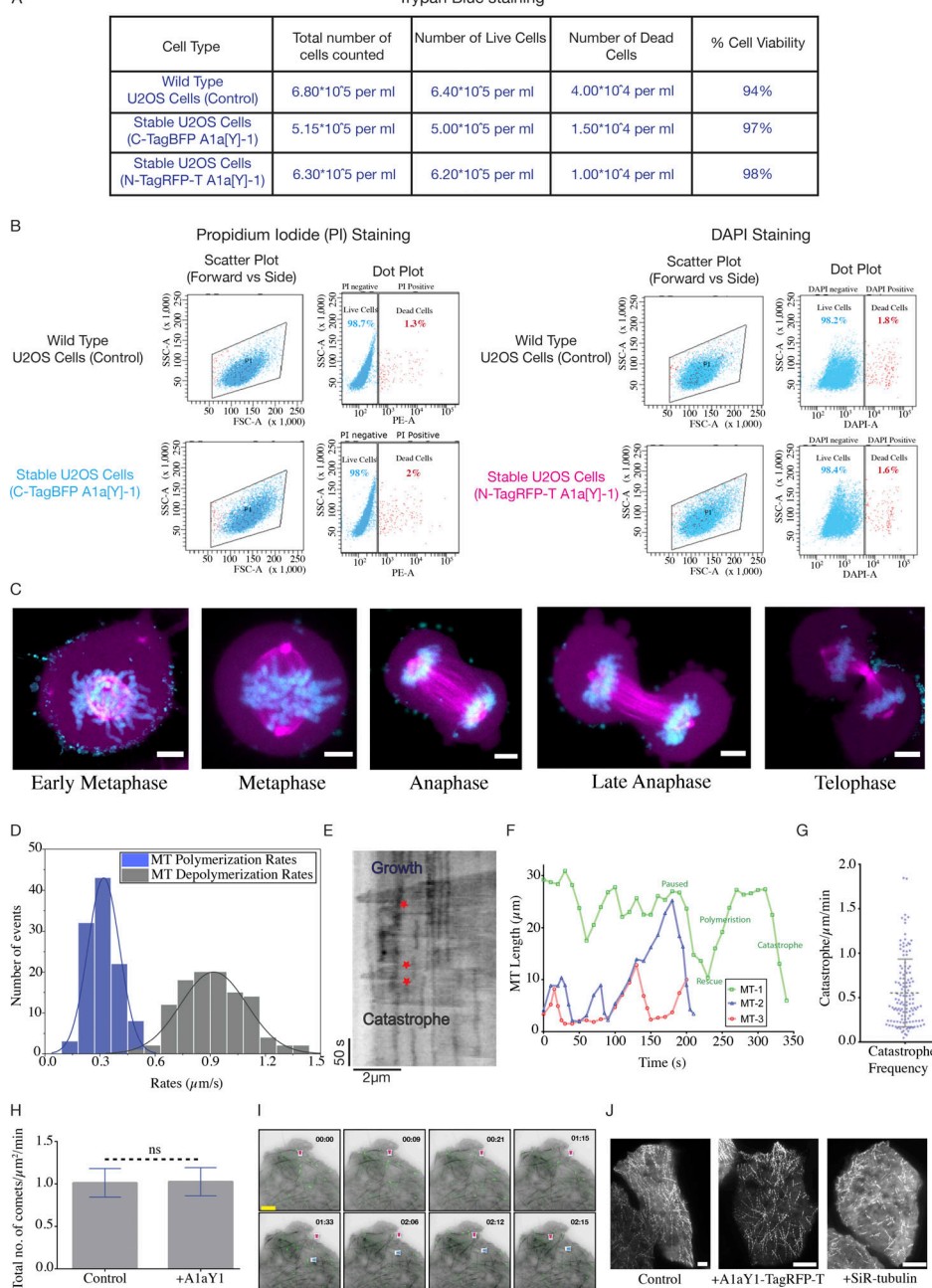

Figure S5. **Cell viability assays and microtubule dynamics measurements**. **(A)** Trypan blue staining of the wild-type U2OS cells and stable U2OS cells expressing the blue sensor or red sensor. In all the cases, the cells were viable around 95%. **(B)** Flow cytometry–based viability test for red sensor–expressing U2OS cells stained with DAPI and blue sensor–expressing U2OS cells stained with PI. In both the cases the percentage of dead cells (DAPI-positive or PI-positive cells) doesn't exceed >2%, which is comparable to the wild-type U2OS cells. **(C)** U2OS cells stably expressing red sensor undergoes mitosis with cells imaged for different stages of the cell cycle (metaphase, anaphase and telophase). Binder-bound microtubules are shown in magenta, and DNA marked with DAPI is shown in cyan. Scale bars = 5 μm. **(D)** Histogram representing the microtubule growth rates (in blue) and depolymerization rates (in gray) obtained from the manual tracking of individual filaments in stable U2OS cells expressing the red tyrosination sensor. A total of 115 filaments analyzed from 15 cells. The growth rate (mean ± SD) and the depolymerization rate (mean ± SD) of microtubules are 0.32 ± 0.09 μm/s and 0.92 ± 0.18 μm/s respectively, in the cells. **(E)** Representative kymograph for a microtubule undergoing growth (polymerization), dynamic instability and catastrophe (depolymerization) events. The red star marks the rescue events followed by microtubule growth. The scale bar on the x axis = 2 μm and y axis = 50 s. **(F)** Microtubule dynamics from three different filaments (MT-1, 2, and 3 shown in green, blue, and red) from the stable U2OS cells. Respective paused, catastrophe, rescue, and growth phases of MT-1 (shown in green) are marked. **(G)** Catastrophe event frequency distribution per micrometer length of microtubule undergoing continuous catastrophe per minute, for a total of 130 filaments analyzed from 15 cells. **(H)** Quantification of total number of EB3-GFP comets per micrometer area per minute with and without A1aY1 binder. **(I)** Representative snapshots of movie frames from stable U2OS cells with the red tyrosination sensor (in grayscale), transiently expressing EB3-GFP (in green). The magenta and blue arrows for reference of typical EB3 comets at two selected microtubule plus ends. Scale bars = 5 μm, and time format (minutes:seconds) as indicated in Fig. 6 A. **(J)** Comparison of time-stack projections from TIRF movies of EB3 movement for control (without A1aY1 binder), with A1aY1 binder, and 0.5 μM SiR-tubulin + 10 μM Verapamil. Scale bar = 5 μm.

Video 1.  **Visualization of cellular microtubules and dynamics using the tyrosination sensor.** Scale bar = 5 µm, and time in minutes:seconds as indicated.

Video 2.  **Microtubule growth rate measurements using EB3-GFP (green) with tyrosination sensor (gray).** Scale bar = 5 µm, and time in minutes:seconds as indicated.

Video 3.  **Control (no drug) treated U2OS cells stably expressing red tyrosination sensor.** The yellow box indicates the region represented in Fig. 5 A. Scale bar = 5 µm, and time in minutes:seconds as indicated.

Video 4.  **Nocodazole-treated (10 µm) U2OS cells stably expressing red tyrosination sensor.** Scale bar = 5 µm, and time in minutes:seconds as indicated.

Video 5.  **Colchicine-treated (0.5 mM) U2OS cells stably expressing red tyrosination sensor.** Scale bar = 5 µm, and time in minutes:seconds as indicated.

Video 6.  **Vincristine-treated (1 µm) U2OS cells stably expressing red tyrosination sensor.** Scale bar = 5 µm, and time in minutes:seconds as indicated.

Video 7.  **3D-SIM reconstructions of interphase and mitotic phase of U2OS cells stably expressing red tyrosination sensor.**

