## [Peer Review File · The Journal of Cell Biology]

Genetically encoded live cell sensor for tyrosinated microtubules

Shubham Kesarwani, Prakash Lama, Anchal Chandra, P. Purushotam Reddy, Jijumon AS, Satish Bodakuntla, Balaji Rao, Carsten Janke, Ranabir Das, and Minhajuddin Sirajuddin

Corresponding Author(s): Minhajuddin Sirajuddin, Institute for Stem Cell Biology and Regenerative Medicine, NCBS-TIFR

Review Timeline:

Submission Date:	2019-12-19
Editorial Decision:	2020-01-23
Revision Received:	2020-04-16
Editorial Decision:	2020-06-04
Revision Received:	2020-07-06
Editorial Decision:	2020-07-14
Revision Received:	2020-07-17

Monitoring Editor: Arshad Desai

Scientific Editor: Andrea Marat

Transaction Report:

DOI: <https://doi.org/10.1083/jcb.201912107>

January 23, 2020

Re: JCB manuscript #201912107

Dr. Minhaj Sirajuddin
Institute for Stem Cell Biology and Regenerative Medicine, NCBS-TIFR
GKVK Post
Bellary Road
Bangalore 560065
India

Dear Minhaj:

Thank you for submitting your manuscript entitled "Genetically encoded live cell sensor for tyrosinated microtubules". The manuscript has been evaluated by 3 expert reviewers, whose reports are appended below. Unfortunately, after an assessment of the reviewer feedback, our editorial decision is against publication in JCB.

After considering the reviews, we are unfortunately unable to consider the work further in its present form. As you will see, the reviewers request substantial more effort to rigorously compare the sensor developed by your group to a prior one, to more carefully address the sensor's functionality/specificity in a cellular context, and to address potential interference of non-tubulin targets (primarily EB1). If you believe you and your colleagues can address the substantial reviewer comments and convince them that the sensor represents a significant improvement over the currently available one and will yield a widely useful tool, you are welcome to return with a new submission to the journal. However, we note that substantial new efforts are required and the outcomes must be persuasive with respect to the issues noted above. If timely publication is important, it may be best if you submitted the work elsewhere. The journal office will transfer your reviews upon request.

Given interest in the topic, I would be open to resubmission to JCB of a significantly revised and extended manuscript that fully addresses the reviewers' concerns and is subject to further peer-review. If you would like to resubmit this work to JCB, please contact the journal office to discuss an appeal of this decision or you may submit an appeal directly through our manuscript submission system. Please note that priority and novelty would be reassessed at resubmission.

Regardless of how you choose to proceed, we hope that the comments below will prove constructive as your work progresses. We would be happy to discuss the reviewer comments further once you've had a chance to consider the points raised in this letter. You can contact the journal office with any questions, cellbio@rockefeller.edu or call (212) 327-8588.

Thank you for thinking of JCB as an appropriate place to publish your work.

Sincerely,

Arshad Desai, PhD
Monitoring Editor

Reviewer #1 (Comments to the Authors (Required)):

Microtubules are subject to many post-translational modifications that can affect motor and MAP binding and possibly microtubule dynamics. A major stumbling block towards understanding the physiological functions of microtubule PTMs has been the lack of good live cell reporters for the various modifications. Many have talked about making such PTM specific sensors, but only one has been reported so far (Cassimieris group) and unfortunately that is not well-suited for live cell imaging. In this manuscript Kesarwani et. al. report a novel live cell sensor for tyrosinated tubulin that they obtained by screening for binders to three types of synthesized peptides (tyrosinated, detyrosinated and mono-glutamylated) using a combinatorial yeast display library employing the SSO7D protein scaffold. They concentrate on one of their good binders, tag it with two fluorescent tags successfully and then they investigate the effect of this sensor on some parameters of microtubule dynamics in cells as well as its effect on kinesin-1 and -3 in vitro. Lastly, they use this sensor to investigate with live-cell imaging the effects of tubulin binding drugs on microtubule dynamics in U2OS cells.

There would be a lot of interest in using such a sensor, but if the specificity is not well characterized, which I think is not the case with the current data presented in this study, it could lead to a lot of false results in cellular studies that will confuse more than illuminate, especially since this study is submitted under the "Tools" section. I do think more validation will need to be performed: (i) on the effects of this sensor on MT dynamics in vitro and a more complete characterization of its effects on MT dynamics in vivo; (ii) its concentration range for robust discrimination between tyr, detyr and glutamylated MTs; and (iii) comparison with the currently used anti-Tyr antibody in a cell type that displays many different types of modified microtubules to address its specificity and sensitivity. If these would be addressed, this would be a highly impactful study for the microtubule field.

My specific comments are below.

1) The binding affinity for tyr and mon-glutamylated peptide are 1.6 vs 13.6 microM. This is a pretty weak discrimination, which the authors themselves acknowledge, but it does raise several important issues regarding the use and reliability of this sensor.

The binding site is pretty open and the Tyr is not recognized in a deep pocket, so I can easily see that this surface will no longer discriminate between Tyr and deTYR in a cell type where there is a lot of deTyr as well as much glutamylation and the sensor concentration is higher.

What is the sensor concentration in cells? What is the concentration range in which it discriminates between tyr and detyr and between tyr and mono-glutamylated MTs? Does it work at the same concentration range in a cell type that has more mono-glutamylated MTs or in cells that express a controlled amount of a modification enzyme, for example? This is not at all addressed in their experiments with WT U2OS cells and cells that overexpress TTLL5 and 7. What is the

concentration of their sensor in all these experiments? different expression levels of the sensor would have to be tried with different levels of detyrosination and glutamylation to establish the dynamic range of this reporter

2) The authors should compare the readout of their sensor with that of anti-Tyr antibodies in a cell type that has many different types of modifications like a neuron. Does it discriminate there? How well does it overlap with the antibody which has been used in the field?

3) In figure 2 it would be nice to see a correlation coefficient between the A1aT1 tag and the SiR tubulin. Does this give you a readout of the % of total MT that are tyrosinated? In these cells it is clearly most of the MTs, but again, how does this look like in a cell type where MT are tyrosinated at lower levels. Can this sensor be used under those conditions? What is the minimal amount (or %) of Tyr tubulin needed for the A1aY1 to recognize tyr. tubulin in a cell?

4) The authors examine the effect of their sensor on microtubule dynamics in cells by tracking EB1 comets and reporting growth speeds. However, catastrophe is another important dynamic instability parameters that should be quantified and usually is in MT dynamics studies. Likewise for depolymerization rates.

5) in vitro microtubule dynamics assays with the sensor would go a long way towards establishing the effect of this sensor on microtubule dynamics and its concentration range where these are minimally perturbed.

6) Also, in vitro assays with detyrosinated and tyrosinated microtubules at different ratios should be used to establish the concentration ranges in which this binder is able to robustly discriminate between different microtubule subpopulations.

7) In figure 4D-G. The kymograph for kinesin in the presence of the +A1aY1 binder appears to more long pausing events. Were these analyzed? While motor velocity is not perturbed, run lengths should also be reported.

8) On page 9, the authors state: "Our live cell imaging with and without EB3-GFP comets also reveals that the tyrosination sensor binds to microtubules promptly during polymerization and disappears during catastrophe events".

This is a confusing statement. It seems to imply that the sensor no longer binds when the MT undergoes catastrophe? It sounds that way, but I suspect what the authors simply mean is that it is binding the microtubule and then it disappears when the microtubule disappears.

9) The motivation for the drug experiments is not clear - these could have easily been done with GFP-tubulin, so I do not see the power of this sensor for these types of experiments. Maybe the authors wanted to emphasize that now one could watch live the effect of drugs on different subpopulations of MTs in a cell that have different modifications?

10) In Figure 5 it looks like the MT remain close to what their initial intensity is in the first frame. Does this mean the binder is continuously being replenished? Would be nice to do a FRAP experiment on the MTs to look at this

Minor comments

1. The authors use odd phrasing throughout the manuscript, which impacts clarity. Also, there are many grammatical errors.
2. When citing the role of tyrosination in regulating motor transport together with the McKenney study, the Nirschl et al study should be included also
3. In the introduction the authors state that a major limitation of in vitro reconstitution experiments is that in vivo microtubules are modified with multiple different PTMs. That in itself is not a reason. One can make combinations of modifications in vitro also - so the fact that they exist in combination in vivo does not preclude their analyses through in vitro reconstitution
4. The authors mention the deleterious effect of GFP tagged alpha tubulin - they should give a reference here
5. Can the authors show an electrostatic surface for their binder? It looks like most of their interactions are electrostatic which would impact specificity.

Reviewer #2 (Comments to the Authors (Required)):

In the present work, Sirajuddin and colleagues used a yeast display library to develop a nanobody specific towards tyrosinated tubulin. Although there is a general lack of information about this nanobody (e.g. it wasn't clear to me whether this was a ScVF or a single domain/camelid antibody), it could be expressed in cells as fluorescent fusion proteins, and the authors performed several characterization and validation steps that confirmed specificity and non-perturbing nature of this new tool. All this convincingly built a case for a new genetically encoded live cell sensor for tyrosinated microtubules. Unfortunately, a similar reagent (an ScVF antibody) has already been developed and thoroughly characterized several years ago (Cassimeris et al., PLoS ONE, 2013) and although I agree that this tool has not yet been fully explored (possibly due to a much smaller community interested in tubulin PTMs at that time), I do not think that the current work offers a significant conceptual and/or technological advance to justify publication in the Journal of Cell Biology. In the present form, it is the opinion of this reviewer that the work is more suitable for a more specialized journal. Although the authors justify the need for such live-cell reporter, we didn't learn anything new with this tool, or saw any improvements/benefits relative to the previously published ScVF antibody. I think this would be required to meet the high publication standards expected for the Journal of Cell Biology. Moreover, I found the way that was used to present the case quite misleading since tyrosinated tubulin is also genetically encoded and thus this reagent does not really exclusively report on a true tubulin PTM. We therefore stand on the same position: there continues to exist no tool that can specifically report tubulin PTMs in live cells.

Reviewer #3 (Comments to the Authors (Required)):

Kesarwani et al. used a yeast display library to identify a specific nanobody against the C-terminal tyrosine residue of alpha-tubulin. They found one promising candidate (denoted A1aY1), whose binding properties was extensively characterized by structural and biochemical methods. In cells, A1aY1 was shown to specifically label tyrosinated microtubules without significantly affecting microtubule dynamics, EB3 comet formation and kinesin activity. One important outcome of the

study is that A1aY1 can also be used for live cell imaging. The potential of this possibility is demonstrated by live cell imaging of the effect of different microtubule-destabilizing agents in real time.

I have a few points that the authors need to address before I can recommend publication in JCB.

One of the main claims of the authors is that they for the first time developed a tubulin tyrosine sensor that can be used for live cell imaging. However, as mentioned by the authors in the Introduction section, Cassimeris et al. (PLOS One 8, e59812 (2013)) already published in 2013 a GFP-tagged, cytoplasmically expressed recombinant antibody (denoted 2G4-GFP), which also works in a live cell imaging setting. How do the properties of A1aY1 compare to 2G4-GFP? Is A1aY1 doing a better job and thus keeps indeed the promise of being the first useful tubulin nanobody that can be employed to follow tyrosinated microtubules in real time in live cells as claimed by the authors?

As stated by the authors in the Discussion section of their manuscript, EBs contain a very similar C-terminal tail sequence like alpha-tubulin including the C-terminal EEY/F motif. It is well established that the C-terminal tails of both alpha-tubulin and EBs are specifically recognized by the CAP-Gly domains of CLIP-170 and p150glued. How do the authors know that A1aY1 does not interfere with the binding of CLIP-170 and p150glued to EBs in cells? What is the affinity of A1aY1 for the C-terminal tail of EB? Note that CLIP-170 does also contain a C-terminal EEY/F motif that potentially could be recognized by A1aY1 and thus perturb the function of CLIP-170. Does A1aY1 interfere with the activity of tubulin tyrosine carboxy peptidases? I think that answering at least some of these questions is important for the author to claim that A1aY1 is a unique, robust and non-perturbing tool to investigate tyrosinated microtubules in cells.

Kesarwani et al. solved the complex structure between A1aY1 and a peptide derived from the C-terminal tail of alpha-tubulin by NMR. It would be useful for the interested reader if the authors would compare and contrast side by side the A1aY1-mediated interaction mode with the ones observed for CAP-Gly domains and the VASH-SVBP complex.

Several groups in the past studied the molecular mechanism of action of microtubule-destabilizing agents in living cells including the ones studied by Kesarwani et al. However, a comparison and discussion of the author's data with previous finding is not presented in their current manuscript.

Minor comments:

I was missing the mentioning of the commercially available and frequently used monoclonal IgG2a antibody YL1/2 in the Introduction section, which specifically recognizes tyrosinated tubulin and microtubules.

The authors should carefully check their reference list. I feel that they missed to cite several relevant papers to give a balanced and adequate view of some of the discussed topics.

Summary of Reviewer response and revision:

We thank the reviewers, in particular reviewer 1 and reviewer 3 for taking valuable time in providing constructive feedback. A common concern raised by the reviewers is that a 2013 study (Cassimieris et al 2013) has reported ScFv fragment (2G4), which can label tyrosinated microtubules and therefore the tyrosination sensor presented here is not unique. In our revision, we tested the 2G4 using the same specificity experiments performed with our tyrosination sensor. Unlike our tyrosination sensor, we found that 2G4C in addition to tyrosinated microtubules, labels detyrosinated and polyglutamylated microtubules inside the cells (Supplement Figure 8). Suggesting that 2G4C non-specifically recognizes microtubules and the A1aY1 (SSO7D binder) is the only valid live cell tyrosination sensor or nanobody currently available. In our revision, we have also tested the dynamic range of tyrosination sensor (Figure 2A and 2B), comparative staining of anti-tyrosination tubulin antibody with tyrosination sensor in U2OS and H9C2 cells (Supplement Figure 3), plus-end staining of CLIP170 and expanded the motor analysis (Figure 4C and Supplement Figure 11). Together, we have comprehensively addressed the reviewers' comments that have strengthened our manuscript and conclusions. We thank again the reviewers for suggestions in improving our manuscript.

Reviewer #1 (Comments to the Authors (Required)):

Microtubules are subject to many post-translational modifications that can affect motor and MAP binding and possibly microtubule dynamics. A major stumbling block towards understanding the physiological functions of microtubule PTMs has been the lack of good live cell reporters for the various modifications. Many have talked about making such PTM specific sensors, but only one has been reported so far (Cassimieris group) and unfortunately that is not well-suited for live cell imaging. In this manuscript Kesarwani et. al. report a novel live cell sensor for tyrosinated tubulin that they obtained by screening for binders to three types of synthesized peptides (tyrosinated, detyrosinated and mono-glutamylated) using a combinatorial yeast display library employing the SSO7D protein scaffold. They concentrate on of their good binders, tag it with two fluorescent tags successfully and then they investigate the effect of this sensor on some parameters of microtubule dynamics in cells as well as its effect on kinesin-1 and -3 in vitro. Lastly, they use this sensor to investigate with live-cell imaging the effects of tubulin binding drugs on microtubule dynamics in U2OS cells.

There would be a lot of interest in using such a sensor, but if the specificity is not well characterized, which I think is not the case with the current data presented in this study, it could lead to a lot of false results in cellular studies that will confuse more than illuminate, especially since this study is submitted under the "Tools" section. I do think

more validation will need to be performed: (i) on the effects of this sensor on MT dynamics in vitro and a more complete characterization of its effects on MT dynamics in vivo; (ii) its concentration range for robust discrimination between tyr, detyr and glutamylated MTs; and (iii) comparison with the currently used anti-Tyr antibody in a cell type that displays many different types of modified microtubules to address its specificity and sensitivity. If these would be addressed, this would be a highly impactful study for the microtubule field.

My specific comments are below.

1) The binding affinity for tyr and mon-glutamylated peptide are 1.6 vs 13.6 microM This is a pretty weak discrimination, which the authors themselves acknowledge, but it does raise several important issues regarding the use and reliability of this sensor.

The binding site is pretty open and the Tyr is not recognized in a deep pocket, so I can easily see that this surface will no longer discriminate between Tyr and deTYR in a cell type where there is a lot of deTyr as well as much glutamylation and the sensor concentration is higher.

What is the is the sensor concentration in cells? What Is the concentration range in which it discriminates between tyr and detyr and between tyr and mono-glutamylated MTs? Does it work at the same concentration range in a cell type that has more mono-glutamylated MTs or in cells that express a controlled amount of a modification enzyme, for example? This is not at all addressed in their experiments with WT U2OS cells and cells that overexpress TTLL5 and 7. What is the concentration of their sensor in all these experiments? different expression levels of the sensor would have to be tried with different levels of detyrosination and glutamylation to establish the dynamic range of this reporter

We appreciate the concern raised here and the opportunity to clarify our observed binding affinity vs specificity claims. The A1aY1 sensor affinity between Tyr Vs deTyr are 1.6 and >60 microM respectively, this is a significant difference which is also exemplified in our cell specificity experiments (Figure 3A and 3B data).

As pointed out the difference between Tyr and mono-glu peptide is ~13-fold, which may not seem much of a difference, but in the cell experiments (Figure 3A and 3B) we have used TTLL5, a polyglutamylation enzyme. The specificity experiment in Figure 3A and its quantification in 3B shows that A1aY1 binder does not bind to polyglutamylated microtubules (alpha-tubulin). We believe that the affinity difference between Tyr and poly-glu alpha tubulin will be much more than that observed with the mono-glu peptide. We could not perform biochemical experiments with poly-glu because the commercial

peptide synthesis companies do not provide that services and we therefore restricted our biochemical titration experiments with mono-glu peptide.

Regarding the concentration of sensor in the cells, as with any ectopic expression it is difficult to quantify the concentration of expressed proteins inside cells. We have however segregated our stable cell lines as low, medium and high expression of A1aY1 sensor in cells (Supplement Fig 4). We have mentioned this in our results section (line 222 - 226).

2) The authors should compare the readout of their sensor with that of anti-Tyr antibodies in a cell type that has many different types of modifications like a neuron. Does it discriminate there? How well does it overlap with the antibody which has been used in the field?

This is a valid point, first we have addressed this concern using *in vitro* assays with varying levels of tyrosination/detyrosination tubulin (see Figure 2A and 2B and reviewer #1 point 6). Our new results suggest there is a linear gradation in detection of tyrosination tubulin levels in mixed microtubules. Secondly, we have stained U2OS and H9C2 cells with commercial anti-Tyr along with anti-detyrosination antibodies and have compared with our tyrosination sensor (Supplement Figure 3). In H9C2 cells, which have higher levels of detyrosination, we have observed that microtubules that have higher levels of detyrosination are devoid of our tyrosination sensor. Further exemplifying the applicability of the sensor presented in this study.

3) In figure 2 it would be nice to see a correlation coefficient between the A1aT1 tag and the SiR tubulin. Does this give you a readout of the % of total MT that are tyrosinated? In these cells it is clearly most of the MTs ,but again, how does this look like in a cell type where MT are tyrosinated at lower levels. Can this sensor be used under those conditions? What is the minimal amount (or %) of Tyr tubulin needed for the A1aY1 to recognize tyr. tubulin in a cell?

Correlation coefficient between A1aY1 and SiR-tubulin have been shown in Supplement Figure 5 (for Figure 2C cell images) and Figure 6B as co-localization color map and Manders coefficient for 3D-SIM images (Fig 6B). Similarly, we have quantified this colocalization in specificity experiments represented in Figure 3B.

To address the question of how does tyrosination sensor stains in cells which have lower levels of tyrosinated MT, we have used H9C2 cells (see Supplement Figure 3 and above comment).

4) The authors examine the effect of their sensor on microtubule dynamics in cells by tracking EB1 comets and reporting growth speeds. However, catastrophe is another

important dynamic instability parameters that should be quantified and usually is in MT dynamics studies. Likewise for depolymerization rates.

The catastrophe, depolymerization and rescue rates have been quantified as dynamic instability and shown in Figure 5. We have also counted the total number of EB3 comets with and without A1aY1 binder and found no significant difference (Supplement Figure 10). Additionally, we have stained the microtubule plus-ends using CLIP-170 antibody and found no difference between cells with and without A1aY1 binder (Figure 4C). For CLIP170 staining experiment we have used VASH2+SVBP (detyrosinase) enzyme complex as a control, and found that cells with detyrosinated microtubules have reduced CLIP170 at the microtubule plus-ends.

5) in vitro microtubule dynamics assays with the sensor would go a long way towards establishing the effect of this sensor on microtubule dynamics and its concentration range where these are minimally perturbed.

We disagree with this comment, simply because our goal is to establish the robustness of intracellular tyrosinated microtubule sensor. Additionally, there are already great many tools to perform in vitro microtubule dynamics assay (cell free) with purified components. Our experiments with EB3 (see reviewer #1 comment no. 4 and no. 6, and reviewer#3 CLIP170 comment) will establish the non-interfering nature and the dynamic range of A1aY1 sensor inside cells.

6) Also, in vitro assays with detyrosinated and tyrosinated microtubules at different ratios should be used to establish the concentration ranges in which this binder is able to robustly discriminate between different microtubule subpopulations.

We have addressed this concern by generating pure tyrosinated tubulin (Hela tubulin) and spike them with different ratios of detyrosinated tubulin (CPA treated tubulin) (Supplement Figure 11D) and checked the A1aY1 binding using *in vitro* TIRF experiments. Quantification of binding to differential levels of detyrosination shows a linear gradation in A1aY1 binding corresponding to the levels of tyrosination levels. We have added this data and description in revised manuscript (Figure 2A and 2B).

7) In figure 4D-G. The kymograph for kinesin in the presence of the +A1aY1 binder appears to more long pausing events. Were these analyzed? While motor velocity is not perturbed, run lengths should also be reported.

Our kymograph analysis does not show any difference in the pausing events (see Supplement Figure 11). In order to address this, we have analyzed the run lengths from the motility data and found no significant effects due to A1aY1 binder (Figure 4D and 4E).

8) On page 9, the authors state: "Our live cell imaging with and without EB3-GFP comets also reveals that the tyrosination sensor binds to microtubules promptly during polymerization and disappears during catastrophe events".

This is a confusing statement. It seems to imply that the sensor no longer binds when the MT undergoes catastrophe? It sounds that way, but I suspect what the authors simply mean is that it is binding the microtubule and then it disappears when the microtubule disappears.

We thank the reviewer for bringing this to our attention, it is indeed as the reviewer has pointed out that the A1aY1 signal disappears when the microtubule depolymerizes. We have rephrased this sentence appropriately (see line 295).

"..Our live cell imaging with and without EB3-GFP comets also reveals that the tyrosination sensor signal disappears promptly during microtubule depolymerization events.."

9) The motivation for the drug experiments is not clear - these could have easily been done with GFP-tubulin, so I do not see the power of this sensor for these types of experiments. Maybe the authors wanted to emphasize that now one could watch live the effect of drugs on different subpopulations of MTs in a cell that have different modifications?

We again thank the reviewer for pointing out the right phrasing for our motivations towards these experiments. We have incorporated this in the main text of revised manuscript (see line 321-323).

10) In Figure 5 it looks like the MT remain close to what their initial intensity is in the first frame. Does this mean the binder is continuously being replenished? Would be nice to do a FRAP experiment on the MTs to look at this

We originally intended to perform a FRAP experiment to determine the recovery rates of A1aY1 sensor binding to microtubules. However, due to the COVID-19 pandemic situation our institution and city is under lockdown, under these circumstances we could not perform this experiment. Indeed, as this reviewer has pointed out, the binder is continuously being replenished this is perhaps due to the micromolar affinity of A1aY1 binders towards alpha tubulin CTT. Thus, the tyrosination sensor can be continuously imaged over long periods of time without significant photobleaching. For example, see Supplement movie 1; 30 minutes, Supplement Movie 4, 50 minutes and Supplement movie 5, 25 minutes durations.

Minor comments

1. *The authors use odd phrasing throughout the manuscript, which impacts clarity. Also, there are many grammatical errors.*

We thank the reviewer for pointing out these errors, we have modified the text with more clear phrasings.

2. *When citing the role of tyrosination in regulating motor transport together with the McKenney study, the Nirschl et al study should be included also*

The missing citation has been added in the introduction (line 67).

3. *In the introduction the authors state that a major limitation of in vitro reconstitution experiments is that in vivo microtubules are modified with multiple different PTMs. That in itself is not a reason. One can make combinations of modifications in vitro also - so the fact that they exist in combination in vivo does not preclude their analyses through in vitro reconstitution.*

We have changed the wordings to “..However, a shortcoming of majority of the in vitro reconstitution experiments with homogenous modified microtubules is that they may not reflect the in vivo scenario, since microtubules inside cells can possess multiple PTMs at the same time..” (line 72).

4. *The authors mention the deleterious effect of GFP tagged alpha tubulin - they should give a reference here*

Appropriate reference has been added in line 96-97.

5. *Can the authors show an electrostatic surface for their binder? It looks like most of their interactions are electrostatic which would impact specificity.*

We have added a figure showing color coded based on the residues of A1aY1 binder surface interacting with alpha tubulin CTT peptide (Supplement Figure 2E).

We thank the reviewer for pointing out the missing citations and suggestions in improving the text, we have incorporated these changes appropriately in the revised manuscript.

Reviewer #2 (Comments to the Authors (Required)):

In the present work, Sirajuddin and colleagues used a yeast display library to develop a nanobody specific towards tyrosinated tubulin. Although there is a general lack of information about this nanobody (e.g. it wasn't clear to me whether this was a ScVF or a single domain/camelid antibody), it could be expressed in cells as fluorescent fusion proteins, and the authors performed several characterization and validation steps that confirmed specificity and non-perturbing nature of this new tool. All this convincingly built a case for a new genetically encoded live cell sensor for tyrosinated microtubules. Unfortunately, a similar reagent (an ScVF antibody) has already been developed and thoroughly characterized several years ago (Cassimeris et al., PLoS ONE, 2013) and although I agree that this tool has not yet been fully explored (possibly due to a much smaller community interested in tubulin PTMs at that time), I do not think that the current work offers a significant conceptual and/or technological advance to justify publication in the Journal of Cell Biology. In the present form, it is the opinion of this reviewer that the work is more suitable for a more specialized journal. Although the authors justify the need for such live-cell reporter, we didn't learn anything new with this tool, or saw any improvements/benefits relative to the previously published ScVF antibody. I think this would be required to meet the high publication standards expected for the Journal of Cell Biology. Moreover, I found the way that was used to present the case quite misleading since tyrosinated tubulin is also genetically encoded and thus this reagent does not really exclusively report on a true tubulin PTM. We therefore stand on the same position: there continues to exist no tool that can specifically report tubulin PTMs in live cells.

The major criticism from reviewer#2 is the lack of comparison with the existing ScFV (2G4-GFP) reagent. While there is no proof or citation in microtubule literature of application of 2G4-GFP reagent, except from the original report Cassimeris et al., PLoS ONE, 2013. We have addressed this concern by comparing our A1aY1 to 2G4 in live cell experiments, which conclusively shows that our tyrosination sensor is specific towards tyrosinated microtubules, but not the 2G4 ScFv. For more details see summary above and Supplement Figure 8.

As this reviewer points out that there has been little information provided about the nanobody scaffold. We have included this information in the Methods section in detail in the revised manuscript.

We also would like to point out to this reviewer that indeed the terminal tyrosine is genetically encoded, however there are tubulin PTM enzymes to remove and add terminal tyrosine. Perhaps this cycle of tyrosination/detyrosination may not occur robustly in epithelial or fibroblast cell culture systems widely used in cell biology. But

tyrosination/detyrosination cycle has been reported and important for microtubule function in cardiomyocytes, neurons and other terminally differentiated cell types. Moreover, the knockout mice lacking TTL (the enzyme that reinstates the terminal tyrosine) are embryonically lethal, further providing a strong evidence that tyrosination modification is a *bona fide* tubulin PTM site and therefore our tyrosination sensor represents the first tubulin PTM sensor or nanobody.

Reviewer #3 (Comments to the Authors (Required)):

Kesarwani et al. used a yeast display library to identify a specific nanobody against the C-terminal tyrosine residue of alpha-tubulin. They found one promising candidate (denoted A1aY1), whose binding properties was extensively characterized by structural and biochemical methods. In cells, A1aY1 was shown to specifically label tyrosinated microtubules without significantly affecting microtubule dynamics, EB3 comet formation and kinesin activity. One important outcome of the study is that A1aY1 can also be used for live cell imaging. The potential of this possibility is demonstrated by live cell imaging of the effect of different microtubule-destabilizing agents in real time.

I have a few points that the authors need to address before I can recommend publication in JCB.

One of the main claims of the authors is that they for the first time developed a tubulin tyrosine sensor that can be used for live cell imaging. However, as mentioned by the authors in the Introduction section, Cassimeris et al. (PLOS One 8, e59812 (2013)) already published in 2013 a GFP-tagged, cytoplasmically expressed recombinant antibody (denoted 2G4-GFP), which also works in a live cell imaging setting. How do the properties of A1aY1 compare to 2G4-GFP? Is A1aY1 doing a better job and thus keeps indeed the promise of being the first useful tubulin nanobody that can be employed to follow tyrosinated microtubules in real time in live cells as claimed by the authors?

We agree with this comment that the 2G4-GFP needs to be compared with our A1aY1, therefore we have performed the cell specificity experiment with 2G4, similar to Figure 3. The surprising outcome is that 2G4 remains bound to microtubules upon detyrosination and polyglutamylation with PTM enzymes (Supplement Figure 8). This could be due to non-specific binding, thus addressing the concerns regarding robustness of our A1aY1 sensor against existing reagents. Our tyrosination sensor is specific and because of its micromolar affinity towards alpha-tubulin CTT with terminal tyrosine, it does not interfere with normal microtubule function.

As stated by the authors in the Discussion section of their manuscript, EBs contain a very similar C-terminal tail sequence like alpha-tubulin including the C-terminal EEY/F

motif. It is well established that the C-terminal tails of both alpha-tubulin and EBs are specifically recognized by the CAP-Gly domains of CLIP-170 and p150glued. How do the authors know that A1aY1 does not interfere with the binding of CLIP-170 and p150glued to EBs in cells? What is the affinity of A1aY1 for the C-terminal tail of EB? Note that CLIP-170 does also contain a C-terminal EEY/F motif that potentially could be recognized by A1aY1 and thus perturb the function of CLIP-170. Does A1aY1 interfere with the activity of tubulin tyrosine carboxy peptidases? I think that answering at least some of these questions is important for the author to claim that A1aY1 is a unique, robust and non-perturbing tool to investigate tyrosinated microtubules in cells.

We thank the reviewer for raising this point. The concerns here are that the EBs have similar C-terminal tail sequence as alpha-tubulin (EEY/F, a tripeptide motif). A unique feature of our A1aY1 is its ability recognize up to 7 residues of alpha tubulin carboxy-terminal tail (see Fig 1C, Fig 3C, 3D, 3E and supplement figure 12). This is different from CAP-Gly protein which recognize only the up to 5 residues. Our claims are supported by the biochemical and cell specificity experiments, which show that the 7th residue of alpha tubulin carboxy-terminal tail, a poly-glutamylation site is crucial (See Fig 1B and 3A and 3B). Therefore, we think that a tri-peptide of EEY/F will have poor affinity towards our A1aY1 and therefore will not interfere with CAP-Gly of CLIP170 and p150glued. To clarify this, we have added structural comparison of CAP-Gly and VASH-SVBP structures with alpha-tubulin carboxy terminal tails as suggested by this reviewer (Supplement Figure 12). We have also extensively discussed in these points in the discussion section (see line 414 – 436).

To further strengthen this argument, we have compared the CLIP170 localization to the microtubule plus-ends and found no differences between the cells with and without binder (Figure 4C).

Kesarwani et al. solved the complex structure between A1aY1 and a peptide derived from the C-terminal tail of alpha-tubulin by NMR. It would be useful for the interested reader if the authors would compare and contrast side by side the A1aY1-mediated interaction mode with the ones observed for CAP-Gly domains and the VASH-SVBP complex.

We thank the reviewer for this suggestion, we have compared the structures and added this in Supplement Figure 12 in the revised manuscript.

Several groups in the past studied the molecular mechanism of action of microtubule-destabilizing agents in living cells including the ones studied by Kesarwani et al. However, a comparison and discussion of the author's data with previous finding is not presented in their current manuscript.

We have expanded the discussion regarding the microtubule destabilizing drugs and have presented our observations in the light of previous findings. Please see discussion section from line 438 – 469.

Minor comments:

I was missing the mentioning of the commercially available and frequently used monoclonal IgG2a antibody YL1/2 in the Introduction section, which specifically recognizes tyrosinated tubulin and microtubules.

Appropriate citations for widely used antibodies have been added in the introduction (line 84-85). Additionally, the antibodies used in this study have been mentioned in the detailed methodology section.

The authors should carefully check their reference list. I feel that they missed to cite several relevant papers to give a balanced and adequate view of some of the discussed topics.

We thank the reviewer for pointing out the missing citations, we have incorporated these changes appropriately in the revised manuscript.

June 4, 2020

Re: JCB manuscript #201912107R-A

Dr. Minhajuddin Sirajuddin
Institute for Stem Cell Biology and Regenerative Medicine, NCBS-TIFR
GKVK Post
Bellary Road
Bangalore 560065
India

Dear Dr. Sirajuddin,

Thank you for submitting your revised manuscript entitled "Genetically encoded live cell sensor for tyrosinated microtubules".

Please find enclosed feedback from two of the original reviewers on your revised manuscript. Reviewer 3 supports publication; however, Reviewer 1 reiterates concerns about expression level and specificity (the Rev also stresses analysis of in vitro microtubule dynamics; while we agree with the reviewer that this is very important, as we had indicated it was not essential in our prior discussion regarding your revision plan we will not require this point to be experimentally addressed). Please note that the comments from the reviewer are targeted at promoting the utility of your tool and need to be taken seriously. While we typically only consider one round of revision, we are willing to give you the opportunity to experimentally address the reviewer's comments on specificity and expression as well as address through text changes the remaining comments.

Please submit the final revision, along with a cover letter that includes a point by point response to the remaining reviewer comments. Given current lab closures, please advise the editorial office as to your anticipated timeline.

Thank you for this interesting contribution to Journal of Cell Biology. You can contact me or the scientific editor listed below at the journal office with any questions, cellbio@rockefeller.edu or call (212) 327-8588.

Sincerely,

Arshad Desai, PhD
Monitoring Editor

Andrea L. Marat, PhD
Scientific Editor

Journal of Cell Biology

Reviewer #1 (Comments to the Authors (Required)):

While the authors have addressed how their sensor compares with the published old one from the Cassimeris group, the authors failed to address several important issues that were raised by the initial review. While I understand that these are difficult times, I think minimizing the importance of doing some of these additional validation experiments is problematic since the authors are trying to present here a tool that will be reliably used by the cell biology community without the danger of artifacts. I still remain enthusiastic about this manuscript, but cannot recommend publication until these questions are addressed

(1) The effect of the sensor on microtubule dynamics in cells

- It is not at all clear what the authors report in Figure 5E - What is the parameter called "dynamic instability" shown as a violet bar? Microtubule dynamics are characterized by rate of growth, catastrophe, rate of depolymerization, rescue

At the very least they should report rate of growth, catastrophe in cells that express low and medium levels of the reporter. It is even clear from visual inspection that the high-expressing cells have altered MT network appearance, which is why doing rigorous, quantitative measurements of the effect of this sensor on MT dynamics is important

As in my original review, it is important to know at what levels this reporter is expressed in their cells - μM , nM, what is its stoichiometry to tubulin?

Having transient ectopic expression does not preclude one from getting an idea of the overall expression levels in cells as the authors state in their rebuttal- one can FACS sort for example and then do a Western.

(2) Following up on point (1) and the comment that I made in my original review - the effect of this sensor should be tested on MT dynamics in vitro. This is completely within the realm of the doable and it will give an idea of the concentration ranges that do not affect microtubule dynamics. The EB3 experiments that the authors refer to do not address this question - just seeing similar number of EB3 comets or CLP170 at the tips in the presence of the sensor is a pretty gross assessment of MT dynamics.

(3) I am still concerned by the potential crossreactivity with mono-glutamylated MTs. There is less than an order of magnitude difference between the Kds of the tyrosinated and monoglutamylated peptides, which raises the question of how specific the sensor will be in vivo. Yes, the authors shows that it is able to discriminate against polyglutamylated tubulin when they overexpress TLL5, but that is not the concern that I raised in my first review. Given the not to impressive difference between the kd for the tyrosinated and mono-glutamylated peptide, can they show robust discrimination against monoglutamylated MTs? Why not overexpress a TLL that is known to monoglutamylate?

(4) The authors make a statement in the revised ms that still does not make sense and/or is not accurate "...However, a shortcoming of majority of the in vitro reconstitution experiments with homogenous modified microtubules is that they may not reflect the in vivo scenario, since microtubules inside cells can possess multiple PTMs at the same time."

One can make combinations of modifications on the same MT in vitro also - so the fact that they exist in combination in vivo does not preclude their analyses through in vitro reconstitution. This is certainly not a limitation of the in vitro experiments. There are others, but this is not one of them.

(5) The authors should strive to take a more scholarly approach in their citations. Several seminal studies into the turnover of modified MTs (Webster et al 1990 from the Borisy group; Schulze et al

1987 from the Kirschner group) are not cited when mentioning stability of modified MTs. When mentioning tubulin engineering efforts, a more balanced citation should be given to include Minoura et al. 2013 and Vemu et al. 2016. The authors also state in their introduction that SiR-tubulin affects microtubule dynamics - citation for this should be given (the original report does not show an effect on MT dynamics and shows undisturbed mitotic index)

The authors should check their ms for grammatical errors like the one below;

Line 214: "So far, our biochemical and structural experiments with A1aY1 binder were purified with to purified binder"

Reviewer #3 (Comments to the Authors (Required)):

I am satisfied with how the authors addressed my points and am happy to support publication in JCB.

Reviewer #1 (Comments to the Authors (Required)):

While the authors have addressed how their sensor compares with the published old one from the Cassimeris group, the authors failed to address several important issues that were raised by the initial review. While I understand that these are difficult times, I think minimizing the importance of doing some of these additional validation experiments is problematic since the authors are trying to present here a tool that will be reliably used by the cell biology community without the danger of artifacts. I still remain enthusiastic about this manuscript, but cannot recommend publication until these questions are addressed

We thank the reviewer for taking time in providing valuable comments. We have addressed all the comments in the best possible way under the current circumstances. We hope the revised manuscript clarifies the queries and strengthens our manuscript for publication.

(1) The effect of the sensor on microtubule dynamics in cells

- It is not at all clear what the authors report in Figure 5E - What is the parameter called "dynamic instability" shown as a violet bar? Microtubule dynamics are characterized by rate of growth, catastrophe, rate of depolymerization, rescue

At the very least they should report rate of growth, catastrophe in cells that express low and medium levels of the reporter. It is even clear from visual inspection that the high-expressing cells have altered MT network appearance, which is why doing rigorous, quantitative measurements of the effect of this sensor on MT dynamics is important

We thank the reviewer for pointing out the confusion in 5E, we have clarified in the figure legend (line 731-737). Here we have defined dynamic instability events (violet bar) as the number of microtubules that undergo cycles of polymerization and depolymerization in given time frame. For polymerization and depolymerization events, we have counted the number of microtubules that undergo only either polymerization or depolymerization events in the given time frame. In the case of drug treated cells we predominantly observe microtubules that are undergoing depolymerization events in the analyzed time frame.

In the revised manuscript we have measured the polymerization and depolymerization rates (Supplement figure 10A) and have shown several examples of microtubule dynamics (Supplement Figure 10B, 10C and 10D). The analysis has revealed polymerization rates similar to previous reports and have updated the text (see line 270-272). Together with our EB3 quantification and microtubule dynamics analysis we conclude that A1aY1 sensor does not interfere with microtubule dynamics.

As in my original review, it is important to know at what levels this reporter is expressed in their cells - microM, nM, what is its stoichiometry to tubulin?

Having transient ectopic expression does not preclude one from getting an idea of the overall expression levels in cells as the authors state in their rebuttal- one can FACS sort for example and then do a Western.

As suggested, to find out the concentration of sensor we have performed FACS and western blot of low/medium and high expression of cells and further titrated with known concentrations of purified tubulin and recombinant A1aY1 binder. From this we conclude that the A1aY1 sensor is expressed roughly in the same concentration range of tubulin (micromolar range) inside cells. We have added this data in Supplement Figure 6 and described in main text (line 203-205) and methods.

(2) Following up on point (1) and the comment that I made in my original review - the effect of this sensor should be tested on MT dynamics in vitro. This is completely within the realm of the doable and it will give an idea of the concentration ranges that do not affect microtubule dynamics. The EB3 experiments that the authors refer to do not address this question - just seeing similar number of EB3 comets or CLP170 at the tips in the presence of the sensor is a pretty gross assessment of MT dynamics.

We politely disagree with this comment, our goal here is to describe the invention of live cell sensor, of course this can be applied to in in-vitro studies too, however it is beyond the scope of this manuscript. We have also

extensively characterized microtubule dynamics inside cells using our tyrosination sensor and found to be non-perturbing (see comment 1).

(3) I am still concerned by the potential crossreactivity with mono-glutamylated MTs. There is less than an order of magnitude difference between the Kds of the tyrosinated and monoglutamylated peptides, which raises the question of how specific the sensor will be in vivo. Yes, the authors shows that it is able to discriminate against polyglutamylated tubulin when they overexpress TTLL5, but that is not the concern that I raised in my first review. Given the not to impressive difference between the kd for the tyrosinated and mono-glutamylated peptide, can they show robust discrimination against monoglutamylated MTs? Why not overexpress a TTLL that is known to monoglutamylate?

We understand the concern raised here. In other sensor examples we have observed that an order of magnitude differences in Kd is sufficient to disrupt binding between two proteins in cells. As suggested, to further strengthen our claim, we have performed similar specificity experiments with TTLL4 and found to perturb A1aY1 binding to microtubules. Therefore, the A1aY1 binder can robustly discriminate between non-glutamylated versus glutamylated alpha-tubulin CTT.

(4) The authors make a statement in the revised ms that still does not make sense and/or is not accurate "...However, a shortcoming of majority of the in vitro reconstitution experiments with homogenous modified microtubules is that they may not reflect the in vivo scenario, since microtubules inside cells can possess multiple PTMs at the same time."

One can make combinations of modifications on the same MT in vitro also - so the fact that they exist in combination in vivo does not preclude their analyses through in vitro reconstitution. This is certainly not a limitation of the in vitro experiments. There are others, but this is not one of them.

We thank the reviewer to revisit this statement in the introduction and have rephrased to avoid any confusion. Please see line 65-68.

(5) The authors should strive to take a more scholarly approach in their citations. Several seminal studies into the turnover of modified MTs (Webster et al 1990 from the Borisy group; Schulze et al 1987 from the Kirschner group) are not cited when mentioning stability of modified MTs. When mentioning tubulin engineering efforts, a more balanced citation should be given to include Minoura et al. 2013 and Vemu et al. 2016. The authors also state in their introduction that SiR-tubulin affects microtubule dynamics - citation for this should be given (the original report does not show an effect on MT dynamics and shows undisturbed mitotic index)

We thank the reviewer for pointing out the missing references, we have updated the references, please see line 57 and line 70 for updated citations.

Regarding the SiR-tubulin, in the original report it has been shown that at higher concentrations MT growth is affected. Therefore, we modified the sentence accordingly and have cited the original report (see line 91).

The authors should check their ms for grammatical errors like the one below;
Line 214: "So far, our biochemical and structural experiments with A1aY1 binder were purified with to purified binder"

We thank the reviewer for pointing out this error, we have updated in the revised manuscript. See line 198.

July 14, 2020

RE: JCB Manuscript #201912107RR

Dr. Minhajuddin Sirajuddin
Institute for Stem Cell Biology and Regenerative Medicine, NCBS-TIFR
GKVK Post
Bellary Road
Bangalore 560065
India

Dear Dr. Sirajuddin:

Thank you for submitting your revised manuscript entitled "Genetically encoded live cell sensor for tyrosinated microtubules". We would be happy to publish your paper in JCB pending final revisions necessary to meet our formatting guidelines (see details below).

A. MANUSCRIPT ORGANIZATION AND FORMATTING:

Full guidelines are available on our Instructions for Authors page, <http://jcb.rupress.org/submission-guidelines#revised>. **Submission of a paper that does not conform to JCB guidelines will delay the acceptance of your manuscript.**

1) Text limits: Character count for Tools is < 40,000, not including spaces. Count includes title page, abstract, introduction, results, discussion, acknowledgments, and figure legends. Count does not include materials and methods, references, tables, or supplemental legends.

2) Figures limits: Tools may have up to 10 main text figures.

3) Figure formatting: Scale bars must be present on all microscopy images, including inset magnifications. Molecular weight or nucleic acid size markers must be included on all gel electrophoresis.

4) Statistical analysis: Error bars on graphic representations of numerical data must be clearly described in the figure legend. The number of independent data points (n) represented in a graph must be indicated in the legend. Statistical methods should be explained in full in the materials and methods. For figures presenting pooled data the statistical measure should be defined in the figure legends. Please also be sure to indicate the statistical tests used in each of your experiments (either in the figure legend itself or in a separate methods section) as well as the parameters of the test (for example, if you ran a t-test, please indicate if it was one- or two-sided, etc.). Also, if you used parametric tests, please indicate if the data distribution was tested for normality (and if so, how). If not, you must state something to the effect that "Data distribution was assumed to be normal but this was not formally tested."

- 5) Abstract and title: The abstract should be no longer than 160 words and should communicate the significance of the paper for a general audience. The title should be less than 100 characters including spaces. Make the title concise but accessible to a general readership.
- 6) Materials and methods: Should be comprehensive and not simply reference a previous publication for details on how an experiment was performed. Please provide full descriptions in the text for readers who may not have access to referenced manuscripts.
- 7) Please be sure to provide the sequences for all of your primers/oligos and RNAi constructs in the materials and methods. You must also indicate in the methods the source, species, and catalog numbers (where appropriate) for all of your antibodies. Please also indicate the acquisition and quantification methods for immunoblotting/western blots.
- 8) Microscope image acquisition: The following information must be provided about the acquisition and processing of images:
- Make and model of microscope
 - Type, magnification, and numerical aperture of the objective lenses
 - Temperature
 - Imaging medium
 - Fluorochromes
 - Camera make and model
 - Acquisition software
 - Any software used for image processing subsequent to data acquisition. Please include details and types of operations involved (e.g., type of deconvolution, 3D reconstitutions, surface or volume rendering, gamma adjustments, etc.).
- 9) References: There is no limit to the number of references cited in a manuscript. References should be cited parenthetically in the text by author and year of publication. Abbreviate the names of journals according to PubMed. * Please note supplemental references are not permitted.
- 10) * Supplemental materials: There are strict limits on the allowable amount of supplemental data. Tools may have up to 5 supplemental display items (figures and tables), be sure to correct the callouts in the text when reducing the SI figures. Please also note that tables, like figures, should be provided as individual, editable files. A summary of all supplemental material should appear at the end of the Materials and methods section.
- 11) eTOC summary: A ~40-50-word summary that describes the context and significance of the findings for a general readership should be included on the title page. The statement should be written in the present tense and refer to the work in the third person.
- 12) Conflict of interest statement: JCB requires inclusion of a statement in the acknowledgements regarding competing financial interests. If no competing financial interests exist, please include the following statement: "The authors declare no competing financial interests." If competing interests are declared, please follow your statement of these competing interests with the following statement: "The authors declare no further competing financial interests."
- 13) ORCID IDs: ORCID IDs are unique identifiers allowing researchers to create a record of their various scholarly contributions in a single place. At resubmission of your final files, please consider providing an ORCID ID for as many contributing authors as possible.

B. FINAL FILES:

-- High-resolution figure and video files: See our detailed guidelines for preparing your production-ready images, <http://jcb.rupress.org/fig-vid-guidelines>.

****It is JCB policy that if requested, original data images must be made available to the editors. Failure to provide original images upon request will result in unavoidable delays in publication. Please ensure that you have access to all original data images prior to final submission.****

****The license to publish form must be signed before your manuscript can be sent to production. A link to the electronic license to publish form will be sent to the corresponding author only. Please take a moment to check your funder requirements before choosing the appropriate license.****

Thank you for this interesting contribution, we look forward to publishing your paper in Journal of Cell Biology.

Sincerely,

Arshad Desai, PhD
Monitoring Editor

Andrea L. Marat, PhD
Senior Scientific Editor

Journal of Cell Biology